# RETHINKING THE SOLUTION TO CURSE OF DIMENSIONALITY ON RANDOMIZED SMOOTHING

## ABSTRACT

Randomized Smoothing (RS) is currently a scalable certified defense method providing robustness certification against adversarial examples. Although significant progress has been achieved in providing defenses against $\ell_p$ adversaries, early investigations found that RS suffers from the curse of dimensionality, indicating that the robustness guarantee offered by RS decays significantly with increasing input data dimension. Double Sampling Randomized Smoothing (DSRS) is the state-of-the-art method that provides a theoretical solution to the curse of dimensionality under concentration assumptions on the base classifier. However, we speculate the solution to the curse of dimensionality can be deepened from the perspective of the smoothing distribution. In this work, we further address the curse of dimensionality by theoretically showing that some Exponential General Gaussian (EGG) distributions with the exponent $\eta$ can provide $\Omega(\sqrt{d})$ lower bounds for the $\ell_2$ certified radius with tighter constant factors than DSRS. Our theoretical analysis shows that the lower bound improves with monotonically decreasing $\eta \in (0, 2)$. Intriguingly, we observe a contrary phenomenon that EGG provides greater certified radii at larger $\eta$, on real-world tasks. Further investigations show these discoveries are not contradictory, which are in essence dependent on whether the assumption in DSRS absolutely holds. Our experiments on real-world datasets demonstrate that EGG distributions bring significant improvements for certified accuracy, up to 4%-6% on ImageNet. Furthermore, we also report the performance of Exponential Standard Gaussian (ESG) distributions on DSRS.

## 1 INTRODUCTION

Deep neural networks (DNNs) have achieved great success in various applications. However, DNNs are susceptible to adversarial perturbations in their inputs. To tackle the problem of adversarial attacks, a series of empirical defenses, such as adversarial training (Goodfellow et al., 2015; Kurakin et al., 2017; Madry et al., 2018), have been proposed. Nevertheless, this strategy quickly evolved into an *arms race* because no matter how robust the DNNs are, well-crafted adversarial examples are capable of bypassing the defenses (Carlini & Wagner, 2017; Uesato et al., 2018; Athalye et al., 2018). Recently, researchers proposed and developed certified defenses (Wong & Kolter, 2018; Wong et al., 2018; Raghunathan et al., 2018), a series of methodologies that assure that adversarial examples take zero measure within some neighborhood of clean examples. Among certified defense methods, randomized smoothing (RS) (Lecuyer et al., 2019; Li et al., 2019; Cohen et al., 2019) gains in popularity since it can provide scalable robustness certifications for black-box functions. Cohen et al. (2019) first introduced the Neyman-Pearson (NP) lemma into the certification, which provided tight $\ell_2$ certified radii for linear classifiers. Later, a series of attempts further extended the certification process of RS using functional optimization frameworks (Zhang et al., 2020; Dvijotham et al., 2020).

However, though RS certifies the $\ell_p$ radius at a low cost, it suffers from the problem of the curse of dimensionality (Yang et al., 2020; Blum et al., 2020; Kumar et al., 2020), meaning the maximal certifiable radius shrinks with the increasing input dimension of the base classifiers when $p > 2$. Despite being noticed, only a few works is committed to solving the issue of the curse of dimensionality. Recently, Li et al. (2022) proposed a double-sampling randomized smoothing (DSRS) framework that for the first time addresses the curse of dimensionality theoretically by introducing

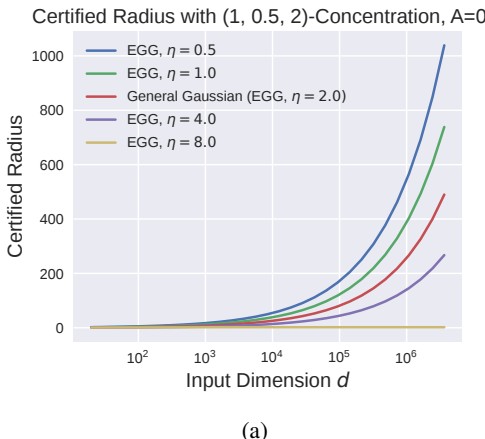
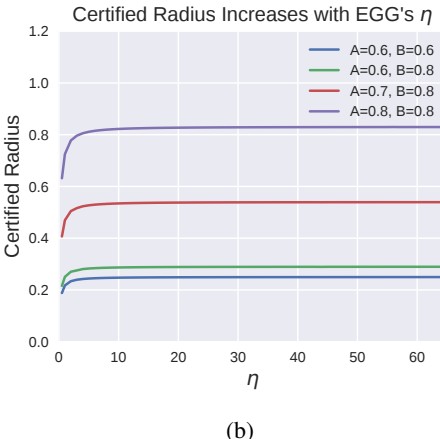

(a)                                          (b)

Figure 1: Numerical simulation results for DSRS certification by EGG. **(a).** This subfigure shows $\ell_2$ certified radius w.r.t. $d$ under a concentration property (i.e., $B = 1$). There is a clear monotonous decreasing trend for certified radius w.r.t. $\eta$. **(b).** When $B < 1$, the $\ell_2$ certified radius increases monotonically with $\eta$ of EGG, under different simulated $(A, B)$ values for ImageNet. Note that an EGG with $\eta = 2$ is the General Gaussian. For definitions of $A$ and $B$, see Equation 4.

an additional smoothing distribution. Under a concentration assumption that the adversarial examples take zero measure within a restricted neighborhood around the original input, DSRS certifies $\ell_2$ radii with an $\Omega(\sqrt{d})$ lower bound when taking the General Gaussian distribution as the smoothing distribution and its truncated counterpart as the additional distribution.

In this work, we further address the problem of the curse of dimensionality by tightening the DSRS lower bound via better smoothing distributions. Our theoretical analysis shows some EGG distributions provide much tighter theoretical lower bounds than General Gaussian, the distribution used in the SOTA solution to the curse of dimensionality. Specifically, the smaller the exponent $\eta$ in EGG, the better the lower bound. Nevertheless, experiments on real-world tasks give a seemingly opposite conclusion, that the certified radius provided by EGG is greater at greater $\eta$. Our further study demonstrates that these phenomena are not paradoxical. From the numerical simulation experiments, we find that the soundness of the concentration assumption significantly impacts the certified radius. In brief, if the assumption strictly holds, then the theoretical conclusion is valid. Otherwise, the experimental conclusion takes effect (See Figure 1). Moreover, we also investigate the experimental performance of ESG distributions, which perform as great as the Gaussian distribution in providing certified radii under the single-sampling setting. In summary, injecting EGG into DSRS provides augmented results than the SOTA method both theoretically and experimentally, though in slightly different mechanisms. Overall, this work greatly enriches the current solution to the curse of dimensionality and provides distributions with good properties for training certifiably robust base classifiers. Our main contributions include:

- Theoretically, EGG distributions with $\eta \in (0, 2)$ can certify $\ell_2$ radii at $\Omega(\sqrt{d})$ lower bounds with much tighter constant factors than the SOTA method, for base classifiers satisfying proper concentration assumptions. Our conclusion essentially implies the current solution to the curse of dimensionality can be systematically enhanced by introducing EGG distributions.

- Experiments on real-world datasets and numerical simulation demonstrate that the certified radius provided by EGG monotonically increases with $\eta$. On ImageNet, the increase of certified accuracy brought by EGG can reach 6.4% compared to the baseline. Additionally, we show that the certification provided by ESG keeps almost invariant to $\eta$, which ties the state-of-the-art.

## 2   RELATED WORK

Randomized smoothing was first proposed as an extension for differential privacy, which provides a certified robustness bound for classifiers (Lecuyer et al., 2019). Subsequently, a series of im-

provements were made to obtain tighter $\ell_2$ norm certificates through the Rényi divergence and the Neyman-Pearson Lemma (Li et al., 2019; Cohen et al., 2019). Furthermore, there are methods modelling the certification process as functional optimization problems (Zhang et al., 2020; Dvijotham et al., 2020). A line of work focuses on extending the robustness certification from $\ell_2$-only to certifications against adversaries with other $\ell_p$ norms. $\ell_0$ radius certification was made possible by constructing Neyman-Pearson sets for discrete random variables (Lee et al., 2019; Jia et al., 2020). Different to $\ell_2$ certified regions, the $\ell_1$ certified region is asymmetric in the space, which poses a new challenge to the certification algorithm. By perturbing input data under other noise distributions, such as Laplace and uniform distributions, the previous works have obtained $\ell_1$ certificates (Teng et al.; Yang et al., 2020; Levine & Feizi, 2021). In addition to the anisotropy, some work discovered the phenomenon of the curse of dimensionality when trying to certify against $\ell_p$ adversaries whose $p > 2$ (Yang et al., 2020; Blum et al., 2020; Kumar et al., 2020). To deal with this issue, a recent work offered an $\Omega(1)$ bound w.r.t. the input dimension for $\ell_\infty$ certified radius by introducing an additional smoothing distribution (Li et al., 2022), which breaks the curse of dimensionality theoretically for the first time. Lately, a study found that the computation of certified radius can be improved by incorporating the geometric information from adjacent decision domains of the same class (Cullen et al., 2022).

There were also investigations using anisotropic or sample-specific smoothing noise to improve the certification (Eiras et al., 2022; Sukenik et al., 2022). In addition, a chain of work focused on improving the performance of base classifiers through adopting better training techniques (Salman et al., 2019; Zhai et al., 2019; Jeong & Shin, 2020; Jeong et al., 2021), or introducing denoising modules (Salman et al., 2020; Carlini et al., 2023; Wu et al.). There were also attempts to adapt RS to broader application scenarios. RS was extended into defenses against adversarial patches (Levine & Feizi, 2020; Yatsura et al., 2022) and semantic perturbations (Li et al., 2021; Hao et al., 2022; Alfarra et al., 2022; Pautov et al., 2022). Moreover, RS has been shown to be capable of providing provable guarantees for tasks such as object detection, semantic segmentation, and watermarking (Chiang et al., 2020; Fischer et al., 2021; Bansal et al., 2022).

## 3 PRELIMINARIES

**Problem setup.** We focus on the typical multi-class classification task in this work. Let $x_i \in \mathbb{R}^d$ be the $i$-th $d$-dimensional data point and $y_i \in \mathcal{Y} = \{1, 2, \cdots, N\}$ be its corresponding ground-truth label. We assume a dataset $\mathcal{J}$ contains data pairs $(x_i, y_i), i \in \mathbb{N}_{\leq n}$ that are i.i.d drawn from the sample space $\mathbb{R}^d \times \mathcal{Y}$. A $N$-way *base classifier* (neural networks in this work) $f : \mathbb{R}^d \to \mathcal{Y}$ can be trained to maximize the empirical classification accuracy $\frac{1}{|\mathcal{J}|} \sum_{(x,y) \in \mathcal{J}} \mathbb{1}_{f(x)=y}$ on dataset $\mathcal{J}$. Given an arbitrary data point $x$ and its label $y$, it is known that in practice, most classifiers trained using standard training techniques are susceptible to adversarial perturbations within a small $\epsilon$-ball.

**Randomized smoothing.** To mitigate the adversarial perturbations, RS has been employed as a certified defense method that can supply a robustness guarantee on the correctness of the classification results from classifiers. It provides the robustness certification for the base classifier $f$ by constructing its smoothed counterpart $\bar{f}$. Given a base classifier $f$, an input $x_0 \in \mathbb{R}^d$ and a smoothing distribution $\mathcal{D}$, the *smoothed classifier* is defined as follows:

$$\bar{f}_{\mathcal{D}}(x_0) = \underset{a \in \mathcal{Y}}{\operatorname{argmax}} \, \mathbb{P}_{z \sim \mathcal{D}} \{ f(x_0 + z) = a \}. \quad (1)$$

With the definition of smoothed classifier, we can evaluate its $\ell_p$ certified robustness by $\ell_p$ certified radius defined below.

**Definition 1.** *Given a base classifier $f : \mathbb{R}^d \to \mathcal{Y}$, its smoothed counterpart $\bar{f}_{\mathcal{D}} : \mathbb{R}^d \to \mathcal{Y}$ under a distribution $\mathcal{D}$ and a labeled example $(x_0, y_0) \in \mathbb{R}^d \times \mathcal{Y}$. Then $r$ is called $\ell_p$ **certified radius** of $\bar{f}_{\mathcal{D}}$ if*

$$\forall x, \, \|x - x_0\|_p < r, \, \bar{f}_{\mathcal{D}}(x) = y_0. \quad (2)$$

The state-of-the-art method to compute the certified radius for RS is based on the Neyman-Pearson Lemma, which simply provides tight certifications for classifiers through sampling results from a single distribution.

**Double sampling randomized smoothing.** Essentially, the DSRS framework is based on a generalization of the Neyman-Pearson Lemma (Chernoff & Scheffe, 1952) that introduces one more

subjection into the system. In short, DSRS provides a method to construct the most conservative classifier by the prediction probabilities of the base classifier under two different distributions. In this work, we abstract the classifier to be optimized into a *true binary classifier*. Given a base classifier $f : \mathbb{R}^d \to \mathcal{Y}$, we call $\tilde{f}_{x_0} : \mathbb{R}^d \to \{0, 1\}$ a true binary classifier of $f$ if for $(x_0, y_0) \in \mathbb{R}^d \times \mathcal{Y}$ and random vector $z \in \mathbb{R}^d$:

$$\tilde{f}_{x_0}(z) = \mathbb{1}_{f(x_0+z)=y_0}. \tag{3}$$

Then the problem of DSRS can be formulated as a functional optimization problem for the true binary classifier $\tilde{f}_{x_0}$. Given $\tilde{f}_{x_0}$ from the function space $\mathcal{F} = \{h(x) \mid h(x) \in [0, 1], \forall x \in \mathbb{R}^d\}$, the worst-case expected prediction over randomization is given below:

$$
\begin{aligned}
\min_{\tilde{f}_{x_0} \in \mathcal{F}} \quad & \mathbb{E}_{z \sim \mathcal{P}} \left( \tilde{f}_{x_0}(\delta + z) \right), \\
\text{s.t.} \quad & \mathbb{E}_{z \sim \mathcal{P}} \left( \tilde{f}_{x_0}(z) \right) = A, \\
& \mathbb{E}_{z \sim \mathcal{Q}} \left( \tilde{f}_{x_0}(z) \right) = B.
\end{aligned}
\tag{4}
$$

In the equations (4), $A, B \in [0, 1]$ are probabilities that the base classifier $f$ outputs the right label $y_0$ for example $x_0$ under noise distributions $\mathcal{P}$, $\mathcal{Q}$, respectively. Practically, they are usually estimated by Monte Carlo sampling. For an example $x_0$ and a given combination of $\mathcal{P}$, $\mathcal{Q}$, $A$ and $B$, we are able to derive a unique $\tilde{f}_{x_0}$. Finally, by finding the maximum $\|\delta\|_2$ which satisfies $\mathbb{P}_{z \sim \mathcal{P}}\{f(x_0+\delta+z) = y_0\} > 0.5$, we obtain the certified radius of $x_0$. We leave further preliminaries to Appendix F.

## 4 EXPONENTIAL GAUSSIAN DISTRIBUTIONS, WHY AND HOW?

The curse of dimensionality is a daunting problem in RS-based robustness certifications. One of the state-of-the-art theoretical solutions for this problem is to take the General Gaussian distribution as the smoothing distribution under the DSRS framework given the concentration assumption (Li et al., 2022). Unlike the Gaussian distribution abstracted from natural laws, General Gaussian does not contain any physical meaning as a pure mathematical generalization for Gaussian, which may imply some loss of intrinsic optimality. Intuitively, generalizing the General Gaussian distribution from an exponential perspective preserves the essential $r^{-2k}$ term, and is thereby likely to give better lower bounds for the certified radius in DSRS. Our conjecture is confirmed by theoretical analyses.

In this section, we show the definition of EGG distributions and prove that some of them indeed provide better theoretical lower bounds than that of the General Gaussian distribution. In other words, in addition to the General Gaussian distribution, several members of EGG distributions show stronger ability in providing the $\Omega(\sqrt{d})$ lower bound for the $\ell_2$ certified radius under the DSRS framework, which largely broadens the current solution to the curse of dimensionality in RS. Additionally, we also investigate ESG distributions under the DSRS framework, which are exponential generalizations to the Gaussian distribution. Our experimental results show that the robustness offered by ESG is insensitive to the exponent, which proves the optimality of Gaussian from one perspective.

### 4.1 FAMILY OF SMOOTHING DISTRIBUTIONS AND NOTATIONS

We start by introducing the EGG distributions and the ESG distributions used throughout the paper. Under the DSRS framework, when using EGG as the smoothing distribution, the Truncated Exponential General Gaussian (TEGG) distribution is employed as the additional distribution. Similarly, Truncated Exponential Standard Gaussian (TESG) serves as the additional distribution when adopting ESG as the smoothing distribution. We let $\mathcal{G}(\sigma, \eta, k)$ and $\mathcal{S}(\sigma, \eta)$ be the probability density functions (PDFs) of EGG and ESG, respectively. Table 1 shows the definitions and basic properties of the distributions. More details for the distributions are deferred to Appendix A.

In the table, $r, T, \sigma, \eta \in \mathbb{R}_+$ and $d \in \mathbb{N}_+$. $\Gamma(\cdot)$ is the gamma function. Following the convention in the previous studies (Yang et al., 2020; Li et al., 2022), we set the substitution variance to ensure $\mathbb{E}r^2$ is a constant for all distributions. We let $\sigma_g$ and $\sigma_s$ be the substitution variances of EGG and ESG, respectively. The CDFs of the beta distribution $\text{Beta}(\alpha, \alpha)$ and the gamma distribution $\Gamma(\alpha, 1)$ are denoted respectively by $\Psi_\alpha(\cdot)$ and $\Lambda_\alpha(\cdot)$. We write $\phi_g(r)$ and $\phi_s(r)$ corresponding to the PDFs of $\mathcal{G}(\sigma, \eta, k)$ and $\mathcal{S}(\sigma, \eta)$ respectively.

Table 1: Properties and definitions of distributions.

| Distribution | PDF | Notation | Substitution Variance |
|---|---|---|---|
| Standard Gaussian | $\propto \exp\left(-\frac{r^2}{2\sigma^2}\right)$ | $\mathcal{N}(\sigma)$ | $\sigma$ |
| Exponential Standard Gaussian | $\propto \exp\left(-\frac{r^\eta}{2\sigma_s^\eta}\right)$ | $\mathcal{S}(\sigma,\eta)$ | $\sigma_s = 2^{-\frac{1}{\eta}}\sqrt{\frac{d\Gamma(\frac{d}{\eta})}{\Gamma(\frac{d+2}{\eta})}}\sigma$ |
| Truncated Exponential Standard Gaussian | $\propto \exp\left(-\frac{r^\eta}{2\sigma_s^\eta}\right)\mathbb{1}_{r\leq T}$ | $\mathcal{S}_t(\sigma,\eta,T)$ | $\sigma_s = 2^{-\frac{1}{\eta}}\sqrt{\frac{d\Gamma(\frac{d}{\eta})}{\Gamma(\frac{d+2}{\eta})}}\sigma$ |
| Exponential General Gaussian | $\propto r^{-2k}\exp\left(-\frac{r^\eta}{2\sigma_g^\eta}\right)$ | $\mathcal{G}(\sigma,\eta,k)$ | $\sigma_g = 2^{-\frac{1}{\eta}}\sqrt{\frac{d\Gamma(\frac{d-2k}{\eta})}{\Gamma(\frac{d-2k+2}{\eta})}}\sigma$ |
| Truncated Exponential General Gaussian | $\propto r^{-2k}\exp\left(-\frac{r^\eta}{2\sigma_g^\eta}\right)\mathbb{1}_{r\leq T}$ | $\mathcal{G}_t(\sigma,\eta,k,T)$ | $\sigma_g = 2^{-\frac{1}{\eta}}\sqrt{\frac{d\Gamma(\frac{d-2k}{\eta})}{\Gamma(\frac{d-2k+2}{\eta})}}\sigma$ |

## 4.2 THEORETICAL ANALYSIS ON THE LOWER BOUND

It is proved that taking General Gaussian distribution as the smoothing distribution for DSRS provides an $\Omega(\sqrt{d})$ lower bound for the $\ell_2$ certified radius (Li et al., 2022). This bound can be converted to an $\Omega(1)$ lower bound for the $\ell_\infty$ certified radius, which breaks the curse of dimensionality in the certification against $\ell_\infty$-adversaries under high-dimensional settings (Kumar et al., 2020). Nevertheless, specialized studies on the curse of dimensionality are in great lack, on which numerous questions can be raised. For instance, *does the present solution end the curse of dimensionality? If not, how to improve it? Can the theoretical solution be transferred to real scenes?* In this work, we investigate these questions by employing exponential general Gaussian distributions in DSRS, which not only comprehensively ameliorates the current solution, but also reveals its potential limitations.

We first define a $(\sigma, p, \eta)$-concentration property to start our analysis:

**Definition 2.** *($(\sigma, p, \eta)$-Concentration) Let $f : \mathbb{R}^d \to \mathcal{Y}$ be an arbitrarily determined base classifier, $(x_0, y_0) \in \mathbb{R}^d \times \mathcal{Y}$ be a labeled example. We say $f$ satisfies $(\sigma, p, \eta)$-concentration assumption at $(x_0, y_0)$ if for $p \in (0, 1)$ and $T$ satisfying*

$$\mathbb{P}_{z\sim\mathcal{S}(\sigma,\eta)}\{\|z\|_2 \leq T\} = p, \tag{5}$$

*$f$ satisfies*

$$\mathbb{P}_{z\sim\mathcal{S}(\sigma,\eta)}\{f(x_0 + z) = y_0 \mid \|z\|_2 \leq T\} = 1. \tag{6}$$

Namely, if $f$ satisfies the $(\sigma, p, \eta)$-concentration assumption for some specific $(x_0, y_0)$, the random perturbation $z \sim \mathcal{S}(\sigma, \eta)$ that makes $f(x_0 + z) \neq y_0$ within the $T$-radius $\ell_2$ sphere takes zero measure in $\mathbb{R}^d$. Despite the assumption being seemingly harsh, it is satisfied approximately in the light of observations from Li et al. (2022), where a considerable proportion of points in the dataset are predicted to be nearly completely correct under noise on well-trained base classifiers.

For simplicity, we first let the base classifier $f$ satisfy $(\sigma, p, 2)$-concentration property, the original assumption used in Li et al. (2022). Here we show a theorem that some EGG distributions with their corresponding truncated counterparts can certify the $\ell_2$ radius with $\Omega(\sqrt{d})$ lower bounds by DSRS. In other words, the current solution to the curse of dimensionality provided by General Gaussian can be significantly augmented, that all EGG with $\eta \in (0, 2)$ have the potential to break the curse, even with better lower bounds.

**Theorem 1** (Lower Bound for the Certified Radius with EGG). *Let $d \in \mathbb{N}_+$ be a sufficiently large input dimension, $(x_0, y_0) \in \mathbb{R}^d \times \mathcal{Y}$ be a labeled example and $f : \mathbb{R}^d \to \mathcal{Y}$ be a base classifier which satisfies $(\sigma, p, 2)$-concentration property w.r.t. $(x_0, y_0)$. For the DSRS method, let $\mathcal{P} = \mathcal{G}(\sigma, \eta, k)$ be the smoothing distribution to give a smoothed classifier $\bar{f}_\mathcal{P}$, and $\mathcal{Q} = \mathcal{G}_t(\sigma, \eta, k, T)$ be the additional distribution with $T = \sigma\sqrt{2\Lambda_{\frac{d}{2}}^{-1}(p)}, d - 2k \in [1, 30] \cap \mathbb{N}$ and $\eta \in \{1, \frac{1}{2}, \frac{1}{3}, \cdots, \frac{1}{50}\}$. Then for the smoothed classifier $\bar{f}_\mathcal{P}(x)$, the certified $\ell_2$ radius satisfies*

$$r_{DSRS} \geq 0.02\sigma\sqrt{d}. \tag{7}$$

We briefly summarize the proof here and leave the details in Appendix C. In essence, Theorem 1 is generalizing Theorem 2 of Li et al. (2022) to EGG. Practically, it is intractable to derive a solution for

the certified radius $r_{DSRS}$. Therefore, the problem is simplified by introducing the concentration assumption, whereupon we calculate the solution for Problem (4) only considering the truncated distribution $\mathcal{Q}$. In short, the certification of the radius $\rho$ is dependent on the discriminant

$$\mathbb{E}_{u \sim \Gamma(\frac{d-2k}{\eta}, 1)} \Psi_{\frac{d-1}{2}} \left( \frac{T^2 - (\sigma_g(2u)^{\frac{1}{\eta}} - \rho)^2}{4\rho\sigma_g(2u)^{\frac{1}{\eta}}} \right) - \frac{1}{2} \geq 0. \tag{8}$$

With this formulation, whether a radius $\rho$ can be certified can directly be judged since the LHS of Equation (8) is a function of $\rho$. We take the proof of Theorem 1 as an example. If we substitute $\rho = 0.02\sigma\sqrt{d}$ into Equation (8), and the LHS of (8) is positive, then radius $\rho$ is certified. Otherwise, if the value is negative, $\rho$ is not a certified radius. For the derivation and proof of Problem (8), please see the proof of Lemma C.4 in the appendix.

Plus, we explore the effects of EGG under $(\sigma, p, \eta)$-concentration assumption, where we can construct $\Omega(d^{1/\eta})$ lower bounds for the certified radius. However, though it is formally tighter than $\Omega(\sqrt{d})$, we find these two lower bounds are fundamentally equivalent. See Theorem 4 in Appendix C.3.

**Study on the constant factor.** Besides certifying the lower bound radius $0.02\sigma\sqrt{d}$, the proof of Theorem 1 naturally contains an approach to determine tight constant factors for each EGG distribution. In fact, the value of the LHS of Equation (8) monotonically decreases with $\rho$, meaning that performing a simple binary search on $\rho$ can provide the accurate certified radius. Therefore, we consider parameterizing the radius $\rho$ into $\mu\sigma\sqrt{d}$, thus binary searching on the constant factor will derive the tight constant factor $\mu$ for the $\Omega(\sqrt{d})$ lower bound (Algorithm 1). We report computational results for $\mu$ in Figure 2, where for values of $d-2k$ except 1, the tight constant factor $\mu$ increases monotonically as the $\eta$ decreases. Essen-

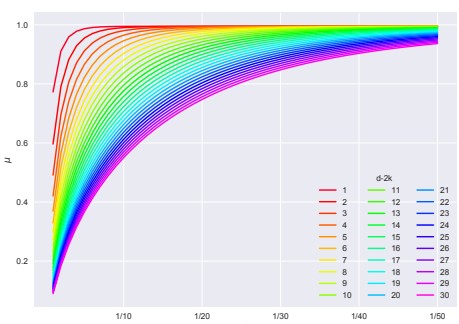

Figure 2: Tight factor $\mu$ grows as $\eta$ shrinks, for most $d - 2k \in [1, 30] \cap \mathbb{N}$.

tially, these results demonstrate that the solution to the curse of dimensionality provided by Li et al. (2022) (with General Gaussian, $\eta = 2$ in EGG) can be improved by choosing smaller $\eta \in (0, 2)$, for most $d - 2k \in [1, 30] \cap \mathbb{N}$.

Overall, theoretical analysis above shows taking EGG as the smoothing distribution in DSRS, we are likely to obtain much tighter lower bounds for the certified radius on base classifiers, which substantially improved the current solution to the curse of dimensionality from Li et al. (2022). Following the analysis, *will the EGG distribution behave as excellent as theoretical cases in certifying real-world tasks?*

## 4.3 Computations for Exponential Gaussian distributions on DSRS

To evaluate the performance of EGG/ESG on real-world datasets, we consider solving Problem (9), the strong dual problem of Problem (4):

$$\max_{\nu_1, \nu_2 \in \mathbb{R}} \quad \mathbb{P}_{z \sim \mathcal{P}+\delta}\{p(z-\delta) + \nu_1 p(z) + \nu_2 q(z) < 0\},$$
$$\text{s.t.} \quad \mathbb{P}_{z \sim \mathcal{P}}\{p(z-\delta) + \nu_1 p(z) + \nu_2 q(z) < 0\} = A, \tag{9}$$
$$\mathbb{P}_{z \sim \mathcal{Q}}\{p(z-\delta) + \nu_1 p(z) + \nu_2 q(z) < 0\} = B.$$

We do not elaborate on the solution process since previous works (Yang et al., 2020; Li et al., 2022) have done well. Herein, we directly give solutions to Problem (9) for EGG and ESG. Overall, the case of EGG is relatively a straightforward generalization of General Gaussian, while the derivation for ESG includes nontrivial branches. See details in Appendix B. In the next section, we show experiments that verifying the computations for EGG and ESG.

## 5 EXPERIMENTS

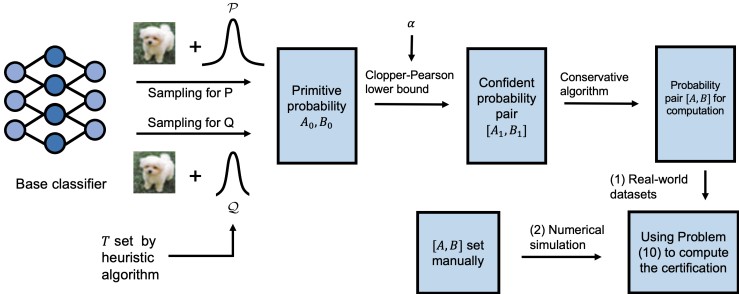

Figure 3: Illustration for experiments.

In this section, we report the effects of EGG and ESG distributions on certified radius under the DSRS framework. Figure 3 illustrates the pipeline of our experiments, which includes evaluations on real-world datasets and numerical simulations. Here we are mainly focused on real-world datasets and leave details for numerical simulation experiments, which comprehensively display the properties of distribution families we use, to Appendix H. For the case of real-world datasets, $A$ and $B$ in Problem (9) are initially from Monte Carlo sampling results on a given base classifier, then reduced to respective Clopper-Pearson confidence interval (Clopper & Pearson, 1934), and finally determined by a conservative algorithm (Li et al., 2022). The procedure for real-world datasets can also be seen in Algorithm 2 in Appendix G.

**Experimental setups.** All base classifiers used in this work are trained by CIFAR-10 (Krizhevsky et al., 2009) or ImageNet (Russakovsky et al., 2015), taking EGG with $\eta = 2$ as the noise distribution. The settings of hyperparameters for EGG and ESG are slightly different. For EGG experiments, we take $\mathcal{P} = \mathcal{G}(\sigma, \eta, k)$ and $\mathcal{Q} = \mathcal{G}_t(\sigma, \eta, k, T)$. To display the increasing trends more clearly, we choose $\eta \in \{0.25, 0.5, 1.0, 2.0, 4.0, 8.0\}$ for them. For ESG, each DSRS certification is computed by taking $\mathcal{P} = \mathcal{S}(\sigma, \eta)$ as the smoothing distribution and $\mathcal{Q} = \mathcal{S}_t(\sigma, \eta, T)$ as the additional distribution. We choose $\eta \in \{1.0, 2.0, 4.0, 8.0\}$ as the exponent of the ESG distributions. We leave the details of setting other parameters to Appendix G.2.

**Evaluation metrics.** We consider the $\ell_2$ certified radius in all experiments. To evaluate the certified robustness of smoothed classifiers, we take *certified accuracy* $\triangleq CA(r, \mathcal{J})$ at radius $r$ on test dataset $\mathcal{J}$ as the basic metric (Cohen et al., 2019; Zhang et al., 2020; Li et al., 2022). For $(x_i, y_i)$ in $\mathcal{J}$, if the certified radius computed by a certification method (e.g. NP, DSRS) is $r_i$, and the output for $x_i$ through the smoothed classifier is $y_i$, we say $(x_i, y_i)$ is certified accurate at radius $r_i$ for the smoothed classifier. On this basis, $CA(r, \mathcal{J})$ is the ratio of examples in $\mathcal{J}$ whose certified radius $r_i \geq r$. Defined on the certified accuracy, *Average Certified Radius* (ACR) (Zhai et al., 2019; Jeong & Shin, 2020) is the main metric that we use to show results of different distributions for real-world datasets. Formally, we have $ACR \triangleq \int_{r \geq 0} CA(r, \mathcal{J}) \cdot \mathrm{d}r$. All our DSRS certification results are compared with the baselines from the Neyman-Pearson method, see a brief introduction in Appendix F.1.

**Soundness for numerical integration.** Considering the extensive use of numerical integration in this work, we comprehensively test the precision of our integral results. For EGG distributions, we inherit the method from Yang et al. (2020) and Li et al. (2022), where we compute the integrals on the interval $(0, +\infty)$ by `scipy` package. As for ESG distributions, we implement an integration algorithm for ESG separately since `scipy` is not applicable to the gamma distributions with large parameters. We notice the following lemma:

**Lemma 5.1.** *(Bilateral Concentration of the Gamma Distribution) Let $X \sim \Gamma(\frac{d}{\eta}, 1)$ be a random variable, where $\eta \in \mathbb{R}_+, d \in \mathbb{N}_+$. Let $\iota = \frac{\eta}{\epsilon^2 d}$, then for any $\iota \in (0, 1)$, the following inequality holds:*

$$\mathbb{P}\{(1 - \epsilon)\frac{d}{\eta} < X < (1 + \epsilon)\frac{d}{\eta}\} \geq 1 - \iota. \tag{10}$$

The proof is quite succinct, see Appendix G.5. Lemma 5.1 indicates that the integral interval $(0, +\infty)$ can be substituted by $(\max\{0, (1 - \epsilon)\frac{d}{\eta}\}, (1 + \epsilon)\frac{d}{\eta})$ under tolerable and controllable error.

Table 2: Certified radius at $r$ for standardly augmented models, certified by EGG under DSRS

| Dataset | Method | Certified accuracy at $r$ | | | | | | | | | | | | | |
|---|---|---|---|---|---|---|---|---|---|---|---|---|---|---|---|
| | | 0.25 | 0.50 | 0.75 | 1.00 | 1.25 | 1.50 | 1.75 | 2.00 | 2.25 | 2.50 | 2.75 | 3.00 | 3.25 | 3.50 |
| CIFAR10 | EGG, $\eta=0.25$ | 54.2% | 37.6% | 23.5% | 16.5% | 9.4% | 4.5% | 0.5% | 0.1% | 0.0% | 0.0% | 0.0% | 0.0% | 0.0% | 0.0% |
| | EGG, $\eta=0.5$ | 55.5% | 40.4% | 25.2% | 19.1% | 13.4% | 8.5% | 5.5% | 2.0% | 0.4% | 0.1% | 0.0% | 0.0% | 0.0% | 0.0% |
| | EGG, $\eta=1.0$ | 56.3% | 41.7% | 28.2% | 20.0% | 15.1% | 10.5% | 7.1% | 4.2% | 1.9% | 0.9% | 0.1% | 0.0% | 0.0% | 0.0% |
| | EGG, $\eta=2.0$ | 56.7% | 42.4% | 29.3% | 20.2% | 15.7% | 11.5% | 8.0% | 5.5% | 2.6% | 1.5% | 0.6% | 0.1% | 0.0% | 0.0% |
| | EGG, $\eta=4.0$ | 57.5% | 42.5% | 30.0% | 20.2% | 15.9% | 12.2% | 8.5% | 6.5% | 3.4% | 1.8% | 0.9% | 0.4% | 0.0% | 0.0% |
| | DSRS (Li et al., 2022) | 57.4% | **42.7%** | 30.6% | 20.6% | **16.1%** | **12.5%** | 8.4% | 6.4% | 3.5% | 1.8% | 0.7% | 0.1% | / | / |
| | Ours (EGG, $\eta=8.0$) | **57.6%** | 42.5% | **30.9%** | **20.6%** | 15.8% | 12.3% | **8.6%** | **6.6%** | **3.7%** | **2.1%** | **1.1%** | **0.5%** | **0.2%** | 0.0% |
| ImageNet | EGG, $\eta=0.25$ | 53.8% | 41.4% | 28.4% | 20.1% | 7.1% | 0.8% | 0.0% | 0.0% | 0.0% | 0.0% | 0.0% | 0.0% | 0.0% | 0.0% |
| | EGG, $\eta=0.5$ | 54.9% | 46.3% | 36.4% | 26.3% | 22.1% | 15.2% | 8.7% | 3.1% | 0.8% | 0.0% | 0.0% | 0.0% | 0.0% | 0.0% |
| | EGG, $\eta=1.0$ | 57.0% | 47.8% | 39.9% | 32.8% | 24.9% | 22.0% | 18.5% | 13.1% | 9.2% | 5.0% | 2.1% | 0.5% | 0.0% | 0.0% |
| | DSRS (Li et al., 2022) (EGG, $\eta=2.0$) | 58.4% | 48.5% | 41.5% | 35.2% | 28.9% | 23.3% | 21.3% | 18.8% | 14.1% | 11.1% | 8.9% | 6.1% | 2.2% | 1.4% |
| | EGG, $\eta=4.0$ | 58.7% | 49.9% | 42.6% | 36.4% | 31.0% | 23.9% | 22.3% | 20.2% | 17.3% | 13.2% | 10.7% | 9.2% | 6.8% | 4.0% |
| | Ours (EGG, $\eta=8.0$) | **59.1%** | **50.8%** | **42.9%** | **36.8%** | **31.8%** | **24.6%** | **22.6%** | **20.7%** | **18.9%** | **14.5%** | **11.7%** | **10.1%** | **8.6%** | **5.2%** |

Table 3: Certified radius at $r$ for standardly augmented models, certified by ESG under DSRS

| Dataset | Method | Certified accuracy at $r$ | | | | | | | | | | | | | |
|---|---|---|---|---|---|---|---|---|---|---|---|---|---|---|---|
| | | 0.25 | 0.50 | 0.75 | 1.00 | 1.25 | 1.50 | 1.75 | 2.00 | 2.25 | 2.50 | 2.75 | 3.00 | 3.25 | 3.50 |
| CIFAR10 | ESG, $\eta=1.0$ | 57.6% | 42.6% | 31.3% | 21.5% | 15.8% | 12.8% | 8.6% | 6.8% | 4.3% | 2.3% | 1.3% | 0.8% | 0.3% | 0.1% |
| | ESG, $\eta=2.0$ (Gaussian) | 57.6% | 42.6% | 31.6% | 21.5% | 15.8% | 12.7% | 8.8% | 6.8% | 4.5% | 2.4% | 1.3% | 0.7% | 0.2% | 0.2% |
| | ESG, $\eta=4.0$ | 57.6% | 42.6% | 31.3% | 21.5% | 15.9% | 12.9% | 8.6% | 6.9% | 4.3% | 2.4% | 1.3% | 0.8% | 0.2% | 0.1% |
| | ESG, $\eta=8.0$ | 57.8% | 42.6% | 31.6% | 21.6% | 15.9% | 12.9% | 8.9% | 6.7% | 4.2% | 2.4% | 1.3% | 0.9% | 0.2% | 0.1% |
| ImageNet | ESG, $\eta=1.0$ | 59.6% | 51.5% | 43.2% | 37.9% | 33.0% | 26.8% | 23.1% | 21.5% | 19.9% | 17.4% | 13.8% | 11.5% | 10.3% | 7.7% |
| | ESG, $\eta=2.0$, (Gaussian) | 59.6% | 51.6% | 43.1% | 38.0% | 32.9% | 26.9% | 23.1% | 21.5% | 19.7% | 17.4% | 13.6% | 11.4% | 10.1% | 8.3% |
| | ESG, $\eta=4.0$ | 59.6% | 51.5% | 43.2% | 38.0% | 32.9% | 27.2% | 23.1% | 21.6% | 19.9% | 17.2% | 13.6% | 11.4% | 10.2% | 8.0% |
| | ESG, $\eta=8.0$ | 59.6% | 51.5% | 43.2% | 38.0% | 33.0% | 26.8% | 23.1% | 21.6% | 19.7% | 17.3% | 13.6% | 11.5% | 10.1% | 8.4% |

Practically, we use simple linear segmented integration to compute the certified radius for ESG distributions, and $\epsilon$ is set to $10^{-4}$. The results indicate that though we have abandoned some tiny mass of the distribution, our integral method for ESG is pretty accurate.

Additionally, to show the level of accuracy of the integral methods in this work, we borrow the approach from Yang et al. (2020), to compare the results of numerical integration and Monte Carlo simulation for the certified radius list obtained by its corresponding probability list. Overall, the orders of magnitude of error between numerical integration and 100000 Monte Carlo sampling are $10^{-3}$ for EGG and $10^{-5}$ for ESG. See details in Appendix G.6.

**Experimental results.** Table 2 and Table 3 report the maximum certified accuracy among base classifiers with $\sigma \in \{0.25, 0.50, 1.00\}$, which is widely adopted by previous work to show experimental results of certified robustness. Table 2 reveals the phenomenon that certified accuracy at $r$ increases monotonically with the $\eta$ of EGG. On both CIFAR10 and ImageNet, our strategy to use EGG with larger $\eta$ (8.0 in the tables) performs obviously better than General Gaussian (EGG with $\eta = 2.0$) used in DSRS (Li et al., 2022), which provides the state-of-the-art on ImageNet under the DSRS framework. See full experimental data of EGG on standardly augmented models in Table 11 and Table 12.

Unlike EGG, as shown in Table 3, the certification provided by ESG distributions is highly insensitive to the alternation of $\eta$. Additionally, we find this insusceptibility to $\eta$ also applies to NP certification, meaning numbers of ESG distributions are likely to give the best $\ell_2$ certified results for a given classifier, in addition to the current mainstream view that Gaussian provides the best (Yang et al., 2020). See details on the NP certification in Appendix F.1, and full experimental data of ESG in Table 13 and Table 14.

The details for ACR brought by Figure 4 further uncover the properties of our distributions in the DSRS certification. For EGG, beyond the monotonically increasing property with $\eta$, Figure 4b reveals the growth brought by DSRS relative to NP shrinks with $\eta$, despite the results of DSRS keep increasing. Furthermore, though the DSRS certifications offered by EGG exhibit incrementality with $\eta$, there is likely to be an upper bound for the certification provided by Gaussian distribution (in this work, $\eta = 2$ for ESG), which is currently recognized as the best distribution for single-distribution randomized smoothing. This is clear when we observe Figure 4a and 4c together. Besides, the DSRS certification does not work well for ESG because of the negative growth relative to NP certification. The explanation for this could be that truncated Gaussian distribution impaired the optimality of Gaussian distribution under the randomized smoothing framework.

**Analyses.** The reason for different sensitivity to $\eta$ for EGG and ESG may lie in their shape in the high-dimensional space. Zhang et al. (2020) pointed out that the Laplace distribution and the Standard Gaussian distribution (corresponds to $\eta = 1, \eta = 2$ in ESG, respectively) are concentrated on a thin sphere in the space. The EGG distributions overcome *thin-shell* property by introducing

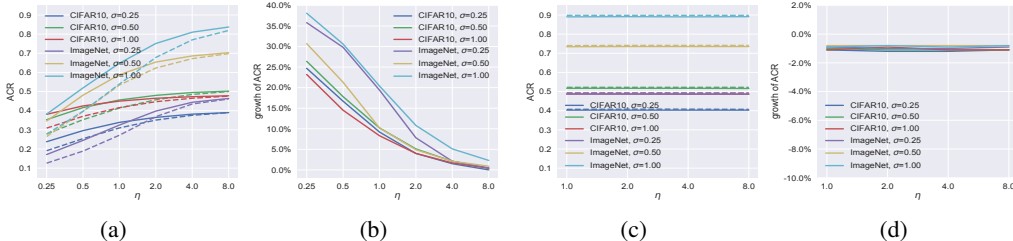

Figure 4: ACR results on real-world datasets. **(a).** ACR monotonically increases with $\eta$ in EGG. **(b).** The ACR growth gain by DSRS relative to NP shrinks with $\eta$ in EGG. **(c).** ACR keeps almost constant in ESG. **(d).** The ACR growth gain by DSRS remains almost constant in ESG. For (a) and (c), solid lines represent results from DSRS, and dotted lines represent results from NP.

the $r^{-2k}$ term, which ensures EGG provides significantly broader and more uniform coverage in the truncated critical space. In contrast, the performance of ESG presents inertia towards the change of $\eta$. We assert that the thin-shell phenomenon is common for the ESG distributions, which may make them extremely insensitive to truncating due to almost the identical local property in the thin shell.

Let us go back to the question we raised earlier: *does EGG perform great in real tasks as well?* Though our experiments clearly give an answer *yes*, reasons for the improvement have subtle differences from the theory, since we finally find EGG performs great with small $\eta$ in theory, while with large $\eta$ in practice. To understand this, we employ numerical simulations to study the effect of concentration assumption. From Figure 10, whether the concentration assumption strictly holds plays a significant role. Briefly, the hold probability for the assumption has a gap between 0.99 and 1. The theoretical bound is invalid even when the assumption holds at 0.99, meaning the current solution to the curse of dimensionality can be susceptible to relaxations on the assumption.

## 6 CONCLUSION AND DISCUSSION

We find in this paper that EGG distributions provide significant amelioration on the current solution to the curse of dimensionality on randomized smoothing. Beyond the theoretical progress, we also discover that EGG can improve the $\ell_2$ certified radii of smoothed classifiers on real datasets. Though the phenomena for the theoretical and experimental results are seemingly opposite, we conclude they are not contradictory because the concentration assumption influences the certified radii heavily. Besides, our experiments present that not only the Gaussian distribution but also ESG distributions can give the best $\ell_2$ certified radius in the NP certification.

**Limitations.** We remark that the current solution to the curse of dimensionality is still not perfect, since the concentration assumption is essentially strong for real-world settings. In practice, limited by current methods of training base classifiers, the certification provided by DSRS can not defeat the NP method in most cases. Accordingly, it would be intriguing to develop training techniques specifically for EGG distributions and/or the DSRS framework. We hope this work could enlighten the community on better addressing the curse of dimensionality in RS and building stronger certified defenses towards adversarial examples.

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

# Appendix

## Table of Contents

## A  SUPPLEMENTARY FOR DEFINITIONS OF DISTRIBUTIONS

### A.1  DERIVATION FOR PDFS

Since $\phi_g(r) \propto r^{-2k} \exp\left(-\frac{r^\eta}{2\sigma_g^\eta}\right)$, we have

$$\phi_g(r) = \frac{\eta}{2} \frac{1}{(2\sigma_g^\eta)^{\frac{d-2k}{\eta}} \pi^{\frac{d}{2}}} \frac{\Gamma(\frac{d}{2})}{\Gamma(\frac{d-2k}{\eta})} r^{-2k} \exp(-\frac{1}{2}(\frac{r}{\sigma_g})^\eta), \tag{11a}$$

$$\phi_g^{-1}(r) = \sigma_g \left( \frac{4k}{\eta} W \left( \frac{\eta}{2k} \left( \frac{\Gamma(\frac{d-2k}{\eta})}{\Gamma(\frac{d}{2})} 2^{\frac{d}{\eta}+1} \sigma_g^d \pi^{\frac{d}{2}} \frac{r}{\eta} \right)^{-\frac{\eta}{2k}} \right) \right)^{\frac{1}{\eta}}. \tag{11b}$$

In the equations above, $W(\cdot)$ is the principal branch of the Lambert W function. Let $\phi_g(r, T)$ be PDF of $\mathcal{G}_t(\sigma, \eta, k, T)$, then

$$\phi_g(r, T) = \frac{\eta}{2} \frac{1}{(2\sigma_g^\eta)^{\frac{d-2k}{\eta}} \pi^{\frac{d}{2}}} \frac{\Gamma(\frac{d}{2})}{\gamma(\frac{d-2k}{\eta}, \frac{T^\eta}{2\sigma_g^\eta})} r^{-2k} \exp(-\frac{1}{2}(\frac{r}{\sigma_g})^\eta), \tag{12}$$

where $\gamma(\cdot)$ is the lower incomplete gamma function. We see

$$C_g = \frac{\phi_g(r, T)}{\phi_g(r)} = \frac{\Gamma(\frac{d-2k}{\eta})}{\gamma(\frac{d-2k}{\eta}, \frac{T^\eta}{2\sigma_g^\eta})}. \tag{13}$$

Let $\phi_s(r), \phi_s(r, T)$ be PDFs of $\mathcal{S}(\sigma, \eta)$ and $\mathcal{S}_t(\sigma, \eta, T)$ respectively. Given $\mathcal{S}(\sigma, \eta) \propto \exp\left(-\frac{r^\eta}{2\sigma_s^\eta}\right)$, we have

$$\phi_s(r) = \frac{\eta}{2} \frac{1}{(2\sigma_s^\eta)^{\frac{d}{\eta}} \pi^{\frac{d}{2}}} \frac{\Gamma(\frac{d}{2})}{\Gamma(\frac{d}{\eta})} \exp(-\frac{1}{2}(\frac{r}{\sigma_s})^\eta), \tag{14a}$$

$$\phi_s^{-1}(r) = \sigma_s \left( -2 \ln \left( \frac{\Gamma(\frac{d}{\eta})}{\Gamma(\frac{d}{2})} 2^{\frac{d}{\eta}+1} \sigma_s^d \pi^{\frac{d}{2}} \frac{r}{\eta} \right) \right)^{\frac{1}{\eta}}, \tag{14b}$$

$$\phi_s(r, T) = \frac{\eta}{2} \frac{1}{(2\sigma_s^\eta)^{\frac{d}{\eta}} \pi^{\frac{d}{2}}} \frac{\Gamma(\frac{d}{2})}{\gamma(\frac{d}{\eta}, \frac{T^\eta}{2\sigma_s^\eta})} \exp(-\frac{1}{2}(\frac{r}{\sigma_s})^\eta), \tag{14c}$$

and the ratio constant

$$C_s = \frac{\phi_s(r, T)}{\phi_s(r)} = \frac{\Gamma(\frac{d}{\eta})}{\gamma(\frac{d}{\eta}, \frac{T^\eta}{2\sigma_s^\eta})}. \tag{15}$$

### A.2  VISUALIZATION FOR DISTRIBUTIONS

We show the distributions for $r$ in the space by setting

$$y = \phi_g(r) \cdot V_d(r) = \frac{\eta}{2} \frac{1}{(2\sigma_g^\eta)^{\frac{d-2k}{\eta}} \pi^{\frac{d}{2}}} \frac{\Gamma(\frac{d}{2})}{\Gamma(\frac{d-2k}{\eta})} r^{-2k} \exp(-\frac{1}{2}(\frac{r}{\sigma_g})^\eta) \cdot \frac{d\pi^{\frac{d}{2}}}{\Gamma(\frac{d}{2}+1)} r^{d-1} \tag{16}$$

for EGG, and

$$y = \phi_s(r) \cdot V_d(r) = \frac{\eta}{2} \frac{1}{(2\sigma_s^\eta)^{\frac{d}{\eta}} \pi^{\frac{d}{2}}} \frac{\Gamma(\frac{d}{2})}{\Gamma(\frac{d}{\eta})} \exp(-\frac{1}{2}(\frac{r}{\sigma_s})^\eta) \cdot \frac{d\pi^{\frac{d}{2}}}{\Gamma(\frac{d}{2}+1)} r^{d-1} \tag{17}$$

for ESG. See the definition of $V_d(r)$ in Lemma C.1. From Figure 5, we learn that the EGG distributions are relatively evenly distributed in the space compared to ESG, while the phenomenon that most mass of the distribution concentrates to a *thin shell* is general for ESG.

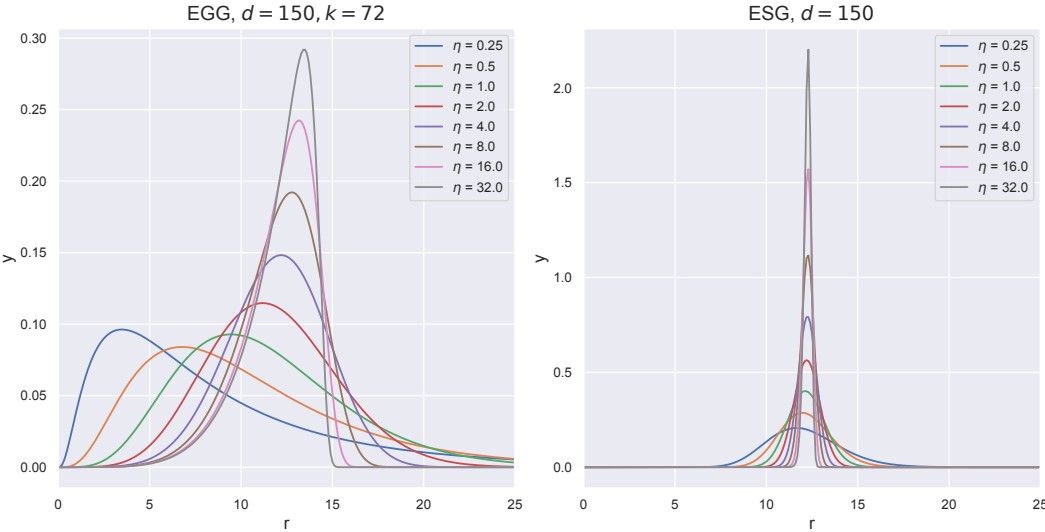

Figure 5: Illustration for distributions defined in Table 1. For each distribution above, we set $\sigma = 1$. $\sigma_g$, $\sigma_s$ are computed respectively according to Table 1, thus we can ensure $\mathbb{E}r^2$ is a constant for all the distributions.

## B    COMPUTATIONAL METHODS FOR EGG AND ESG

We have the solution to Problem (9) for EGG as follows:

**Theorem 2** (Integral form of Problem (9) with EGG). *Let $\mathcal{P} = \mathcal{G}(\sigma, \eta, k)$ with PDF $p(\cdot)$, $\mathcal{Q} = \mathcal{G}_t(\sigma, \eta, k, T)$ with PDF $q(\cdot)$ and $C_g = \frac{\Gamma(\frac{(d-2k)}{\eta})}{\gamma(\frac{d-2k}{\eta}, \frac{T^\eta}{2\sigma_g^\eta})}$, where $\gamma(\cdot, \cdot)$ is the lower incomplete gamma function. Let $\mathcal{V} \triangleq \{z \mid p(z - \delta) + \nu_1 p(z) + \nu_2 q(z) < 0\}$. We write $t$ for $\sigma_g(2u)^{\frac{1}{\eta}}$. Then*

$$
\begin{aligned}
\mathbb{P}_{z \sim \mathcal{P}}\{z \in \mathcal{V}\} = \mathbb{E}_{u \sim \Gamma(\frac{d-2k}{\eta}, 1)} & \begin{cases} \omega_1(u, \nu_1), & u \geq \frac{T^\eta}{2\sigma_g^\eta}, \\[2mm] \omega_1(u, \nu_1 + C_g\nu_2), & u < \frac{T^\eta}{2\sigma_g^\eta}, \end{cases} \\[3mm]
\mathbb{P}_{z \sim \mathcal{Q}}\{z \in \mathcal{V}\} = C_g \mathbb{E}_{u \sim \Gamma(\frac{d-2k}{\eta}, 1)} & \; \omega_1(u, \nu_1 + C_g\nu_2) \cdot \mathbb{1}_{u \leq \frac{T^\eta}{2\sigma_g^\eta}}, \\[3mm]
\mathbb{P}_{z \sim \mathcal{P} + \delta}\{z \in \mathcal{V}\} = & \begin{cases} \mathbb{E}_{u \sim \Gamma(\frac{d-2k}{\eta}, 1)} \; \omega_2(u), & \nu_1 \geq 0, \\[2mm] \mathbb{E}_{u \sim \Gamma(\frac{d-2k}{\eta}, 1)} \{\omega_2(u) + \omega_3(u)\}, & \nu_1 < 0, \end{cases}
\end{aligned} \tag{18}
$$

*where*

$$
\begin{aligned}
\omega_1(u, \nu) &= \Psi_{\frac{d-1}{2}}\left( \frac{(\rho + t)^2 - \left(p^{-1}(-\nu p(t))\right)^2}{4\rho t} \right), \\[3mm]
\omega_2(u) &= \Psi_{\frac{d-1}{2}}\left( \frac{\min\{T^2, \left(p^{-1}(-\frac{1}{\nu_1 + C_g\nu_2} p(t))\right)^2\} - (t - \rho)^2}{4\rho t} \right), \\[3mm]
\omega_3(u) &= \max\left\{ \Psi_{\frac{d-1}{2}}\left( \frac{\left(p^{-1}(-\frac{1}{\nu_1} p(t))\right)^2 - (t - \rho)^2}{4\rho t} \right) - \Psi_{\frac{d-1}{2}}\left( \frac{T^2 - (t - \rho)^2}{4\rho t} \right), 0 \right\}.
\end{aligned} \tag{19}
$$

The proof of Theorem 2 is deferred to Appendix D. When taking ESG as the smoothing distribution and TESG as the additional distribution, we solve Problem (9) by the following theorem, which is slightly more complicated than that of EGG:

**Theorem 3** (Integral form of Problem (9) with ESG). *Let $\mathcal{P} = \mathcal{S}(\sigma, \eta)$ with PDF $p(\cdot)$, $\mathcal{Q} = \mathcal{S}_t(\sigma, \eta, T)$ with PDF $q(\cdot)$ and $C_s = \frac{\Gamma(\frac{d}{\eta})}{\gamma(\frac{d}{\eta}, \frac{T^\eta}{2\sigma_s^\eta})}$, where $\gamma(\cdot, \cdot)$ is the lower incomplete gamma func-*

*tion. Let $\mathcal{V} \triangleq \{z \mid p(z-\delta) + \nu_1 p(z) + \nu_2 q(z) < 0\}$. We write $t$ for $\sigma_s(2u)^{\frac{1}{\eta}}$. Then*

$$
\mathbb{P}_{z\sim\mathcal{P}}\{z \in \mathcal{V}\} = \mathbb{E}_{u\sim\Gamma(\frac{d}{\eta},1)}
\begin{cases}
\omega_1(u, \nu_1), & u \geq \dfrac{T^\eta}{2\sigma_s^\eta}, \\[2mm]
\omega_1(u, \nu_1 + C_s \nu_2), & u < \dfrac{T^\eta}{2\sigma_s^\eta},
\end{cases}
$$

$$
\mathbb{P}_{z\sim\mathcal{Q}}\{z \in \mathcal{V}\} = C_s \mathbb{E}_{u\sim\Gamma(\frac{d}{\eta},1)}\, \omega_1(u, \nu_1 + C_s \nu_2) \cdot \mathbb{1}_{u \leq \frac{T^\eta}{2\sigma_s^\eta}}, \tag{20}
$$

$$
\mathbb{P}_{z\sim\mathcal{P}+\delta}\{z \in \mathcal{V}\} =
\begin{cases}
\mathbb{E}_{u\sim\Gamma(\frac{d}{\eta},1)}\, \omega_2(u), & \nu_1 \geq 0, \\[2mm]
\mathbb{E}_{u\sim\Gamma(\frac{d}{\eta},1)}\{\omega_2(u) + \omega_3(u)\}, & \nu_1 < 0,
\end{cases}
$$

*where*

$$
\omega_1(u,\nu) =
\begin{cases}
\Psi_{\frac{d-1}{2}}\left(\dfrac{(\rho+t)^2 - \left(p^{-1}(-\nu p(t))\right)^2}{4\rho t}\right) & , u - \ln(-\nu) \geq 0, \\[4mm]
1 & , u - \ln(-\nu) < 0,
\end{cases}
$$

$$
\omega_2(u) =
\begin{cases}
\Psi_{\frac{d-1}{2}}\left(\dfrac{\min\{T^2, \left(p^{-1}(-\frac{1}{\nu_1+C_s\nu_2}p(t))\right)^2\} - (t-\rho)^2}{4\rho t}\right) & , u + \ln(-(\nu_1 + C_s\nu_2)) \geq 0, \\[4mm]
0 & , u + \ln(-(\nu_1 + C_s\nu_2)) < 0,
\end{cases} \tag{21}
$$

$$
\omega_3(u) =
\begin{cases}
\max\left\{\Psi_{\frac{d-1}{2}}\left(\dfrac{\left(p^{-1}(-\frac{1}{\nu_1}p(t))\right)^2 - (t-\rho)^2}{4\rho t}\right) - \Psi_{\frac{d-1}{2}}\left(\dfrac{T^2 - (t-\rho)^2}{4\rho t}\right), 0\right\} & , u + \ln(-\nu_1) \geq 0, \\[4mm]
0 & , u + \ln(-\nu_1) < 0.
\end{cases}
$$

See proof in Appendix E. Compared to Theorem 2, we emphasize the following differences: (1) $p(\cdot)$ represents different PDFs in the two theorems; (2) there are more branches for $\omega$ functions in Theorem 3, which originates from different properties of the logarithmic function and the Lambert W function.

## C  PROOF OF THEOREM 1 AND THEOREM 4

We put proofs of Theorem 1 and Theorem 4 together since they share common thinking. Overall, this section includes 3 parts. We first introduce lemmas mainly on concentration properties of beta and gamma distributions, then we derive the solution for a lower bound of Problem (4). Finally, we prove Theorem 1 and Theorem 4 respectively on the basis of those introduced lemmas. Both proofs are essentially generalizations for appendix F.3 in Li et al. (2022).

Here we briefly sketch the proofs of Theorem 1 and Theorem 4. The core thinking for the proofs is to calculate probability under a given perturbation $\delta$ (which appears as a function of $d$ in both the theorems, e.g., $\|\delta\|_2 = \sigma\sqrt{d}$). First of all, suppose that the base classifier satisfies a certain concentration property (Definition 2). Since the solution of the lower bound of Problem (4) is known (by one of the lemmas), we further find the lower bound of the solution by other lemmas just proved until it can be computed easily. At last, if the solution computed is greater than $0.5$, $\delta$ is successfully certified.

### C.1  PRELIMINARIES

We show lemmas for proofs in this section. In a nutshell, Lemma C.1 gives the volume of hypersphere in the $\mathbb{R}^d$ space for later integrals; Lemma C.2 offers the probability that the mass of ESG is within $T$; Lemma C.4 gives a solution for a lower bound of Problem (4); Lemma C.5 and Lemma C.6 reveal concentration properties of beta and gamma distributions, by which we obtain lower bounds of Problem (4); Lemma C.7 proves monotonicity of a function that appears in the proof of Theorem 4.

**Lemma C.1.** *(Equation (16) in Li et al. (2022), Volume of Hypersphere, restated) Let $r \in \mathbb{R}_+, d \in \mathbb{N}_+$. The volume $V_d(r)$ of $d-1$ dimensional hypersphere with radius $r$ is*

$$
V_d(r) = \frac{d\pi^{\frac{d}{2}}}{\Gamma(\frac{d}{2}+1)} r^{d-1}. \tag{22}
$$

For all integrals in this paper, WLOG we let $d$ be an even number.

**Lemma C.2.** *For random variable $z \sim \mathcal{S}(\sigma, \eta)$ and determined threshold $T \in \mathbb{R}_+$,*

$$\mathbb{P}\{\|z\|_2 \leq T\} = \Lambda_{\frac{d}{\eta}}(\frac{T^\eta}{2\sigma_s^\eta}). \tag{23}$$

*Proof.* We have $\phi_s$, the PDF of $z$ from Equation (14a) and $V_d(r)$ from Lemma C.1, then

$$
\begin{aligned}
\mathbb{P}\{\|z\|_2 \leq T\} &= \int_0^T \phi_s(r) \exp(-\frac{r^\eta}{2\sigma_s^\eta}) \frac{d\pi^{\frac{d}{2}}}{\Gamma(\frac{d}{2}+1)} r^{d-1} dr \\
&= \int_0^T \frac{\eta}{2} \frac{1}{(2\sigma_s^\eta)^{\frac{d}{\eta}} \pi^{\frac{d}{2}}} \frac{\Gamma(\frac{d}{2})}{\Gamma(\frac{d}{\eta})} r^{d-1} \exp(-\frac{r^\eta}{2\sigma_s^\eta}) \frac{d\pi^{\frac{d}{2}}}{\Gamma(\frac{d}{2}+1)} dr \\
&= \frac{1}{\Gamma(\frac{d}{\eta})} \int_0^{\frac{T^\eta}{2\sigma_s^\eta}} t^{\frac{d}{\eta}-1} \exp(-r) dt \\
&= \Lambda_{\frac{d}{\eta}}(\frac{T^\eta}{2\sigma_s^\eta}).
\end{aligned} \tag{24}
$$

$\square$

**Lemma C.3.** *Given a base classifier $f : \mathbb{R}^d \to \mathcal{Y}$ satisfies $(\sigma, p, \eta)$-concentration property at input $(x_0, y_0) \in \mathbb{R}^d$. For the additional distribution $\mathcal{Q} = \mathcal{G}_t(\sigma, \eta, k, T)$, where $T = \sigma_s(2\Lambda_{\frac{d}{\eta}}^{-1}(p))^{\frac{1}{\eta}}, \eta \in \mathbb{R}_+$ and $2d - k \in [1, 30] \cap \mathbb{N}$, we have*

$$\mathbb{P}_{z \sim \mathcal{Q}}\{f(x_0 + z) = y_0\} = 1. \tag{25}$$

*Proof.* If $f$ satisfies $(\sigma, p, \eta)$-concentration property, when $p$ is fixed, we have

$$\mathbb{P}_{z \sim S(\sigma_s, \eta)}\{f(x_0 + z) = y_0 \mid \|z\|_2 \leq T\} = 1 \tag{26}$$

for $T = \sigma_s(2\Lambda_{\frac{d}{\eta}}^{-1}(p))^{\frac{1}{\eta}}$ from Definition 2 and Lemma C.2. Notice though Equation (26) is defined by $S(\sigma_s, \eta)$, $f(x_0 + z) = y_0$ holds for all $\|z\|_2 \leq T$ since distribution $\mathcal{S}$ has positive density everywhere. Consider the case $z \sim \mathcal{G}_t(\sigma, \eta, k, T)$, we thereby have

$$\mathbb{P}_{z \sim \mathcal{G}_t(\sigma, \eta, k, T)}\{f(x_0 + z) \neq y_0 \mid 0 < \|z\|_2 \leq T\} = 0. \tag{27}$$

Notice $z \neq 0$ since there is a $z^{-2k}$ term in EGG's PDF. By Equation (11a), we have

$$
\begin{aligned}
&\mathbb{P}_{z \sim \mathcal{G}_t(\sigma, \eta, k, T)}\{z = \mathbf{0}\} \\
&\mathbb{P}_{z \sim \mathcal{G}(\sigma, \eta, k)}\{z = \mathbf{0} \mid \|z\|_2 \leq T\} \\
&\leq \lim_{t \to 0} \mathbb{P}_{z \sim \mathcal{G}(\sigma, \eta, k)}\{\|z\|_2 \leq t\} \\
&= \lim_{t \to 0} \int_0^t \frac{\eta}{2} \frac{1}{(2\sigma_g^\eta)^{\frac{d-2k}{\eta}} \pi^{\frac{d}{2}}} \frac{\Gamma(\frac{d}{2})}{\Gamma(\frac{d-2k}{\eta})} r^{-2k} \exp(-\frac{1}{2}(\frac{r}{\sigma_g})^\eta) \frac{d\pi^{\frac{d}{2}}}{\Gamma(\frac{d}{2}+1)} r^{d-1} dr \\
&= \frac{\eta}{2} \frac{1}{(2\sigma_g^\eta)^{\frac{d-2k}{\eta}} \pi^{\frac{d}{2}}} \frac{\Gamma(\frac{d}{2})}{\Gamma(\frac{d-2k}{\eta})} \frac{d\pi^{\frac{d}{2}}}{\Gamma(\frac{d}{2}+1)} \lim_{t \to 0} \int_0^t \exp(-\frac{1}{2}(\frac{r}{\sigma_g})^\eta) r^{d-2k-1} dr \\
&= 0.
\end{aligned} \tag{28}
$$

Therefore, we see

$$\mathbb{P}_{z \sim \mathcal{G}_t(\sigma, \eta, k, T)}\{f(x_0 + z) = y_0\} = \mathbb{P}_{z \sim \mathcal{G}(\sigma, \eta, k)}\{f(x_0 + z) = y_0 \mid 0 \leq \|z\|_2 \leq T\} = 1, \tag{29}$$

which concludes the proof. $\square$

Here we give a solution for a lower bound of Problem (4). The lower bound found with the help of Lemma C.3 can be solved by the level set method (Yang et al., 2020), sharing the same core technique with proofs of Theorem 2 and 3. We refer the readers to Appendix D and previous papers (Yang et al., 2020; Li et al., 2022) for more details.

**Lemma C.4.** *Under the setting of Lemma C.3, we let*

$$R = \max \quad \rho,$$

$$\text{s.t.} \quad \mathbb{E}_{u \sim \Gamma(\frac{d-2k}{\eta}, 1)} \Psi_{\frac{d-1}{2}} \left( \frac{T^2 - (\sigma_g(2u)^{\frac{1}{\eta}} - \rho)^2}{4\rho\sigma_g(2u)^{\frac{1}{\eta}}} \right) \geq \frac{1}{2}. \tag{30}$$

*If $R_D$ is the tightest $\ell_2$ certified radius when $\mathcal{P} = \mathcal{G}(\sigma, \eta, k)$ is the smoothing distribution by DSRS, then $R_D \geq R$.*

*Proof.* According to Lemma C.3, we define $\mathcal{Q} = \mathcal{G}_t(\sigma, \eta, k, T)$ as the additional distribution for $\mathcal{P}$. Then the Problem (4) can be simplified as

$$\min_{\tilde{f}_{x_0} \in \mathcal{F}} \quad \mathbb{E}_{z \sim \mathcal{P}} \left( \tilde{f}_{x_0}(\delta + z) \right),$$

$$\text{s.t.} \quad \mathbb{E}_{z \sim \mathcal{P}} \left( \tilde{f}_{x_0}(z) \right) = A, \; \mathbb{E}_{z \sim \mathcal{Q}} \left( \tilde{f}_{x_0}(z) \right) = B$$

$$\overset{(a)}{=} \min_{\tilde{f}_{x_0} \in \mathcal{F}} \quad \mathbb{E}_{z \sim \mathcal{P}} \left( \tilde{f}_{x_0}(\delta + z) \right),$$

$$\text{s.t.} \quad \mathbb{E}_{z \sim \mathcal{P}} \left( \tilde{f}_{x_0}(z) \right) = A, \; \mathbb{E}_{z \sim \mathcal{Q}} \left( \tilde{f}_{x_0}(z) \right) = 1 \tag{31}$$

$$\overset{(b)}{\geq} \min_{\tilde{f}_{x_0} \in \mathcal{F}} \quad \mathbb{E}_{z \sim \mathcal{P}} \left( \tilde{f}_{x_0}(\delta + z) \right),$$

$$\text{s.t.} \quad \mathbb{E}_{z \sim \mathcal{Q}} \left( \tilde{f}_{x_0}(z) \right) = 1,$$

where (a) is by Lemma C.3, and (b) is because subjection $\mathbb{E}_{\epsilon \sim \mathcal{P}}[f(\epsilon)] = P_A$ offers extra information outside $\{x \mid \|x - x_0\|_2 \leq T\}$ in $\mathbb{R}^n$, where $p(x) \geq 0$. It is obvious that the equality holds if $A = 0$. In other words, under the special case where $B = 1$, the simplified problem offers a lower bound for the primal problem. For the simplified problem, we have the worst function $\tilde{f}_{x_0}^*(t)$:

$$\tilde{f}_{x_0}^*(t) = \begin{cases} 1, \; \|t\|_2 \leq T, \\ 0, \; \|t\|_2 > T. \end{cases} \tag{32}$$

Under the worst classifier, the lower bound problem for Problem (4) can be finally written as

$$\mathbb{E}_{z \sim \mathcal{P}} \left( \tilde{f}_{x_0}^*(\delta + z) \right) \tag{33}$$

for a fixed $\delta$. Similar to Theorem 2 and Theorem 3, the final simplified problem is solved by the level set method (Yang et al., 2020). We have

$$\mathbb{E}_{z \sim \mathcal{P}} \left( \tilde{f}_{x_0}^*(\delta + z) \right)$$

$$= \mathbb{E}_{\epsilon \sim \mathcal{P}} \{ \|\delta + x\|_2 \leq T \}$$

$$= \int_{\mathbb{R}^d} p(x) \cdot \mathbb{1}_{\|x+\delta\|_2 \leq T} dx$$

$$= \int_0^\infty \phi_g(r) \frac{d\pi^{\frac{d}{2}}}{\Gamma(\frac{d}{2}+1)} r^{d-1} dr \mathbb{P}\{\|x + \delta\|_2 \leq T \mid \|x\|_2 = r\}$$

$$= \int_0^\infty \frac{\eta}{2} \frac{1}{(2\sigma_g^\eta)^{\frac{d-2k}{\eta}} \pi^{\frac{d}{2}}} \frac{\Gamma(\frac{d}{2})}{\Gamma(\frac{d-2k}{\eta})} r^{d-2k-1} \exp[-\frac{1}{2}(\frac{r}{\sigma_g})^\eta] \frac{d\pi^{\frac{d}{2}}}{\Gamma(\frac{d}{2}+1)} dr \mathbb{P}\{\|x + \delta\|_2 \leq T \mid \|x\|_2 = r\}$$

$$= \frac{1}{\Gamma(\frac{d-2k}{\eta})} \int_0^\infty u^{\frac{d-2k}{\eta}-1} \exp(-u) du \mathbb{P}\{\|x + \delta\|_2 \leq T \mid \|x\|_2 = \sigma_g(2u)^{\frac{1}{\eta}}\}. \tag{34}$$

As $\|x + \delta\|_2 \leq T \iff x_1 \leq \frac{T^2 - \rho^2 - \sigma_g^2(2u)^{\frac{2}{\eta}}}{2\rho}$, we have

$$\frac{1 + \frac{x_1}{\sigma_g(2u)^{\frac{1}{\eta}}}}{2} \sim \text{Beta}(\frac{d-1}{2}, \frac{d-1}{2}) \tag{35}$$

by Lemma D.1. Thus, we get

$$\mathbb{P}\{\|x + \delta\|_2 \le T \mid \|x\|_2 = \sigma_g(2u)^{\frac{1}{\eta}}\} = \Psi_{\frac{d-1}{2}}\left(\frac{T^2 - (\sigma_g(2u)^{\frac{1}{\eta}} - \rho)^2}{4\rho\sigma_g(2u)^{\frac{1}{\eta}}}\right). \tag{36}$$

Combining Equation (34) and Equation (36), we finally get

$$\mathbb{E}_{z\sim\mathcal{P}}\left(\tilde{f}_{x_0}^*(\delta + z)\right) = \mathbb{E}_{u\sim\Gamma(\frac{d-2k}{\eta},1)}\Psi_{\frac{d-1}{2}}\left(\frac{T^2 - (\sigma_g(2u)^{\frac{1}{\eta}} - \rho)^2}{4\rho\sigma_g(2u)^{\frac{1}{\eta}}}\right). \tag{37}$$

If we can ensure that $\mathbb{E}_{z\sim\mathcal{P}}\left(\tilde{f}_{x_0}^*(\delta + z)\right) \ge 0.5$ for $\delta = (\rho, 0, 0, \cdots, 0)^T$, $\|\delta\|_2$ will be qualified as a $\ell_2$ certified radius due to $\ell_2$-symmetry in the $R^n$ space (Zhang et al., 2020). As $R$ is the solution to the lower-bound problem of Problem (4), we have $R_D \ge R$, which concludes the proof. □

The next two lemmas describe the concentration of the gamma distribution and the beta distribution, both are required for the lower estimation of Problem (4).

**Lemma C.5.** *(Concentration of the Beta Distribution) Let $\tau \in (\frac{1}{2}, 1)$, $\theta \in (0, 1)$, there exist $d_0 \in \mathbb{N}_+$, for any $d \ge d_0$,*

$$\Psi_{\frac{d-1}{2}}(\tau) \ge \theta. \tag{38}$$

*Proof.* Let X $\sim \text{Beta}(\frac{d-1}{2}, \frac{d-1}{2})$. By property of the beta distribution, we have $\mathbb{E}X = \frac{1}{2}$, $\mathbb{D}X = \frac{1}{4d}$, and

$$\begin{aligned}
\mathbb{P}\{X > \tau\} &= \mathbb{P}\{X - \frac{1}{2} > \tau - \frac{1}{2}\} \\
&\le \mathbb{P}\{|X - \frac{1}{2}| \ge \tau - \frac{1}{2}\} \\
&\le \frac{1}{4d(\tau - \frac{1}{2})^2}.
\end{aligned} \tag{39}$$

We then have

$$\Psi_{\frac{d-1}{2}}(\tau) = 1 - \mathbb{P}\{X \ge \tau\} \ge 1 - \frac{1}{4d(\tau - \frac{1}{2})^2}. \tag{40}$$

Let $1 - \frac{1}{4d(\tau - \frac{1}{2})^2} \ge \theta$, then $d \ge \frac{1}{4(1-\theta)(\tau - \frac{1}{2})^2}$. Picking $d_0 = \lceil \frac{1}{4(1-\theta)(\tau - \frac{1}{2})^2} \rceil$ concludes the proof. □

**Lemma C.6.** *(Concentration of the Gamma Distribution) Let $p \in (0, 1)$, $d \in \mathbb{N}_+$, $\eta \in \mathbb{R}_+$ and $\beta \in (0, 1)$, there exist $d_0 \in \mathbb{N}_+$, for any $d \ge d_0$,*

$$\Lambda_{\frac{d}{\eta}}\left(\frac{\beta d}{\eta}\right) \le p. \tag{41}$$

*Proof.* Let $X \sim \Gamma(\frac{d}{\eta}, 1)$, then $\mathbb{E}X = \frac{d}{\eta}$, $\mathbb{D}X = \frac{d}{\eta}$,

$$\begin{aligned}
\Lambda_{\frac{d}{\eta}}\left(\frac{\beta d}{\eta}\right) &= \mathbb{P}\left\{X \le \frac{\beta d}{\eta}\right\} \\
&= \mathbb{P}\left\{X - \frac{d}{\eta} \le -\frac{(1-\beta)d}{\eta}\right\} \\
&\le \mathbb{P}\left\{|X - \frac{d}{\eta}| \ge \frac{(1-\beta)d}{\eta}\right\} \\
&\le \frac{\eta}{(1-\beta)^2 d}.
\end{aligned} \tag{42}$$

If $\frac{\eta}{(1-\beta)^2 d} \le p$, then $d \ge \frac{\eta}{(1-\beta)^2 p}$. Let $d_0 = \lceil \frac{\eta}{(1-\beta)^2 p} \rceil$, for any $d \ge d_0$, we have $\frac{\eta}{(1-\beta)^2 d} \le \frac{\eta}{(1-\beta)^2 d_0} \le p$, which concludes the proof. □

**Remark.** Both Lemma C.5 and Lemma C.6 illustrate that random variables following beta and gamma distributions are highly concentrated towards their respective expectations under the high-dimensional setting.

The next lemma is required in the proof of Theorem 4.

**Lemma C.7.** *Let* $x \in \mathbb{N}_+$, $\eta = \frac{1}{n}, n \in \mathbb{N}_+$, $g(x) = \dfrac{x}{\left( \Pi_{i=1}^{\frac{2}{\eta}}(\frac{x+2}{\eta} - i) \right)^{\frac{\eta}{2}}}$, *then* $g(x)$ *is a non-decreasing function with respect to* $x$.

*Proof.* Obviously $g(x) > 0$. Let $h(x) = \ln g(x)$, then

$$\frac{\mathrm{d}h(x)}{\mathrm{d}x} \geq 0 \iff \frac{1}{g(x)}\frac{\mathrm{d}g(x)}{\mathrm{d}x} \geq 0 \iff \frac{\mathrm{d}g(x)}{\mathrm{d}x} \geq 0, \tag{43}$$

$$
\begin{aligned}
\frac{\mathrm{d}h(x)}{\mathrm{d}x} &= \frac{\mathrm{d}}{\mathrm{d}x}\left( \ln x - \frac{\eta}{2}\sum_{i=1}^{\frac{2}{\eta}} \ln(\frac{x+2}{\eta} - i) \right) \\
&= \frac{1}{x} - \frac{\eta}{2}\sum_{i=1}^{\frac{2}{\eta}} \frac{1}{x+2-i\eta} \\
&\geq \frac{1}{x} - \frac{\eta}{2}\sum_{i=1}^{\frac{2}{\eta}} \frac{1}{x} \\
&= 0.
\end{aligned}
\tag{44}
$$

Therefore, for all $x \in \mathbb{N}_+$, we have $\frac{\mathrm{d}g(x)}{\mathrm{d}x} \geq 0$, which concludes the proof. □

### C.2 PROOF OF THEOREM 1

*Proof.* Theorem 1 is for Case I in section 4.2, where we derive the lower bound of the $\ell_2$ certified radius for the smoothing distribution $\mathcal{P} = \mathcal{G}(\sigma, \eta, k)$ and the additional distribution $\mathcal{Q} = \mathcal{G}_t(\sigma, \eta, k, T)$ by fixing the ideal base classifier. We suppose the base classifier satisfies $(\sigma, p, 2)$-concentration property, which implies the base classifier is an ideal classifier for restricted Gaussian noise. For the convenience of future discussions, we parameterize 0.02 as $\mu$. In this subsection, the worst classifier $\tilde{f}_{x_0}^*$ is defined the same as that in Lemma C.3.

We see the condition $\eta = 2$ simplifies some lemmas above. Let $\eta = 2$ in Lemma C.2, we get $\mathbb{P}_{z\sim\mathcal{S}(\sigma,2)}\{\|z\|_2 \leq T\} = \Lambda_{\frac{d}{2}}(\frac{T^2}{2\sigma_s^2})$, which means

$$T = \sigma\sqrt{2\Lambda_{\frac{d}{2}}^{-1}(p)}. \tag{45}$$

By definition of $(\sigma, p, 2)$-concentration and Lemma C.3, we have $\mathbb{P}_{z\sim\mathcal{G}_t(\sigma,\eta,k,T)}\{f(x_0+z) = y_0\} = 1$, thus we can find a lower bound to estimate Problem (4) by Lemma C.4. Let $\eta = 2$ in Lemma C.5, we get

$$\Lambda_{\frac{d}{2}}(\frac{\beta d}{2}) \leq p. \tag{46}$$

We have thereby

$$T \geq \sigma\sqrt{\beta d} \tag{47}$$

by combining Equation (45) and Equation (46). Equation (47) will be used for finding the lower bound of Problem (33). We then consider the solution of the lower-bound Problem (33) in Lemma C.4. We have

$$\frac{T^2 - (\sigma_g(2u)^{\frac{1}{\eta}} - \rho)^2}{4\rho\sigma_g(2u)^{\frac{1}{\eta}}} \geq \tau \iff \sigma_g(2u)^{\frac{2}{\eta}} - (2-4\tau)\rho\sigma_g(2u)^{\frac{1}{\eta}} + \rho^2 - T^2 \leq 0. \tag{48}$$

Notice Equation (48) is a one-variable quadratic inequality with respect to $\sigma_g(2u)^{\frac{1}{\eta}}$. Let $\rho = \mu\sigma\sqrt{d}$ where the constant $\mu \in \mathbb{R}_+$. When the discriminant $\Delta$ for Equation (48) is positive, the solution for it is

$$\sigma_g(2u)^{\frac{1}{\eta}} \in \left[0, (1-2\tau)\rho + \sqrt{T^2 + (4\tau^2 - 4\tau)\rho^2}\right)$$
$$\Longleftrightarrow 0 \le u < \frac{1}{2}\left(\frac{(1-2\tau)\rho + \sqrt{T^2 + (4\tau^2 - 4\tau)\rho^2}}{\sigma_g}\right)^{\eta}. \tag{49}$$

Now we are ready to show the minimization for $\mathbb{E}_{z\sim\mathcal{P}}\left(\tilde{f}_{x_0}^*(\delta + z)\right)$ for base classifier satisfies $(\sigma, p, 2)$-concentration property. We have

$$\mathbb{E}_{z\sim\mathcal{P}}\left(\tilde{f}_{x_0}^*(\delta + z)\right)$$
$$= \mathbb{E}_{u\sim\Gamma(\frac{d-2k}{\eta},1)}\Psi_{\frac{d-1}{2}}\left(\frac{T^2 - (\sigma_g(2u)^{\frac{1}{\eta}} - \rho)^2}{4\rho\sigma_g(2u)^{\frac{1}{\eta}}}\right)$$
$$\ge \theta\mathbb{E}_{u\sim\Gamma(\frac{d-2k}{\eta},1)}\mathbb{I}\left(\frac{T^2 - (\sigma_g(2u)^{\frac{1}{\eta}} - \rho)^2}{4\rho\sigma_g(2u)^{\frac{1}{\eta}}} \ge \tau\right)$$
$$\ge \theta\mathbb{E}_{u\sim\Gamma(\frac{d-2k}{\eta},1)}\mathbb{I}\left(u < \frac{1}{2}\left(\frac{(1-2\tau)\rho + \sqrt{T^2 + (4\tau^2 - 4\tau)\rho^2}}{\sigma_g}\right)^{\eta}\right) \tag{50}$$
$$\overset{(a)}{\ge} \theta\mathbb{E}_{u\sim\Gamma(\frac{d-2k}{\eta},1)}\mathbb{I}\left(u < \frac{1}{2}\left(\frac{(1-2\tau)\mu\sigma\sqrt{d} + \sqrt{\sigma^2\beta d + (4\tau^2 - 4\tau)\mu^2\sigma^2 d}}{\sigma_g}\right)^{\eta}\right)$$
$$\overset{(b)}{=} \theta\mathbb{E}_{u\sim\Gamma(\frac{d-2k}{\eta},1)}\mathbb{I}\left(u < \left(\sqrt{\frac{\Gamma(\frac{d-2k+2}{\eta})}{\Gamma(\frac{d-2k}{\eta})}}\left((1-2\tau)\mu + \sqrt{\beta + (4\tau^2 - 4\tau)\mu^2}\right)\right)^{\eta}\right),$$

where $\mathbb{I}(\cdot)$ is the indicator function, (a) is by $\rho = \mu\sigma\sqrt{d}$ and Equation (47), and (b) is because

$$\sigma_g = 2^{-\frac{1}{\eta}}\sqrt{\frac{d\Gamma(\frac{d-2k}{\eta})}{\Gamma(\frac{d-2k+2}{\eta})}}\sigma \tag{51}$$

by definition. We write

$$m = \left(\sqrt{\frac{\Gamma(\frac{d-2k+2}{\eta})}{\Gamma(\frac{d-2k}{\eta})}}\left((1-2\tau)\mu + \sqrt{\beta + (4\tau^2 - 4\tau)\mu^2}\right)\right)^{\eta}, \tag{52}$$

to get

$$\mathbb{E}_{z\sim\mathcal{P}}\left(\tilde{f}_{x_0}^*(\delta + z)\right) \ge \theta\mathbb{E}_{u\sim\Gamma(\frac{d-2k}{\eta},1)}\mathbb{1}_{u<m}$$
$$= \theta\mathbb{P}_{u\sim\Gamma(\frac{d-2k}{\eta},1)}\{u < m\} \tag{53}$$
$$= \theta\Lambda_{\frac{d-2k}{\eta}}(m).$$

Let $\theta = 0.999, \beta = 0.99, \tau = 0.6, \mu = 0.02$ (Li et al., 2022). We show the value of $\Lambda_{\frac{d-2k}{\eta}}(m)$ when $d - 2k \in [1, 30] \cap \mathbb{Z}$ and $\eta = \frac{1}{n}, n \in [1, 50] \cap \mathbb{Z}$ in Table 4. Observing that there's no value greater than $\frac{1}{2\theta}$, we have $\mathbb{E}_{z\sim\mathcal{P}}\left(\tilde{f}_{x_0}^*(\delta + z)\right) \ge \frac{1}{2}$, which concludes the proof. $\qquad\square$

**Computational method for the tight factor $\mu$.** Table 4 reports the computed probability when $\mu = 0.02$. Considering $\mu$ is a constant factor for certified radius, there exists the monotonicity that the larger the $\mu$, the smaller the probability in the table. For this, we offer Algorithm 1 to find the tight $\mu$ for EGG distributions. Given the existence of intractable gamma distribution, we only consider the discrete cases. Limited by length, we do not exhaust all $d - 2k \in [1, 30] \cap \mathbb{N}$. As shown in Table,

---

**Algorithm 1:** Algorithm for finding tight $\mu$ for the $\Omega(\sqrt{d})$ lower bound

---

**Input:** input dimension $d$, hyperparameters $k, \beta, \tau, \theta$, exponent $\eta$, error limitation $e$

1  $\mu_l \leftarrow 0, \mu_r \leftarrow 1$
2  **while** $\mu_r - \mu_l > e$ **do**
3      $\mu_m \leftarrow (\mu_r + \mu_l)/2$
4      $m_m \leftarrow \theta \Lambda_{\frac{d-2k}{\eta}} \left( \left( \left( \sqrt{\frac{\Gamma(\frac{d-2k+2}{\eta})}{\Gamma(\frac{d-2k}{\eta})}} \left( (1-2\tau)\mu_m + \sqrt{\beta + (4\tau^2 - 4\tau)\mu_m^2} \right) \right)^{\eta} \right)$
5                                 ▷ Compute $m_m$ by Equations (52) and (53)
6      **if** $m_m > 1/2\theta$ **then**
7        $\mu_r \leftarrow \mu_m$
8      **else**
9        $\mu_l \leftarrow \mu_m$
10     **end if**
11 **end while**

**Output:** tight constant $\mu_l$ for specified EGG

---

**Remark.** Table 4 actually demonstrates EGG with discrete $\eta \in (0, 2)$ offers a better lower bound in the sense of constant factor than General Gaussian ($\eta = 2$ in EGG), because for every single $d - 2k$ (except 1), $\Lambda_{\frac{d-2k}{\eta}}(m)$ of smaller $\eta$ are always greater than that of the larger. This indicates when we set larger $\mu$ (corresponds to larger certified radius), $\Lambda_{\frac{d-2k}{\eta}}(m)$ of smaller $\eta$ approaches $\frac{1}{2\theta}$ slower. i.e., smaller $\eta$ tolerates larger $\mu$, which is exactly the coefficient of $\sqrt{d}$.

We exhaust the cases since the analytic solution to Problem (33) includes intractable gamma function terms. The proof above can easily be generalized to other $\eta \in \mathbb{R}_+$. e.g., we have tried the sequence of $\eta \in [0.02, 1]$ increasing by 0.001, no value greater than $\frac{1}{2\theta}$ is observed, meaning these $\eta$s are all qualified to provide $\Omega(\sqrt{d})$ lower bounds for the $\ell_2$ certified radius under the setting of Theorem 1. We have also show results for $\eta \in [2, 10] \cap \mathbb{N}$ in Table 4, where the boundary value for $\frac{1}{2\theta}$ is marked red. It is remarkable that $\Lambda_{\frac{d-2k}{\eta}}(m)$ decreases significantly when $\eta > 2$, which is in line with Figure 1a though here we are only considering the lower bound for the certified radius.

One reason for the decreasing effect w.r.t. $\eta$ is that the $(\sigma, p, 2)$-concentration assumption is to some degree more strict for smaller $\eta$ than for larger ones. More specifically, when $\eta$ gets closer to 0, the major mass of EGG gets closer to 0 as well (Figure 5), even if we have set $\mathbb{E}r^2$ to a constant for EGG distributions. As the $(\sigma, p, 2)$-concentration assumption essentially keeps $T$ a constant, more mass is contained by $T$ for smaller $\eta$. In other words, $p = 0.5$ is just for Gaussian distribution, the proportion of the mass of EGG contained within $T$ can be quite large when $T$ is fixed and $\eta$ is small.

### C.3   PROOF OF THEOREM 4

The following theorem introduces $d^{1/\eta}$ into the lower bound using $(\sigma, p, \eta)$-concentration assumption, which can be proved almost equivalent to Theorem 1.

**Theorem 4.** *Let $d \in \mathbb{N}_+$ be a sufficiently large input dimension, $(x_0, y_0) \in \mathbb{R}^d \times \mathcal{Y}$ be a labeled example and $f : \mathbb{R}^d \to \mathcal{Y}$ be a base classifier which satisfies $(\sigma, p, \eta)$-concentration property w.r.t. $(x_0, y_0)$. For the DSRS method, let $\mathcal{P} = \mathcal{G}(\sigma, \eta, k)$ be the smoothing distribution to give a smoothed classifier $\bar{f}_{\mathcal{P}}$, and $\mathcal{Q} = \mathcal{G}_t(\sigma, \eta, k, T)$ be the additional distribution with $T = \sigma_s \sqrt{2\Lambda_{\frac{d}{\eta}}^{-1}(p)}, d - 2k \in [1, 30] \cap \mathbb{N}$ and $\eta \in \{1, \frac{1}{2}, \frac{1}{3}, \cdots, \frac{1}{50}\}$. Then for smoothed classifier $\bar{f}_{\mathcal{D}}(x)$ the certified $\ell_2$ radius*

$$r_\eta \geq 0.02\sigma_s d^{\frac{1}{\eta}}, \tag{54}$$

*where $\sigma_s$ is the substitution variance of $\mathcal{S}(\sigma, \eta)$. When $\sigma_s$ is converted to $\sigma$ keeping $\mathbb{E}r^2$ a constant, we still have*

$$r_{DSRS} = \Omega(\sqrt{d}). \tag{55}$$

The proof shares the same core technique with that of Theorem 1. Since there is formally an $\Omega(d^{\frac{1}{\eta}})$ bound in the result, we further analyze it under a common setting in the community, that converting

$\sigma_s$ to $\sigma$ under a constant $\mathbb{E}r^2$. By definition, we still get an $\Omega(\sqrt{d})$ lower bound for $r_{DSRS}$. Notice the technique we use to prove Theorem 1 and Theorem 4 can be extended to the smaller $\eta$ (say, $\frac{1}{51}$).

*Proof.* In this section, we first prove that a base classifier satisfies a certain concentration property can certify $\Theta(d^{\frac{1}{\eta}})$ $\ell_2$ radii given the smoothing distribution $\mathcal{P} = \mathcal{G}(\sigma, \eta, k)$ and the additional distribution $\mathcal{Q} = \mathcal{G}_t(\sigma, \eta, k, T)$. Then by converting $\sigma_s$ to $\sigma$, we derive a $\Theta(\sqrt{d})$ lower bound for the certified radius. Like Appendix C.2, we find the lower bound for Problem (4) by Lemma C.4. Notice though the solution for the lower bound of Problem (4) looks the same for both the proofs, the base classifiers satisfy different concentration properties, which makes different $T$ and different additional distribution $\mathcal{Q}$ in each case.

In Lemma C.2, when $\eta$ is an arbitrarily determined real number, we have

$$p = \Lambda_{\frac{d}{\eta}}\left(\frac{T^\eta}{2\sigma_s^\eta}\right) \iff T = \sigma_s(2\Lambda_{\frac{d}{\eta}}^{-1}(p))^{\frac{1}{\eta}}. \tag{56}$$

We thereby obtain

$$T \geq (\frac{2\beta}{\eta})^{\frac{1}{\eta}}\sigma_s d^{\frac{1}{\eta}} \tag{57}$$

by Lemma C.6. Then we have Equation (48) and Equation (49) the same as in Appendix C.2. Let $\rho = \zeta\sigma_s d^{\frac{1}{\eta}}$ where $\zeta \in \mathbb{R}_+$. Now we have finished the preparation for the lower estimation of Problem (33). Let $\eta = \frac{1}{n}$, $n \in \mathbb{N}_+, \forall d \geq \tilde{d}$, where $\tilde{d}$ is a sufficiently large real integer which satisfies Lemma C.5 and Lemma C.6. We then have the estimation

$$\mathbb{E}_{z\sim\mathcal{P}}\left(\tilde{f}_{x_0}^*(\delta + z)\right)$$

$$= \mathbb{E}_{u\sim\Gamma(\frac{d-2k}{\eta},1)}\Psi_{\frac{d-1}{2}}\left(\frac{T^2 - (\sigma_g(2u)^{\frac{1}{\eta}} - \rho)^2}{4\rho\sigma_g(2u)^{\frac{1}{\eta}}}\right)$$

$$\overset{(a)}{\geq} \theta\mathbb{E}_{u\sim\Gamma(\frac{d-2k}{\eta},1)}\mathbb{I}\left(\frac{T^2 - (\sigma_g(2u)^{\frac{1}{\eta}} - \rho)^2}{4\rho\sigma_g(2u)^{\frac{1}{\eta}}} \geq \tau\right)$$

$$\overset{(b)}{\geq} \theta\mathbb{E}_{u\sim\Gamma(\frac{d-2k}{\eta},1)}\mathbb{I}\left(u < \frac{1}{2}\left(\frac{(1-2\tau)\rho + \sqrt{T^2 + (4\tau^2 - 4\tau)\rho^2}}{\sigma_g}\right)^\eta\right)$$

$$\overset{(c)}{=} \theta\mathbb{E}_{u\sim\Gamma(\frac{d-2k}{\eta},1)}\mathbb{I}\left(u < \frac{1}{2}\left(\sqrt{\frac{\Gamma(\frac{d}{\eta})\Gamma(\frac{d-2k+2}{\eta})}{\Gamma(\frac{d+2}{\eta})\Gamma(\frac{d-2k}{\eta})}}\frac{(1-2\tau)\rho + \sqrt{T^2 + (4\tau^2 - 4\tau)\rho^2}}{\sigma_s}\right)^\eta\right)$$

$$\overset{(d)}{\geq} \theta\mathbb{E}_{u\sim\Gamma(\frac{d-2k}{\eta},1)}\mathbb{I}\left(u < \frac{1}{2}\left(\sqrt{\frac{\Gamma(\frac{d}{\eta})\Gamma(\frac{d-2k+2}{\eta})}{\Gamma(\frac{d+2}{\eta})\Gamma(\frac{d-2k}{\eta})}}\frac{(1-2\tau)\zeta\sigma_s d^{\frac{1}{\eta}} + \sqrt{(\frac{2\beta}{\eta})^{\frac{2}{\eta}}\sigma_s^2 d^{\frac{2}{\eta}} + (4\tau^2 - 4\tau)(\zeta\sigma_s d^{\frac{1}{\eta}})^2}}{\sigma_s}\right)^\eta\right)$$

$$= \theta\mathbb{E}_{u\sim\Gamma(\frac{d-2k}{\eta},1)}\mathbb{I}\left(u < \frac{d}{2}\left(\sqrt{\frac{\Gamma(\frac{d}{\eta})\Gamma(\frac{d-2k+2}{\eta})}{\Gamma(\frac{d+2}{\eta})\Gamma(\frac{d-2k}{\eta})}}\left((1-2\tau)\zeta + \sqrt{(\frac{2\beta}{\eta})^{\frac{2}{\eta}} + (4\tau^2 - 4\tau)\zeta^2}\right)\right)^\eta\right)$$

$$= \theta\mathbb{E}_{u\sim\Gamma(\frac{d-2k}{\eta},1)}\mathbb{I}\left(u < \frac{d}{2\left(\prod_{i=1}^{\frac{2}{\eta}}(\frac{d+2}{\eta} - i)\right)^{\frac{\eta}{2}}}\left(\sqrt{\frac{\Gamma(\frac{d-2k+2}{\eta})}{\Gamma(\frac{d-2k}{\eta})}}\left((1-2\tau)\zeta + \sqrt{(\frac{2\beta}{\eta})^{\frac{2}{\eta}} + (4\tau^2 - 4\tau)\zeta^2}\right)\right)^\eta\right)$$

$$\overset{(e)}{\geq} \theta\mathbb{E}_{u\sim\Gamma(\frac{d-2k}{\eta},1)}\mathbb{I}\left(u < \frac{\tilde{d}}{2\left(\prod_{i=1}^{\frac{2}{\eta}}(\frac{\tilde{d}+2}{\eta} - i)\right)^{\frac{\eta}{2}}}\left(\sqrt{\frac{\Gamma(\frac{d-2k+2}{\eta})}{\Gamma(\frac{d-2k}{\eta})}}\left((1-2\tau)\zeta + \sqrt{(\frac{2\beta}{\eta})^{\frac{2}{\eta}} + (4\tau^2 - 4\tau)\zeta^2}\right)\right)^\eta\right).$$

$$\tag{58}$$

In the equations above: (a) by Lemma C.5; (b) solve the inequality with respect to $u$ in the indicator function, whose solution is shown in Equation (49); (c) for a constant $\mathbb{E}r^2$, we have

$$\sigma_g = \sqrt{\frac{\Gamma(\frac{d-2k}{\eta})\Gamma(\frac{d+2}{\eta})}{\Gamma(\frac{d}{\eta})\Gamma(\frac{d-2k+2}{\eta})}}\sigma_s; \tag{59}$$

(d) by Lemma C.6 and $\rho = \zeta\sigma_s d^{\frac{1}{\eta}}$; (e) by Lemma C.7.

We write

$$m = \frac{\tilde{d}}{2\left(\prod_{i=1}^{\frac{2}{\eta}}(\frac{\tilde{d}+2}{\eta} - i)\right)^{\frac{\eta}{2}}} \left(\sqrt{\frac{\Gamma(\frac{d-2k+2}{\eta})}{\Gamma(\frac{d-2k}{\eta})}}\left((1-2\tau)\zeta + \sqrt{(\frac{2\beta}{\eta})^{\frac{2}{\eta}} + (4\tau^2 - 4\tau)\zeta^2}\right)\right)^{\eta},$$

(60)

and then we have

$$\mathbb{E}_{z\sim\mathcal{P}}\left(\tilde{f}_{x_0}^*(\delta + z)\right) \geq \Lambda_{\frac{d-2k}{\eta}}(m)$$

(61)

by Equation (58). Notice the lower estimation is slightly different to Appendix C.2 since there is a $\tilde{d}$ in the expression. Pick $p = 0.5$, $\tilde{d} = 25000, \theta = 0.999, \beta = 0.99, \tau = 0.6$, $\zeta = 0.02$ (Li et al., 2022). We show the value of $\Lambda_{\frac{d-2k}{\eta}}(m)$ when $d - 2k \in [1, 30] \cap \mathbb{Z}$ and $\eta = \frac{1}{n}, n \in [1, 50] \cap \mathbb{Z}$ in Table 5. As a result, all values in Table 5 are greater than $\frac{1}{2\theta} \approx 0.5005$, which means

$$\mathbb{E}_{z\sim\mathcal{P}}\left(\tilde{f}_{x_0}^*(\delta + z)\right) \geq \theta \cdot \frac{1}{2\theta} = \frac{1}{2}.$$

(62)

Recalling that our goal here is to check whether $\mathbb{E}_{z\sim\mathcal{P}}\left(\tilde{f}_{x_0}^*(\delta + z)\right) \geq \frac{1}{2}$ holds for some determined $\rho$, we have finished the proof that for base classifiers satisfy $(\sigma, p, \eta)$-concentration property with the smoothing distribution $\mathcal{P} = \mathcal{G}(\sigma, \eta, k)$ and the additional distribution $\mathcal{Q} = \mathcal{G}_t(\sigma, \eta, k, T)$ , $R_D \geq \rho = \zeta\sigma_s d^{\frac{1}{\eta}}$. Superficially, we get an $\Omega(d^{\frac{1}{\eta}})$ bound for the $\ell_2$ certified radius. Considering the convention that $\mathbb{E}r^2$ keeps a constant, we have

$$\sigma_s = 2^{-\frac{1}{\eta}}\sqrt{\frac{d\Gamma(\frac{d}{\eta})}{\Gamma(\frac{d+2}{\eta})}}\sigma,$$

(63)

then

$$\rho = \zeta\sigma 2^{-\frac{1}{\eta}}\sqrt{\frac{d\Gamma(\frac{d}{\eta})}{\Gamma(\frac{d+2}{\eta})}}d^{\frac{1}{\eta}} = \zeta\sigma 2^{-\frac{1}{\eta}}\sqrt{\frac{d}{\prod_{i=1}^{\frac{2}{\eta}}(\frac{d+2}{\eta} - i)}}d^{\frac{1}{\eta}}.$$

(64)

We notice

$$\lim_{d\to+\infty}\frac{\rho}{\sqrt{d}} = \lim_{d\to+\infty}\zeta\sigma 2^{-\frac{1}{\eta}}\sqrt{\frac{d^{\frac{2}{\eta}}}{\prod_{i=1}^{\frac{2}{\eta}}(\frac{d+2}{\eta} - i)}} = \zeta\sigma\left(\frac{\eta}{2}\right)^{\frac{1}{\eta}},$$

(65)

which by definition means $\rho = \Theta(\sqrt{d})$. Since $R_D \geq \rho$, we have $R_D = \Omega(\sqrt{d})$ at the scale of $\sigma$, which concludes the proof of Theorem 4. $\qquad\square$

**Remark.** Different with Theorem 1, this proof can not be generalized to $\eta > 2$ due to the property of factorial. In essence, $(\sigma, p, \eta)$-concentration assumption is slightly less strict for base classifiers than $(\sigma, p, 2)$-concentration assumption, and Theorem 1 and Theorem 4 can be considered as two different expressions of the same fact.

## D    PROOF OF THEOREM 2

Our proof is based on the level set method and the DSRS computational method from Yang et al. (2020) and Li (Li et al., 2022). We don't show mechanisms that have been clarified by them.

Theoretically, for every radius $\rho \in \mathbb{R}_+$, we have to exhaust all $\delta \in \mathbb{R}^n$ that satisfy $\|\delta\|_2 = \rho$ to confirm whether the value $\rho$ can be certified. However, as a result of the $\ell_2$-symmetry, when can just consider $\delta = (\rho, 0, 0, \cdots, 0)^T$ when computing, as a sufficient condition for all $\|\delta\|_2 = \rho$. See detailed proof for this in Zhang et al. (2020). By doing this, the certification for $\delta$ can be reduced to a one-dimensional binary search on $\rho$, which largely simplified the problem.

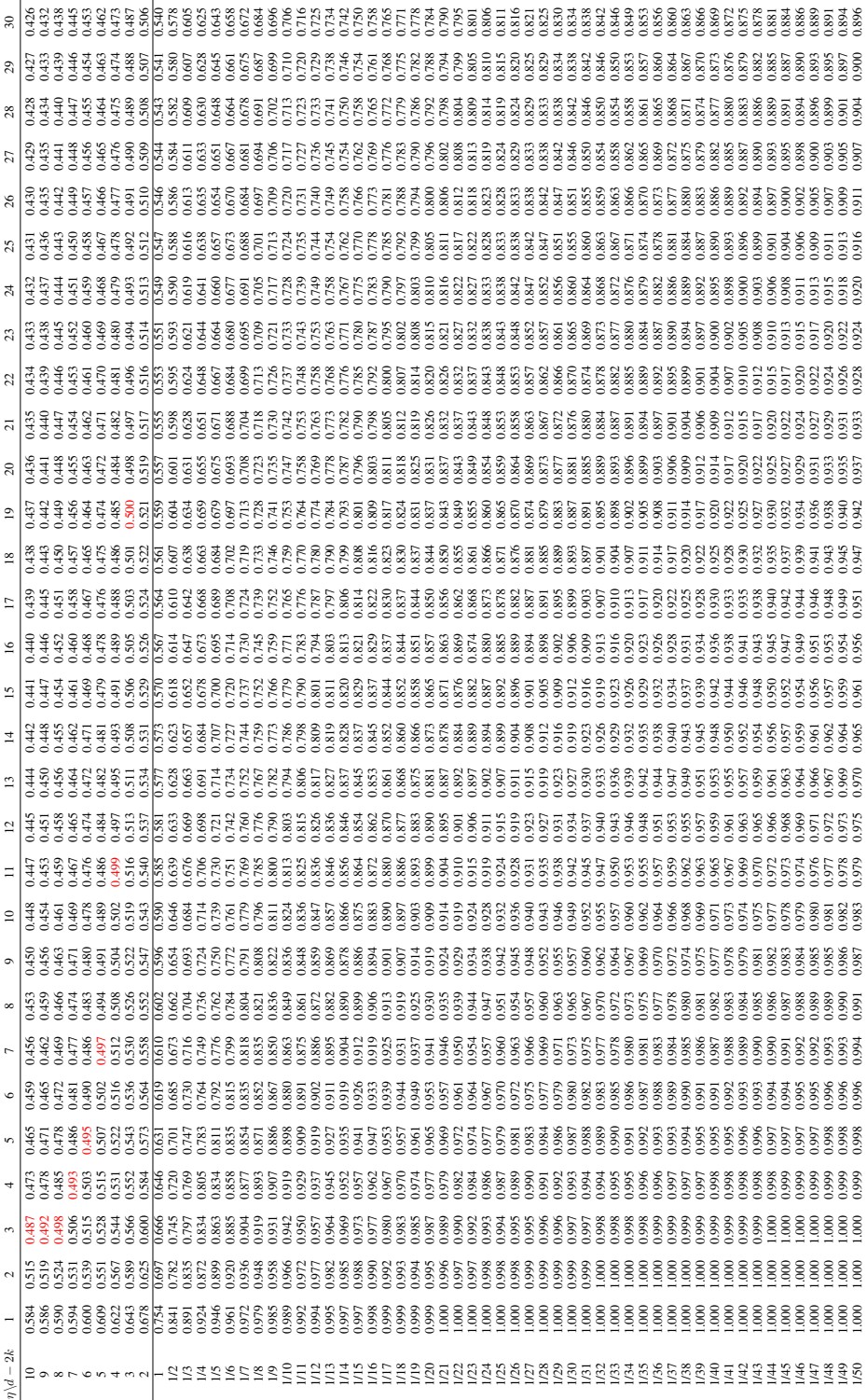

Table 4: Value of $\Lambda_{d-2k\frac{\cdot}{\eta}}(m)$ for Theorem 1.

Table 5: Value of $\Lambda_{\frac{d-2k}{\eta}}(m)$ for Theorem 4.

| $\eta\backslash d-2k$ | 1 | 2 | 3 | 4 | 5 | 6 | 7 | 8 | 9 | 10 | 11 | 12 | 13 | 14 | 15 | 16 | 17 | 18 | 19 | 20 | 21 | 22 | 23 | 24 | 25 | 26 | 27 | 28 | 29 | 30 |
|---|---|---|---|---|---|---|---|---|---|---|---|---|---|---|---|---|---|---|---|---|---|---|---|---|---|---|---|---|---|---|
| 1 | 0.753 | 0.696 | 0.665 | 0.644 | 0.628 | 0.617 | 0.607 | 0.599 | 0.592 | 0.586 | 0.581 | 0.577 | 0.572 | 0.569 | 0.565 | 0.562 | 0.559 | 0.556 | 0.554 | 0.552 | 0.549 | 0.547 | 0.545 | 0.543 | 0.541 | 0.540 | 0.538 | 0.537 | 0.535 | 0.534 |
| 1/2 | 0.838 | 0.778 | 0.741 | 0.714 | 0.694 | 0.678 | 0.665 | 0.654 | 0.645 | 0.636 | 0.629 | 0.623 | 0.617 | 0.611 | 0.606 | 0.602 | 0.598 | 0.594 | 0.590 | 0.587 | 0.584 | 0.581 | 0.578 | 0.575 | 0.572 | 0.570 | 0.568 | 0.565 | 0.563 | 0.561 |
| 1/3 | 0.889 | 0.830 | 0.790 | 0.761 | 0.738 | 0.720 | 0.704 | 0.691 | 0.680 | 0.670 | 0.661 | 0.654 | 0.646 | 0.640 | 0.634 | 0.628 | 0.623 | 0.619 | 0.614 | 0.610 | 0.606 | 0.602 | 0.599 | 0.596 | 0.592 | 0.589 | 0.586 | 0.584 | 0.581 | 0.579 |
| 1/4 | 0.921 | 0.866 | 0.827 | 0.796 | 0.772 | 0.752 | 0.735 | 0.721 | 0.708 | 0.697 | 0.687 | 0.678 | 0.670 | 0.663 | 0.656 | 0.650 | 0.644 | 0.638 | 0.633 | 0.629 | 0.624 | 0.620 | 0.616 | 0.612 | 0.608 | 0.605 | 0.602 | 0.598 | 0.595 | 0.592 |
| 1/5 | 0.943 | 0.894 | 0.855 | 0.824 | 0.799 | 0.778 | 0.760 | 0.745 | 0.732 | 0.720 | 0.709 | 0.699 | 0.690 | 0.682 | 0.675 | 0.668 | 0.661 | 0.655 | 0.650 | 0.645 | 0.640 | 0.635 | 0.630 | 0.626 | 0.622 | 0.618 | 0.614 | 0.611 | 0.608 | 0.604 |
| 1/6 | 0.959 | 0.915 | 0.878 | 0.847 | 0.822 | 0.801 | 0.782 | 0.766 | 0.752 | 0.739 | 0.728 | 0.717 | 0.708 | 0.699 | 0.691 | 0.684 | 0.677 | 0.670 | 0.664 | 0.659 | 0.653 | 0.648 | 0.643 | 0.639 | 0.634 | 0.630 | 0.626 | 0.622 | 0.618 | 0.615 |
| 1/7 | 0.970 | 0.931 | 0.896 | 0.867 | 0.841 | 0.820 | 0.801 | 0.784 | 0.770 | 0.756 | 0.745 | 0.734 | 0.724 | 0.715 | 0.706 | 0.698 | 0.691 | 0.684 | 0.677 | 0.671 | 0.666 | 0.660 | 0.655 | 0.650 | 0.645 | 0.641 | 0.636 | 0.632 | 0.628 | 0.624 |
| 1/8 | 0.978 | 0.944 | 0.911 | 0.883 | 0.858 | 0.836 | 0.817 | 0.801 | 0.785 | 0.772 | 0.760 | 0.748 | 0.738 | 0.728 | 0.719 | 0.711 | 0.703 | 0.696 | 0.689 | 0.683 | 0.677 | 0.671 | 0.666 | 0.660 | 0.655 | 0.651 | 0.646 | 0.642 | 0.637 | 0.633 |
| 1/9 | 0.983 | 0.954 | 0.924 | 0.897 | 0.873 | 0.851 | 0.832 | 0.815 | 0.800 | 0.786 | 0.773 | 0.762 | 0.751 | 0.741 | 0.732 | 0.723 | 0.715 | 0.708 | 0.700 | 0.694 | 0.687 | 0.681 | 0.675 | 0.670 | 0.665 | 0.660 | 0.655 | 0.650 | 0.646 | 0.641 |
| 1/10 | 0.988 | 0.962 | 0.935 | 0.909 | 0.885 | 0.864 | 0.845 | 0.828 | 0.813 | 0.799 | 0.786 | 0.774 | 0.763 | 0.753 | 0.743 | 0.734 | 0.726 | 0.718 | 0.711 | 0.704 | 0.697 | 0.691 | 0.685 | 0.679 | 0.673 | 0.668 | 0.663 | 0.658 | 0.654 | 0.649 |
| 1/11 | 0.991 | 0.969 | 0.944 | 0.919 | 0.897 | 0.876 | 0.857 | 0.840 | 0.824 | 0.810 | 0.797 | 0.785 | 0.774 | 0.763 | 0.754 | 0.744 | 0.736 | 0.728 | 0.720 | 0.713 | 0.706 | 0.699 | 0.693 | 0.687 | 0.682 | 0.676 | 0.671 | 0.666 | 0.661 | 0.656 |
| 1/12 | 0.993 | 0.974 | 0.952 | 0.928 | 0.907 | 0.886 | 0.868 | 0.851 | 0.835 | 0.821 | 0.808 | 0.795 | 0.784 | 0.773 | 0.763 | 0.754 | 0.745 | 0.737 | 0.729 | 0.722 | 0.715 | 0.708 | 0.701 | 0.695 | 0.689 | 0.684 | 0.678 | 0.673 | 0.668 | 0.663 |
| 1/13 | 0.995 | 0.979 | 0.958 | 0.936 | 0.915 | 0.896 | 0.878 | 0.861 | 0.845 | 0.831 | 0.817 | 0.805 | 0.794 | 0.783 | 0.773 | 0.763 | 0.754 | 0.746 | 0.737 | 0.730 | 0.723 | 0.716 | 0.709 | 0.703 | 0.697 | 0.691 | 0.685 | 0.680 | 0.675 | 0.670 |
| 1/14 | 0.996 | 0.983 | 0.964 | 0.943 | 0.923 | 0.904 | 0.886 | 0.870 | 0.854 | 0.840 | 0.827 | 0.814 | 0.803 | 0.792 | 0.781 | 0.772 | 0.762 | 0.754 | 0.746 | 0.738 | 0.730 | 0.723 | 0.716 | 0.710 | 0.704 | 0.698 | 0.692 | 0.686 | 0.681 | 0.676 |
| 1/15 | 0.997 | 0.986 | 0.969 | 0.950 | 0.930 | 0.912 | 0.895 | 0.878 | 0.863 | 0.849 | 0.835 | 0.823 | 0.811 | 0.800 | 0.789 | 0.780 | 0.770 | 0.762 | 0.753 | 0.745 | 0.738 | 0.730 | 0.723 | 0.717 | 0.710 | 0.704 | 0.698 | 0.692 | 0.687 | 0.682 |
| 1/16 | 0.998 | 0.988 | 0.973 | 0.955 | 0.937 | 0.919 | 0.902 | 0.886 | 0.871 | 0.856 | 0.843 | 0.831 | 0.819 | 0.808 | 0.797 | 0.787 | 0.778 | 0.769 | 0.760 | 0.752 | 0.744 | 0.737 | 0.730 | 0.723 | 0.717 | 0.710 | 0.704 | 0.698 | 0.693 | 0.687 |
| 1/17 | 0.998 | 0.990 | 0.976 | 0.960 | 0.943 | 0.925 | 0.909 | 0.893 | 0.878 | 0.864 | 0.851 | 0.838 | 0.826 | 0.815 | 0.804 | 0.794 | 0.785 | 0.776 | 0.767 | 0.759 | 0.751 | 0.744 | 0.736 | 0.729 | 0.723 | 0.716 | 0.710 | 0.704 | 0.698 | 0.693 |
| 1/18 | 0.999 | 0.992 | 0.979 | 0.964 | 0.948 | 0.931 | 0.915 | 0.900 | 0.885 | 0.871 | 0.858 | 0.845 | 0.833 | 0.822 | 0.811 | 0.801 | 0.792 | 0.783 | 0.774 | 0.765 | 0.757 | 0.750 | 0.742 | 0.735 | 0.729 | 0.722 | 0.716 | 0.710 | 0.704 | 0.698 |
| 1/19 | 0.999 | 0.993 | 0.982 | 0.968 | 0.952 | 0.937 | 0.921 | 0.906 | 0.891 | 0.877 | 0.864 | 0.852 | 0.840 | 0.829 | 0.818 | 0.808 | 0.798 | 0.789 | 0.780 | 0.772 | 0.764 | 0.756 | 0.748 | 0.742 | 0.734 | 0.728 | 0.721 | 0.715 | 0.709 | 0.703 |
| 1/20 | 0.999 | 0.994 | 0.984 | 0.971 | 0.957 | 0.941 | 0.926 | 0.911 | 0.897 | 0.884 | 0.870 | 0.858 | 0.846 | 0.835 | 0.824 | 0.814 | 0.804 | 0.795 | 0.786 | 0.778 | 0.769 | 0.761 | 0.754 | 0.747 | 0.740 | 0.733 | 0.726 | 0.720 | 0.714 | 0.708 |
| 1/21 | 0.999 | 0.995 | 0.986 | 0.974 | 0.960 | 0.946 | 0.931 | 0.917 | 0.903 | 0.889 | 0.876 | 0.864 | 0.852 | 0.841 | 0.830 | 0.820 | 0.810 | 0.801 | 0.792 | 0.783 | 0.775 | 0.767 | 0.759 | 0.752 | 0.745 | 0.738 | 0.731 | 0.725 | 0.719 | 0.713 |
| 1/22 | 1.000 | 0.996 | 0.988 | 0.977 | 0.964 | 0.950 | 0.936 | 0.922 | 0.908 | 0.895 | 0.882 | 0.870 | 0.858 | 0.847 | 0.836 | 0.826 | 0.816 | 0.806 | 0.797 | 0.789 | 0.780 | 0.772 | 0.765 | 0.757 | 0.750 | 0.743 | 0.736 | 0.730 | 0.723 | 0.717 |
| 1/23 | 1.000 | 0.996 | 0.989 | 0.979 | 0.967 | 0.954 | 0.940 | 0.926 | 0.913 | 0.900 | 0.887 | 0.875 | 0.863 | 0.852 | 0.841 | 0.831 | 0.821 | 0.812 | 0.803 | 0.794 | 0.786 | 0.777 | 0.770 | 0.762 | 0.755 | 0.748 | 0.741 | 0.734 | 0.728 | 0.722 |
| 1/24 | 1.000 | 0.997 | 0.990 | 0.981 | 0.970 | 0.957 | 0.944 | 0.931 | 0.917 | 0.905 | 0.892 | 0.880 | 0.869 | 0.857 | 0.847 | 0.836 | 0.826 | 0.817 | 0.808 | 0.799 | 0.791 | 0.782 | 0.774 | 0.767 | 0.759 | 0.752 | 0.745 | 0.739 | 0.732 | 0.726 |
| 1/25 | 1.000 | 0.998 | 0.991 | 0.983 | 0.972 | 0.960 | 0.948 | 0.935 | 0.922 | 0.909 | 0.897 | 0.885 | 0.873 | 0.862 | 0.852 | 0.841 | 0.831 | 0.822 | 0.813 | 0.804 | 0.795 | 0.787 | 0.779 | 0.772 | 0.764 | 0.757 | 0.750 | 0.743 | 0.737 | 0.730 |
| 1/26 | 1.000 | 0.998 | 0.992 | 0.985 | 0.975 | 0.963 | 0.951 | 0.938 | 0.926 | 0.913 | 0.901 | 0.890 | 0.878 | 0.867 | 0.856 | 0.846 | 0.836 | 0.826 | 0.816 | 0.809 | 0.800 | 0.792 | 0.784 | 0.776 | 0.769 | 0.761 | 0.754 | 0.747 | 0.741 | 0.734 |
| 1/27 | 1.000 | 0.998 | 0.993 | 0.986 | 0.977 | 0.966 | 0.954 | 0.942 | 0.930 | 0.918 | 0.906 | 0.894 | 0.883 | 0.872 | 0.861 | 0.851 | 0.841 | 0.831 | 0.822 | 0.813 | 0.805 | 0.796 | 0.788 | 0.780 | 0.773 | 0.765 | 0.758 | 0.751 | 0.745 | 0.738 |
| 1/28 | 1.000 | 0.999 | 0.994 | 0.988 | 0.979 | 0.968 | 0.957 | 0.945 | 0.933 | 0.921 | 0.910 | 0.898 | 0.887 | 0.876 | 0.865 | 0.855 | 0.845 | 0.836 | 0.827 | 0.818 | 0.809 | 0.801 | 0.792 | 0.785 | 0.777 | 0.770 | 0.762 | 0.755 | 0.749 | 0.742 |
| 1/29 | 1.000 | 0.999 | 0.995 | 0.989 | 0.981 | 0.971 | 0.960 | 0.948 | 0.937 | 0.925 | 0.913 | 0.902 | 0.891 | 0.880 | 0.870 | 0.860 | 0.850 | 0.840 | 0.831 | 0.822 | 0.813 | 0.805 | 0.797 | 0.789 | 0.781 | 0.774 | 0.766 | 0.759 | 0.753 | 0.746 |
| 1/30 | 1.000 | 0.999 | 0.996 | 0.990 | 0.982 | 0.973 | 0.962 | 0.951 | 0.940 | 0.929 | 0.917 | 0.906 | 0.895 | 0.884 | 0.874 | 0.864 | 0.854 | 0.844 | 0.835 | 0.826 | 0.818 | 0.809 | 0.800 | 0.793 | 0.785 | 0.778 | 0.770 | 0.763 | 0.756 | 0.750 |
| 1/31 | 1.000 | 0.999 | 0.996 | 0.991 | 0.984 | 0.975 | 0.965 | 0.954 | 0.943 | 0.932 | 0.921 | 0.910 | 0.899 | 0.888 | 0.878 | 0.868 | 0.858 | 0.848 | 0.839 | 0.830 | 0.821 | 0.813 | 0.805 | 0.797 | 0.789 | 0.781 | 0.774 | 0.767 | 0.760 | 0.753 |
| 1/32 | 1.000 | 0.999 | 0.997 | 0.992 | 0.985 | 0.977 | 0.967 | 0.957 | 0.946 | 0.935 | 0.924 | 0.913 | 0.902 | 0.892 | 0.882 | 0.871 | 0.862 | 0.852 | 0.843 | 0.834 | 0.825 | 0.817 | 0.808 | 0.800 | 0.793 | 0.785 | 0.778 | 0.770 | 0.763 | 0.757 |
| 1/33 | 1.000 | 0.999 | 0.997 | 0.993 | 0.986 | 0.978 | 0.969 | 0.959 | 0.949 | 0.938 | 0.927 | 0.916 | 0.906 | 0.895 | 0.885 | 0.875 | 0.865 | 0.856 | 0.847 | 0.838 | 0.829 | 0.820 | 0.812 | 0.804 | 0.796 | 0.789 | 0.781 | 0.774 | 0.767 | 0.760 |
| 1/34 | 1.000 | 1.000 | 0.998 | 0.993 | 0.987 | 0.980 | 0.971 | 0.961 | 0.951 | 0.941 | 0.930 | 0.920 | 0.909 | 0.899 | 0.889 | 0.879 | 0.869 | 0.860 | 0.850 | 0.841 | 0.833 | 0.824 | 0.816 | 0.808 | 0.800 | 0.792 | 0.785 | 0.777 | 0.770 | 0.763 |
| 1/35 | 1.000 | 1.000 | 0.998 | 0.994 | 0.988 | 0.981 | 0.973 | 0.963 | 0.954 | 0.943 | 0.933 | 0.923 | 0.912 | 0.902 | 0.892 | 0.882 | 0.873 | 0.863 | 0.854 | 0.845 | 0.836 | 0.828 | 0.819 | 0.811 | 0.803 | 0.796 | 0.788 | 0.781 | 0.774 | 0.767 |
| 1/36 | 1.000 | 1.000 | 0.998 | 0.995 | 0.989 | 0.982 | 0.974 | 0.965 | 0.956 | 0.946 | 0.936 | 0.926 | 0.915 | 0.905 | 0.895 | 0.885 | 0.876 | 0.866 | 0.857 | 0.848 | 0.840 | 0.831 | 0.823 | 0.815 | 0.807 | 0.799 | 0.791 | 0.784 | 0.777 | 0.770 |
| 1/37 | 1.000 | 1.000 | 0.998 | 0.995 | 0.990 | 0.984 | 0.976 | 0.967 | 0.958 | 0.948 | 0.938 | 0.928 | 0.918 | 0.908 | 0.898 | 0.889 | 0.879 | 0.870 | 0.861 | 0.852 | 0.843 | 0.834 | 0.826 | 0.818 | 0.810 | 0.802 | 0.795 | 0.787 | 0.780 | 0.773 |
| 1/38 | 1.000 | 1.000 | 0.999 | 0.996 | 0.991 | 0.985 | 0.977 | 0.969 | 0.960 | 0.951 | 0.941 | 0.931 | 0.921 | 0.911 | 0.901 | 0.892 | 0.882 | 0.873 | 0.864 | 0.855 | 0.846 | 0.838 | 0.829 | 0.821 | 0.813 | 0.805 | 0.798 | 0.790 | 0.783 | 0.776 |
| 1/39 | 1.000 | 1.000 | 0.999 | 0.996 | 0.992 | 0.986 | 0.979 | 0.971 | 0.962 | 0.953 | 0.943 | 0.934 | 0.924 | 0.914 | 0.904 | 0.895 | 0.885 | 0.876 | 0.867 | 0.858 | 0.849 | 0.841 | 0.832 | 0.824 | 0.816 | 0.809 | 0.801 | 0.793 | 0.786 | 0.779 |
| 1/40 | 1.000 | 1.000 | 0.999 | 0.996 | 0.992 | 0.987 | 0.980 | 0.972 | 0.964 | 0.955 | 0.946 | 0.936 | 0.926 | 0.917 | 0.907 | 0.898 | 0.888 | 0.879 | 0.870 | 0.861 | 0.852 | 0.844 | 0.836 | 0.827 | 0.819 | 0.812 | 0.804 | 0.796 | 0.789 | 0.782 |
| 1/41 | 1.000 | 1.000 | 0.999 | 0.997 | 0.993 | 0.988 | 0.981 | 0.974 | 0.966 | 0.957 | 0.948 | 0.938 | 0.929 | 0.919 | 0.910 | 0.901 | 0.891 | 0.882 | 0.873 | 0.864 | 0.855 | 0.847 | 0.839 | 0.830 | 0.822 | 0.815 | 0.807 | 0.799 | 0.792 | 0.785 |
| 1/42 | 1.000 | 1.000 | 0.999 | 0.997 | 0.994 | 0.989 | 0.982 | 0.975 | 0.967 | 0.959 | 0.950 | 0.941 | 0.931 | 0.922 | 0.913 | 0.903 | 0.894 | 0.885 | 0.876 | 0.867 | 0.858 | 0.850 | 0.842 | 0.833 | 0.825 | 0.817 | 0.810 | 0.802 | 0.795 | 0.788 |
| 1/43 | 1.000 | 1.000 | 0.999 | 0.997 | 0.994 | 0.990 | 0.983 | 0.977 | 0.969 | 0.961 | 0.952 | 0.943 | 0.934 | 0.924 | 0.915 | 0.906 | 0.897 | 0.888 | 0.879 | 0.870 | 0.861 | 0.853 | 0.844 | 0.836 | 0.828 | 0.820 | 0.813 | 0.805 | 0.798 | 0.790 |
| 1/44 | 1.000 | 1.000 | 0.999 | 0.998 | 0.995 | 0.990 | 0.984 | 0.978 | 0.970 | 0.962 | 0.954 | 0.945 | 0.936 | 0.927 | 0.918 | 0.908 | 0.899 | 0.890 | 0.881 | 0.873 | 0.864 | 0.856 | 0.847 | 0.839 | 0.831 | 0.823 | 0.815 | 0.808 | 0.800 | 0.793 |
| 1/45 | 1.000 | 1.000 | 0.999 | 0.998 | 0.995 | 0.991 | 0.985 | 0.979 | 0.972 | 0.964 | 0.956 | 0.947 | 0.938 | 0.929 | 0.920 | 0.911 | 0.902 | 0.893 | 0.884 | 0.875 | 0.867 | 0.858 | 0.850 | 0.842 | 0.834 | 0.826 | 0.818 | 0.810 | 0.803 | 0.796 |
| 1/46 | 1.000 | 1.000 | 0.999 | 0.998 | 0.995 | 0.991 | 0.986 | 0.980 | 0.973 | 0.965 | 0.957 | 0.949 | 0.940 | 0.931 | 0.922 | 0.913 | 0.904 | 0.895 | 0.887 | 0.878 | 0.869 | 0.861 | 0.853 | 0.844 | 0.836 | 0.828 | 0.821 | 0.813 | 0.806 | 0.798 |
| 1/47 | 1.000 | 1.000 | 1.000 | 0.998 | 0.996 | 0.992 | 0.987 | 0.981 | 0.974 | 0.967 | 0.959 | 0.951 | 0.942 | 0.933 | 0.924 | 0.916 | 0.907 | 0.898 | 0.889 | 0.881 | 0.872 | 0.864 | 0.855 | 0.847 | 0.839 | 0.831 | 0.823 | 0.816 | 0.808 | 0.801 |
| 1/48 | 1.000 | 1.000 | 1.000 | 0.998 | 0.996 | 0.993 | 0.988 | 0.982 | 0.976 | 0.968 | 0.961 | 0.952 | 0.944 | 0.935 | 0.927 | 0.918 | 0.909 | 0.900 | 0.892 | 0.883 | 0.874 | 0.866 | 0.858 | 0.850 | 0.842 | 0.834 | 0.826 | 0.818 | 0.811 | 0.803 |
| 1/49 | 1.000 | 1.000 | 1.000 | 0.999 | 0.996 | 0.993 | 0.989 | 0.983 | 0.977 | 0.970 | 0.962 | 0.954 | 0.946 | 0.937 | 0.929 | 0.920 | 0.911 | 0.903 | 0.894 | 0.885 | 0.877 | 0.869 | 0.860 | 0.852 | 0.844 | 0.836 | 0.828 | 0.821 | 0.813 | 0.806 |
| 1/50 | 1.000 | 1.000 | 1.000 | 0.999 | 0.997 | 0.994 | 0.989 | 0.984 | 0.978 | 0.971 | 0.964 | 0.956 | 0.948 | 0.939 | 0.931 | 0.922 | 0.914 | 0.905 | 0.896 | 0.888 | 0.879 | 0.871 | 0.863 | 0.855 | 0.847 | 0.839 | 0.831 | 0.823 | 0.816 | 0.808 |

The following part is the computational method for Problem (9). For $\mathcal{P} = \mathcal{G}(\sigma, \eta, k)$ and $\mathcal{Q} = \mathcal{G}_t(\sigma, \eta, k, T)$ with PDFs $p(\cdot)$ and $q(\cdot)$ respectively, we have

$$\mathbb{P}_{z \sim \mathcal{P}}\{z \in \mathcal{V}\}$$
$$=\mathbb{P}_{z \sim \mathcal{P}}\{p(z - \delta) + \nu_1 p(z) + \nu_2 q(z) < 0\}$$
$$= \int_T^\infty \phi_g(r) \frac{2\pi^{\frac{d}{2}}}{\Gamma(\frac{d}{2})} r^{d-1} dr \cdot \mathbb{P}\{p(x - \delta) + \nu_1 p(x) < 0| \, \|x\|_2 = r\}$$
$$+ \int_0^T \phi_g(r) \frac{2\pi^{\frac{d}{2}}}{\Gamma(\frac{d}{2})} r^{d-1} dr \cdot \mathbb{P}\{p(x - \delta) + (\nu_1 + C_g\nu_2)p(x) < 0| \, \|x\|_2 = r\}$$
$$= \frac{1}{\Gamma(\frac{d-2k}{\eta})} \int_{\frac{T^\eta}{2\sigma_g^\eta}}^\infty u^{\frac{d-2k}{\eta} - 1} \exp(-u)du \cdot \mathbb{P}\{p(x - \delta) + \nu_1 p(x) < 0\}| \, \|x\|_2 = \sigma_g(2u)^{\frac{1}{\eta}}\}$$
$$+ \frac{1}{\Gamma(\frac{d-2k}{\eta})} \int_0^{\frac{T^\eta}{2\sigma_g^\eta}} u^{\frac{d-2k}{\eta} - 1} \exp(-u)du \cdot \mathbb{P}\{p(x - \delta) + (\nu_1 + C_g\nu_2)p(x) < 0| \, \|x\|_2 = \sigma_g(2u)^{\frac{1}{\eta}}\}$$
$$= \mathbb{E}_{u \sim \Gamma(\frac{d-2k}{\eta}, 1)} \begin{cases} \omega_1(u, \nu_1), & u \geq \frac{T^\eta}{2\sigma_g^\eta}, \\ \omega_1(u, \nu_1 + C_g\nu_2), & u < \frac{T^\eta}{2\sigma_g^\eta}, \end{cases}$$

(66)

where $\omega_1(u, \nu) = \mathbb{P}\{p(x - \delta) + \nu p(x) < 0| \, \|x\|_2 = \sigma_g(2u)^{\frac{1}{\eta}}\}$. Here we notice

$$p(x - \delta) < -\nu p(x) \iff \phi_g(\|x - \delta\|_2) < -\nu \phi_g(\|x\|_2) \iff \|x - \delta\|_2 > \phi_g^{-1}(-\nu \phi_g(\|x\|_2)), \quad (67)$$

Since $x = (x_1, x_2, \cdots, x_d)^T, \delta = (\rho, 0, 0, \cdots, 0)^T$, we let $\|x\|_2 = \sigma_g(2u)^{\frac{1}{\eta}}$ to get

$$\begin{cases} x_1^2 + \sum_{i=2}^d x_i^2 = (2\sigma_g^\eta u)^{\frac{2}{\eta}}, \\ (x_1 - \rho)^2 + \sum_{i=2}^d x_i^2 \geq \phi_g^{-1}(-\nu \phi_g(\|x\|_2))^2. \end{cases}$$

(68)

In solving this inequality system, we have

$$x_1 \leq \frac{\rho^2 + \sigma_g(2u)^{\frac{2}{\eta}} - \phi_g^{-1}(-\nu \phi_g(\sigma_g(2u)^{\frac{1}{\eta}})^2}{2\rho}. \quad (69)$$

We write $\xi$ for $\phi_g^{-1}(-\nu \phi_g(\sigma_g(2u)^{\frac{1}{\eta}})$, then

$$\phi_g(\xi) = -\nu \phi_g(\sigma_g(2u)^{\frac{1}{\eta}}). \quad (70)$$

Combining Equation (69) and Equation (11b), we obtain

$$\frac{\eta}{2} \frac{1}{(2\sigma_g^\eta)^{\frac{d-2k}{\eta}} \pi^{\frac{d}{2}}} \frac{\Gamma(\frac{d}{2})}{\Gamma(\frac{d-2k}{\eta})} \xi^{-2k} \exp(-\frac{\xi^\eta}{2\sigma_g^\eta}) = -\frac{\nu \eta}{2} \frac{1}{(2\sigma_g^\eta)^{\frac{d-2k}{\eta}} \pi^{\frac{d}{2}}} \frac{\Gamma(\frac{d}{2})}{\Gamma(\frac{d-2k}{\eta})} \sigma_g^{-2k} (2u)^{\frac{-2k}{\eta}} \exp(-u)$$
$$\iff \frac{1}{(2\sigma_g^\eta)^{\frac{-2k}{\eta}}} \xi^{-2k} \exp(-\frac{1}{2}(\frac{\xi}{\sigma_g})^\eta) = -\nu u^{\frac{-2k}{\eta}} \exp(-u).$$

(71)

Solving Equation (71) to get

$$\xi = \left( \frac{4kW(\frac{\eta u}{2k}(-\nu)^{-\frac{\eta}{2k}} \exp(\frac{\eta u}{2k}))\sigma_g^\eta}{\eta} \right)^{\frac{1}{\eta}}, \quad (72)$$

where $W(\cdot)$ is the principal branch of the Lambert W function. Injecting $\xi^2$ into Equation (69), we have

$$x_1 \leq \frac{\rho^2 + \sigma_g^2(2u)^{\frac{2}{\eta}} - \left( \frac{4kW(\frac{\eta u}{2k}(-\nu)^{-\frac{\eta}{2k}} \exp(\frac{\eta u}{2k}))\sigma_g^\eta}{\eta} \right)^{\frac{2}{\eta}}}{2\rho}. \quad (73)$$

To compute Equation (73), we have the following lemma:

**Lemma D.1.** *(Lemma I.23 in Yang et al. (2020)) If $(x_1, \cdots, x_d)$ is sampled uniformly from the unit hypersphere $S^{d-1} \subseteq \mathbb{R}^d$, then $\frac{1+x_1}{2}$ is distributed as $Beta\left(\frac{d-1}{2}, \frac{d-1}{2}\right)$.*

Obviously,

$$\frac{1 + \frac{x_1}{\sigma_g(2u)^{\frac{1}{\eta}}}}{2} \leq \frac{(\rho + \sigma_g(2u)^{\frac{1}{\eta}})^2 - \left(\frac{4kW(\frac{\eta u}{2k}(-\nu)^{-\frac{\eta}{2k}}\exp(\frac{\eta u}{2k}))\sigma_g^\eta}{\eta}\right)^{\frac{2}{\eta}}}{4\rho\sigma_g(2u)^{\frac{1}{\eta}}}. \tag{74}$$

Combining Equation (74) and Lemma D.1, we get

$$\omega_1(u, \nu) = \Psi_{\frac{d-1}{2}}\left(\frac{(\rho + \sigma_g(2u)^{\frac{1}{\eta}})^2 - \left(\frac{4kW(\frac{\eta u}{2k}(-\nu)^{-\frac{\eta}{2k}}\exp(\frac{\eta u}{2k}))\sigma_g^\eta}{\eta}\right)^{\frac{2}{\eta}}}{4\rho\sigma_g(2u)^{\frac{1}{\eta}}}\right). \tag{75}$$

Now with the expression of $\omega_1(u, \nu)$, $\mathbb{P}_{z\sim\mathcal{P}}\{z \in \mathcal{V}\}$ is calculable by numerical integration.

The computation of $\mathbb{P}_{z\sim\mathcal{Q}}\{z \in \mathcal{V}\}$ is a simplified version of $\mathbb{P}_{z\sim\mathcal{P}}\{z \in \mathcal{V}\}$ with differences lying in the PDF and the integral interval.

$$\begin{aligned}
&\mathbb{P}_{z\sim\mathcal{Q}}\{z \in \mathcal{V}\} \\
=&\mathbb{P}_{z\sim\mathcal{Q}}\{p(z-\delta) + \nu_1 p(z) + \nu_2 q(z) < 0\} \\
=&\int_0^T \phi_g(r, T)\frac{2\pi^{\frac{d}{2}}}{\Gamma(\frac{d}{2})}r^{d-1}dr \cdot \mathbb{P}\{p(x-\delta) + (\nu_1 + C_g\nu_2)p(x) < 0|\,\|x\|_2 = r\} \\
=&\frac{1}{\gamma(\frac{d-2k}{\eta}, \frac{T^\eta}{2\sigma_g^\eta})}\int_0^{\frac{T^\eta}{2\sigma_g^\eta}} u^{\frac{d-2k}{\eta}-1}e^{-u} \cdot \mathbb{P}\{p(x-\delta) < -(\nu_1 + C_g\nu_2)p(x)|\,\|x\|_2 = \sigma_g(2u)^{\frac{1}{\eta}}\}du \\
=&C_g\mathbb{E}_{u\sim\Gamma(\frac{d-2k}{\eta},1)}\,\omega_1(u, \nu_1 + C_g\nu_2) \cdot \mathbb{1}_{u\leq\frac{T^\eta}{2\sigma_g^\eta}}.
\end{aligned} \tag{76}$$

For $\mathbb{P}_{z\sim\mathcal{P}+\delta}\{z \in \mathcal{V}\}$, there are significant differences in the computation as we are now considering the shifted distribution $z \sim \mathcal{P} + \delta$. Noticing

$$\mathbb{P}_{z\sim\mathcal{P}+\delta}\{p(z-\delta) + \nu_1 p(z) + \nu_2 q(z) < 0\} \iff \mathbb{P}_{z\sim\mathcal{P}}\{p(z) + \nu_1 p(z+\delta) + \nu_2 q(z+\delta) < 0\}, \tag{77}$$

we have

$$\begin{aligned}
&\mathbb{P}_{z\sim\mathcal{P}+\delta}\{z \in \mathcal{V}\} \\
=&\mathbb{P}_{z\sim\mathcal{P}+\delta}\{p(z-\delta) + \nu_1 p(z) + \nu_2 q(z) < 0\} \\
=&\mathbb{P}_{z\sim\mathcal{P}}\{p(z) + \nu_1 p(z+\delta) + \nu_2 q(z+\delta) < 0\} \\
=&\int_0^\infty \phi_g(r)\frac{2\pi^{\frac{d}{2}}}{\Gamma(\frac{d}{2})}r^{d-1}dr \cdot \mathbb{P}\{p(x) + \nu_1 p(x+\delta) + \nu_2 q(x+\delta) < 0|\,\|x\|_2 = r\} \\
=&\frac{1}{\Gamma(\frac{d-2k}{\eta})}\int_0^\infty u^{\frac{d-2k}{\eta}-1}e^{-u}du \cdot \mathbb{P}\{p(x) + \nu_1 p(x+\delta) + \nu_2 q(x+\delta) < 0|\,\|x\|_2 = \sigma_g(2u)^{\frac{1}{\eta}}\}.
\end{aligned} \tag{78}$$

Here we have to analyze bound cases because there is a term $\nu_2 q(x+\delta)$, where $q(\cdot)$ is not continuous in the $\mathbb{R}^d$ space, and $x + \delta$ can not be handled well by piecewise integration like $\mathbb{P}_{z\sim\mathcal{P}}\{z \in \mathcal{V}\}$. Thus, we consider the value of $q(x + \delta)$ instead.

Branch (1): $q(x + \delta) > 0$.

This branch is equivalent to $\|x + \delta\|_2 \leq T$. For $\|x\|_2 = \sigma_g(2u)^{\frac{1}{\eta}}$, we have

$$\begin{cases} x_1^2 + \sum_{i=2}^d x_i^2 = (2\sigma_g^\eta u)^{\frac{2}{\eta}}, \\ (x_1 + \rho)^2 + \sum_{i=2}^d x_i^2 \leq T^2. \end{cases} \tag{79}$$

Solving the inequality system, we get

$$x_1 \leq \frac{T^2 - \rho^2 - (2\sigma_g^\eta u)^{\frac{2}{\eta}}}{2\rho}. \tag{80}$$

We also see that

$$
\begin{aligned}
&p(x) + \nu_1 p(x+\delta) + \nu_2 q(x+\delta) < 0 \\
\Longleftrightarrow & p(x) \leq -(\nu_1 p(x+\delta) + \nu_2 q(x+\delta)) \\
\Longleftrightarrow & p(x) \leq -(\nu_1 + C_g\nu_2)p(x+\delta) \\
\Longleftrightarrow & \phi_g(\|x\|_2) \leq -(\nu_1 + C_g\nu_2)\phi_g(\|x+\delta\|_2) \\
\overset{(a)}{\Longleftrightarrow} & \phi_g(\|x+\delta\|_2) \geq -\frac{1}{(\nu_1 + C_g\nu_2)}\phi_g(\|x\|_2) \\
\Longleftrightarrow & \|x+\delta\|_2 \leq \phi_g^{-1}\left(-\frac{1}{(\nu_1 + C_g\nu_2)}\phi_g(\|x\|_2)\right),
\end{aligned}
\tag{81}
$$

where (a) is because $\nu_1 + C_g\nu_2$ is handled as a whole, only the case under $\nu_1 + C_g\nu_2 < 0$ is considered in binary search algorithms. See details in Appendix E.1 from Li et al. (2022). Now we combine Equation (81) and Equation (11b) to obtain

$$x_1 \leq \frac{\sigma_g^2\left(\frac{4kW(\frac{\eta u}{2k}(-\nu_1 - C_g\nu_2)^{\frac{\eta}{2k}}\exp(\frac{\eta u}{2k}))}{\eta}\right)^{\frac{2}{\eta}} - \sigma_g^2(2u)^{\frac{2}{\eta}} - \rho^2}{2\rho}. \tag{82}$$

Utilizing the property of the beta distribution (Lemma D.1) again and taking the intersection of two restrictions, we get

$$
\begin{aligned}
&\mathbb{P}\{p(x) + \nu_1 p(x+\delta) + \nu_2 q(x+\delta) < 0 \wedge q(x+\delta) > 0 | \|x\|_2 = \sigma_g(2u)^{\frac{1}{\eta}}\} \\
=&\Psi_{\frac{d-1}{2}}\left(\frac{\min\{T^2, \sigma_g^2(\frac{4kW(\frac{\eta u}{2k}(-\nu_1 - C_g\nu_2)^{\frac{\eta}{2k}}\exp(\frac{\eta u}{2k}))}{\eta})^{\frac{2}{\eta}}\} - (\sigma_g(2u)^{\frac{1}{\eta}} - \rho)^2}{4\rho\sigma_g(2u)^{\frac{1}{\eta}}}\right).
\end{aligned}
\tag{83}
$$

Branch (2): $q(x+\delta) = 0$.

Noticing that when $q(x+\delta) = 0$,

$$p(x) + \nu_1 p(x+\delta) + \nu_2 q(x+\delta) < 0 \Longleftrightarrow p(x) + \nu_1 p(x+\delta) < 0. \tag{84}$$

Like Equation (81), we only consider $\nu_1 < 0$ since if $\nu_1 \geq 0$, the inequality above can never holds. We have

$$
\begin{cases}
x_1^2 + \sum_{i=2}^{d} x_i^2 = (2\sigma_g^\eta u)^{\frac{2}{\eta}}, \\
(x_1 + \rho)^2 + \sum_{i=2}^{d} x_i^2 > T^2.
\end{cases}
\tag{85}
$$

Thus,

$$x_1 > \frac{T^2 - \rho^2 - \sigma_g^2(2u)^{\frac{2}{\eta}}}{2\rho}. \tag{86}$$

Since

$$
\begin{aligned}
&p(x) + \nu_1 p(x+\delta) < 0 \\
\Longleftrightarrow & p(x) \leq -\nu_1 p(x+\delta) \\
\Longleftrightarrow & \phi_g(\|x\|_2) \leq -\nu_1 \phi_g(\|x+\delta\|_2) \\
\Longleftrightarrow & \phi_g(\|x+\delta\|_2) \geq -\frac{1}{\nu_1}\phi_g(\|x\|_2) \\
\Longleftrightarrow & \|x+\delta\|_2 \leq \phi_g^{-1}\left(-\frac{1}{\nu_1}\phi_g(\|x\|_2)\right),
\end{aligned}
\tag{87}
$$

we also have

$$x_1 \leq \frac{\sigma_g^2\left(\frac{4kW(\frac{\eta u}{2k}(-\nu_1)^{\frac{\eta}{2k}}\exp(\frac{\eta u}{2k}))}{\eta}\right)^{\frac{2}{\eta}} - \sigma_g^2(2u)^{\frac{2}{\eta}} - \rho^2}{2\rho},$$

(88)

by combining Equation (87) and Equation (11b). Thus we get

$$\frac{T^2 - \rho^2 - \sigma_g(2u)^{\frac{2}{\eta}}}{2\rho} < x_1 \leq \frac{\sigma_g^2\left(\frac{4kW(\frac{\eta u}{2k}(-\nu_1)^{\frac{\eta}{2k}}\exp(\frac{\eta u}{2k}))}{\eta}\right)^{\frac{2}{\eta}} - \sigma_g^2(2u)^{\frac{2}{\eta}} - \rho^2}{2\rho}.$$

(89)

By Lemma D.1, we obtain

$$\mathbb{P}\{p(x) + \nu_1 p(x+\delta) + \nu_2 q(x+\delta) < 0 \wedge q(x+\delta) = 0|\ \|x\|_2 = \sigma_g(2u)^{\frac{1}{\eta}}\}$$

$$=\Psi_{\frac{d-1}{2}}\left(\frac{\sigma_g^2\left(\frac{4kW(\frac{\eta u}{2k}(-\nu_1)^{\frac{\eta}{2k}}\exp(\frac{\eta u}{2k}))}{\eta}\right)^{\frac{2}{\eta}} - (\sigma_g(2u)^{\frac{1}{\eta}} - \rho)^2}{4\rho\sigma_g(2u)^{\frac{1}{\eta}}}\right) - \Psi_{\frac{d-1}{2}}\left(\frac{T^2 - (\sigma_g(2u)^{\frac{1}{\eta}} - \rho)^2}{4\rho\sigma_g(2u)^{\frac{1}{\eta}}}\right).$$

(90)

For brevity, we write

$$\omega_2(u) = \Psi_{\frac{d-1}{2}}\left(\frac{\min\{T^2, \sigma_g^2\left(\frac{4kW(\frac{\eta u}{2k}(-\nu_1-C_g\nu_2)^{\frac{\eta}{2k}}\exp(\frac{\eta u}{2k}))}{\eta}\right)^{\frac{2}{\eta}}\} - (\sigma_g(2u)^{\frac{1}{\eta}} - \rho)^2}{4\rho\sigma_g(2u)^{\frac{1}{\eta}}}\right),$$

$$\omega_3(u) = \Psi_{\frac{d-1}{2}}\left(\frac{\sigma_g^2\left(\frac{4kW(\frac{\eta u}{2k}(-\nu_1)^{\frac{\eta}{2k}}\exp(\frac{\eta u}{2k}))}{\eta}\right)^{\frac{2}{\eta}} - (\sigma_g(2u)^{\frac{1}{\eta}} - \rho)^2}{4\rho\sigma_g(2u)^{\frac{1}{\eta}}}\right) - \Psi_{\frac{d-1}{2}}\left(\frac{T^2 - (\sigma_g(2u)^{\frac{1}{\eta}} - \rho)^2}{4\rho\sigma_g(2u)^{\frac{1}{\eta}}}\right).$$

(91)

Finally, combining the two cases gives

$$\mathbb{P}\{p(x) + \nu_1 p(x+\delta) + \nu_2 q(x+\delta) < 0|\ \|x\|_2 = \sigma_g(2u)^{\frac{1}{\eta}}\}$$

$$=\mathbb{P}\{p(x) + \nu_1 p(x+\delta) + \nu_2 q(x+\delta) < 0 \wedge q(x+\delta) > 0|\ \|x\|_2 = \sigma_g(2u)^{\frac{1}{\eta}}\}$$

$$+ \mathbb{P}\{p(x) + \nu_1 p(x+\delta) + \nu_2 q(x+\delta) < 0 \wedge q(x+\delta) = 0|\ \|x\|_2 = \sigma_g(2u)^{\frac{1}{\eta}}\}$$

(92)

$$=\begin{cases} \omega_2(u), & \nu_1 \geq 0, \\ \omega_2(u) + \omega_3(u), & \nu_1 < 0. \end{cases}$$

Therefore,

$$\mathbb{P}_{z\sim\mathcal{P}+\delta}\{z \in \mathcal{V}\}$$

$$=\frac{1}{\Gamma(\frac{d-2k}{\eta})}\int_0^\infty u^{\frac{d-2k}{\eta}-1}e^{-u}du \cdot \mathbb{P}\{p(x) + \nu_1 p(x+\delta) + \nu_2 q(x+\delta) < 0|\ \|x\|_2 = \sigma_g(2u)^{\frac{1}{\eta}}\}$$

$$=\begin{cases} \mathbb{E}_{u\sim\Gamma(\frac{d-2k}{\eta},1)}\ \omega_2(u), & \nu_1 \geq 0, \\ \mathbb{E}_{u\sim\Gamma(\frac{d-2k}{\eta},1)}\{\omega_2(u) + \omega_3(u)\}, & \nu_1 < 0. \end{cases}$$

(93)

Now, we have already completed the proof of Theorem 2.

## E PROOF OF THEOREM 3

The proof of Theorem 3 is similar to that of Theorem 2, thereby we only show the parts with significant differences in this section.

**Probability density functions.** The first difference lies in PDFs of EGG and ESG, see Appendix A.

$\omega$ **Functions.** There are essential differences between the computations of EGG and ESG lying in the $\omega$ functions. In form, these three $\omega$ functions are designed totally symmetric for EGG and ESG. We learn that for EGG, searching $\nu_1 + C_g\nu_2$ and $\nu_1$ in $(-\infty, 0)$ is enough for obtaining the value of $\omega$ functions. Nevertheless, we will see for different $\omega$ functions, there are subtle differences in the computations for EGG and ESG.

(1) Function $\omega_1$.

Consider $\omega_1(u, \nu) = \mathbb{P}\{p(x - \delta) + \nu p(x) < 0 | \; \|x\|_2 = \sigma_s(2u)^{\frac{1}{\eta}}\}$. Obviously, $\phi_s(r)$ is a monotonically decreasing function with respect to $r$. Since all PDFs of $\mathcal{S}(\sigma, \eta)$ are bounded functions, we set the upper bound to be $U$ for convenience. Let $r = 0$, we have

$$U = \frac{\eta}{2} \frac{1}{(2\sigma_s^\eta)^{\frac{d}{\eta}} \pi^{\frac{d}{2}}} \frac{\Gamma(\frac{d}{2})}{\Gamma(\frac{d}{\eta})} \tag{94}$$

for $\phi_s(r)$. Namely, $\forall x \in [0, \infty)$, $p(x) \in (0, U]$, where $U < +\infty$ is a constant for any determined distribution. As a result, if for a specific $x$, we have $-\nu p(x) > U$, the probability $\mathbb{P}\{p(x - \delta) + \nu p(x) < 0 | \; \|x\|_2 = \sigma_s(2u)^{\frac{1}{\eta}}\}$ will always be 1. Next, we suppose $-\nu p(x) \in (0, U]$, otherwise $-\nu p(x)$ is outside the domain of $\phi_s^{-1}(x)$. When $\|x\|_2 = \sigma_s(2u)^{\frac{1}{\eta}}$, we have

$$\begin{aligned}
& p(x - \delta) + \nu p(x) < 0 \\
\Longleftrightarrow & p(x - \delta) \le -\nu p(x) \\
\Longleftrightarrow & \phi_s(\|x - \delta\|_2) \le -\nu \phi_s(\|x\|_2) \\
\Longleftrightarrow & \|x - \delta\|_2 \ge \phi_s^{-1}(-\nu \phi_s(\|x\|_2)) \\
\Longleftrightarrow & \|x - \delta\|_2 \ge \phi_s^{-1}(-\nu U \exp(-u)).
\end{aligned} \tag{95}$$

Then we are ready to solve

$$0 < -\nu U \exp(-u) \le U, \tag{96}$$

where $u \ge 0$. For $\omega_1$, $\nu$ is always negative by our binary search setting, so the left side $0 < -\nu U \exp(-u)$ always holds. For the right side, we notice

$$-\nu U \exp(-u) \le U \Longleftrightarrow \exp(-u) \le -\frac{1}{\nu} \Longleftrightarrow u - \ln(-\nu) \ge 0. \tag{97}$$

Now we know for $\nu < 0$,

$$u - \ln(-\nu) \ge 0 \Longleftrightarrow -\nu p(x) \in (0, U], \tag{98}$$

and

$$u - \ln(-\nu) < 0 \Longleftrightarrow -\nu p(x) \in (U, \infty], \tag{99}$$

which means $\mathbb{P}\{p(x - \delta) + \nu p(x) < 0 | \; \|x\|_2 = \sigma_s(2u)^{\frac{1}{\eta}}\} = 1$ when $u - \ln(-\nu) < 0$. Therefore,

$$\omega_1(u, \nu) = \begin{cases} \Psi_{\frac{d-1}{2}} \left( \dfrac{(\rho + \sigma_s(2u)^{\frac{1}{\eta}})^2 - \left(\phi_s^{-1}(-\nu \phi_s(\sigma_s(2u)^{\frac{1}{\eta}}))\right)^2}{4\rho \sigma_s(2u)^{\frac{1}{\eta}}} \right) & , u - \ln(-\nu) \ge 0, \\ 1 & , u - \ln(-\nu) < 0. \end{cases} \tag{100}$$

(2) Function $\omega_2$.

The $\omega_2$ function is for calculating $\mathbb{P}\{p(x) + \nu_1 p(x + \delta) + \nu_2 q(x + \delta) < 0 \land q(x + \delta) > 0 | \; \|x\|_2 = \sigma_g(2u)^{\frac{1}{\eta}}\}$. When $-\frac{1}{\nu_1 + C_s \nu_2} p(x) \in (0, U]$, we have

$$\begin{aligned}
& p(x) + \nu_1 p(x + \delta) + \nu_2 q(x + \delta) < 0 \\
\Longleftrightarrow & p(x) \le -(\nu_1 + C_s \nu_2) p(x + \delta) \\
\Longleftrightarrow & p(x + \delta) \ge -\frac{1}{\nu_1 + C_s \nu_2} p(x) \\
\Longleftrightarrow & \phi_s(\|x + \delta\|_2) \ge -\frac{1}{\nu_1 + C_s \nu_2} \phi_s(\|x\|_2) \\
\Longleftrightarrow & \|x + \delta\|_2 \le \phi_s^{-1}(-\frac{1}{\nu_1 + C_s \nu_2} \phi_s(\|x\|_2)).
\end{aligned} \tag{101}$$

Injecting $\|x\|_2 = \sigma_s(2u)^{\frac{1}{\eta}}$ into inequalities above, we get

$$-\frac{1}{\nu_1 + C_s \nu_2} p(x) \in (0, U] \Longleftrightarrow u + \ln(-(\nu_1 + C_s \nu_2)) \ge 0, \tag{102}$$

which is the boundary condition for $\omega_2$. Though looks similar, it differs significantly from $\omega_1$. Let $-\frac{1}{\nu_1+C_s\nu_2}\phi_s(\|x\|_2) \in (U, +\infty]$, then

$$p(x) + \nu_1 p(x+\delta) + \nu_2 q(x+\delta) < 0 \Longleftrightarrow \phi_s(\|x+\delta\|_2) \geq U, \tag{103}$$

which means $\mathbb{P}\{p(x) + \nu_1 p(x+\delta) + \nu_2 q(x+\delta) < 0 \wedge q(x+\delta) > 0| \ \|x\|_2 = \sigma_g(2u)^{\frac{1}{\eta}}\} = 0$ under $u + \ln(-(\nu_1 + C_s\nu_2)) < 0$. Finally, we have

$$\omega_2(u) = \begin{cases} \Psi_{\frac{d-1}{2}}\left( \dfrac{\min\{T^2, \left(\phi_s^{-1}(-\frac{1}{\nu_1+C_s\nu_2}\phi_s(\sigma_s(2u)^{\frac{1}{\eta}}))\right)^2\} - (\sigma_s(2u)^{\frac{1}{\eta}} - \rho)^2}{4\rho\sigma_s(2u)^{\frac{1}{\eta}}} \right) & , u + \ln(-\nu_1 - C_s\nu_2) \geq 0, \\ \\ 0 & , u + \ln(-\nu_1 - C_s\nu_2) < 0. \end{cases} \tag{104}$$

(3) Function $\omega_3$.

The computational logic of $\omega_3$ is similar to that of $\omega_2$. For $\mathbb{P}\{p(x) + \nu_1 p(x+\delta) + \nu_2 q(x+\delta) < 0 \wedge q(x+\delta) = 0| \ \|x\|_2 = \sigma_g(2u)^{\frac{1}{\eta}}\}$. When $-\frac{1}{\nu_1}p(x) \in (0, U]$, we have

$$\begin{aligned} &p(x) + \nu_1 p(x+\delta) + \nu_2 q(x+\delta) < 0 \\ \Longleftrightarrow &p(x) \leq -\nu_1 p(x+\delta) \\ \Longleftrightarrow &p(x+\delta) \geq -\frac{1}{\nu_1}p(x) \\ \Longleftrightarrow &\phi_s(\|x+\delta\|_2) \geq -\frac{1}{\nu_1}\phi_s(\|x\|_2) \\ \Longleftrightarrow &\|x+\delta\|_2 \leq \phi_s^{-1}(-\frac{1}{\nu_1}\phi_s(\|x\|_2)). \end{aligned} \tag{105}$$

When $\|x\|_2 = \sigma_s(2u)^{\frac{1}{\eta}}$, we obtain

$$-\frac{1}{\nu_1}\phi_s(\|x\|_2) \in (0, U] \Longleftrightarrow u + \ln(-\nu_1) \geq 0. \tag{106}$$

Thus we have the expression

$$\omega_3(u) = \begin{cases} \max\left\{ \Psi_{\frac{d-1}{2}}\left( \dfrac{\left(\phi_s^{-1}(-\frac{1}{\nu_1}\phi_s(\sigma_s(2u)^{\frac{1}{\eta}}))\right)^2 - (\sigma_s(2u)^{\frac{1}{\eta}} - \rho)^2}{4\rho\sigma_s(2u)^{\frac{1}{\eta}}} \right) - \Psi_{\frac{d-1}{2}}\left( \dfrac{T^2 - (\sigma_s(2u)^{\frac{1}{\eta}} - \rho)^2}{4\rho\sigma_s(2u)^{\frac{1}{\eta}}} \right), 0 \right\}, \\ \hspace{10cm} u + \ln(-\nu_1) \geq 0, \\ 0, u + \ln(-\nu_1) < 0. \end{cases} \tag{107}$$

**Calculation of $\phi_s^{-1}(r)$.** Appearing in all $\omega$ functions in Theorem 3, $\phi_s^{-1}(r)$ is an indispensable value in the system. We let $\xi$ be $\phi_s^{-1}(-\nu\phi_s(\sigma_s(2u)^{\frac{1}{\eta}})$ when $-\nu\phi_s(\sigma_s(2u)^{\frac{1}{\eta}}) \in (0, U]$, then

$$\phi_s(\xi) = -\nu\phi_s(\sigma_s(2u)^{\frac{1}{\eta}}). \tag{108}$$

We thus have

$$\begin{aligned} &\frac{\eta}{2}\frac{1}{(2\sigma_s^\eta)^{\frac{d}{\eta}}\pi^{\frac{d}{2}}}\frac{\Gamma(\frac{d}{2})}{\Gamma(\frac{d}{\eta})}\exp(-\frac{\xi^\eta}{2\sigma_s^\eta}) = -\frac{\nu\eta}{2}\frac{1}{(2\sigma_s^\eta)^{\frac{d}{\eta}}\pi^{\frac{d}{2}}}\frac{\Gamma(\frac{d}{2})}{\Gamma(\frac{d}{\eta})}\exp(-u) \\ \Longleftrightarrow &\exp(-\frac{1}{2}(\frac{\xi}{\sigma_s})^\eta) = -\nu\exp(-u). \end{aligned} \tag{109}$$

Obviously $\xi \geq 0$, then $\exp(-\frac{1}{2}(\frac{\xi}{\sigma_s})^\eta) \in (0, 1]$. When Equation (109) has a solution, we need

$$0 < -\nu\exp(-u) \leq 1. \tag{110}$$

Therefore, we get the boundary condition

$$\nu < 0, u \geq \ln(-\nu). \tag{111}$$

Under Equation (111), Equation (109) can be solved:

$$\xi = 2^{\frac{1}{\eta}}\sigma_s(u - \ln(-\nu))^{\frac{1}{\eta}}. \tag{112}$$

We thus have:

$$
\omega_1(u, \nu) = \begin{cases} \Psi_{\frac{d-1}{2}} \left( \dfrac{(\rho + \sigma_s(2u)^{\frac{1}{\eta}})^2 - 2^{\frac{2}{\eta}} \sigma_s^2(u - \ln(-\nu))^{\frac{2}{\eta}}}{4\rho\sigma_s(2u)^{\frac{1}{\eta}}} \right) & , u - \ln(-\nu) \geq 0, \\ 1 & , u - \ln(-\nu) < 0, \end{cases}
$$

$$
\omega_2(u) = \begin{cases} \Psi_{\frac{d-1}{2}} \left( \dfrac{\min\{T^2, 2^{\frac{2}{\eta}} \sigma_s^2(u + \ln(-\nu_1 - C_s\nu_2))^{\frac{2}{\eta}}\} - (\sigma_s(2u)^{\frac{1}{\eta}} - \rho)^2}{4\rho\sigma_s(2u)^{\frac{1}{\eta}}} \right) & , u + \ln(-\nu_1 - C_s\nu_2) \geq 0, \\ 0 & , u + \ln(-\nu_1 - C_s\nu_2) < 0, \end{cases}
$$

$$
\omega_3(u) = \begin{cases} \max \left\{ \Psi_{\frac{d-1}{2}} \left( \dfrac{2^{\frac{2}{\eta}} \sigma_s^2(u + \ln(-\nu_1))^{\frac{2}{\eta}} - (\sigma_s(2u)^{\frac{1}{\eta}} - \rho)^2}{4\rho\sigma_s(2u)^{\frac{1}{\eta}}} \right) - \Psi_{\frac{d-1}{2}} \left( \dfrac{T^2 - (\sigma_s(2u)^{\frac{1}{\eta}} - \rho)^2}{4\rho\sigma_s(2u)^{\frac{1}{\eta}}} \right), 0 \right\}, \\ \hfill u + \ln(-\nu_1) \geq 0, \\ 0, u + \ln(-\nu_1) < 0. \end{cases}
$$

$$(113)$$

By plugging $\omega$ functions into expressions of expectations in Theorem 3, we obtain the solution to Problem (9) for ESG, which completes the illustration on differences between Theorem 2 and Theorem 3.

Besides solving the dual problem for EGG and ESG, the analysis of boundary conditions is needed frequently in the conservative algorithm (Figure 3). Even though subtle differences may exist in different settings, the core technique is almost the same as the derivation above.

## F  SUPPLEMENTARY PRELIMINARIES

We offer further preliminaries in this section to introduce the backgrounds of the Neyman-Pearson certification and the DSRS certification.

### F.1  NEYMAN-PEARSON CERTIFICATION

The primitive DSRS framework takes Neyman-Pearson certification as the baseline. We follow this setting in this work and introduce it here. With the same notations of Problem (9), the primal problem of NP certification can be formulated as:

$$
\min_{\tilde{f}_{x_0} \in \mathcal{F}} \quad \mathbb{E}_{z \sim \mathcal{P}} \left( \tilde{f}_{x_0}(\delta + z) \right),
$$
$$
\text{s.t.} \quad \mathbb{E}_{z \sim \mathcal{P}} \left( \tilde{f}_{x_0}(z) \right) = A. \tag{114}
$$

Same as DSRS, instead of solving Equation (114), we solve its strong dual problem, which can be easily solved by level set method and numerical integration:

$$
\max_{\nu \in \mathbb{R}} \quad \mathbb{P}_{z \sim \mathcal{P} + \delta}\{p(z - \delta) + \nu p(z) < 0\},
$$
$$
\text{s.t.} \quad \mathbb{P}_{z \sim \mathcal{P}}\{p(z - \delta) + \nu p(z) < 0\} = A. \tag{115}
$$

In practice, we compute DSRS-certified radius by sequential searching taking NP-certified radius as the starting point every time. To show the growth of certified robustness by using additional distribution in the DSRS method relative to the NP method, we define increment (Growth):

$$
INC \triangleq \frac{ACR_{DS} - ACR_{NP}}{ACR_{NP}} \times 100\%, \tag{116}
$$

where $ACR_{DS}$ and $ACR_{NP}$ are the $ACR$ for the base classifier certified by DSRS and NP, respectively. We show abundant results on NP certification in Appendix J. On real-world datasets, our observations on DSRS that certified robustness keeps almost invariant w.r.t. $\eta$ for ESG and monotonically increases w.r.t. $\eta$ for EGG also hold for the NP certification. From these results, we predict that for the NP certification, most ESG distributions have the potential to give the best results for $\ell_2$ certified radius.

### F.2  SOLVING PROBLEM (4) AND PROBLEM (9)

To better clarify the mechanisms behind the optimizaiton problems in this paper, here we provide a concise introduction to them. One essential contribution of the randomized smoothing framework

(Cohen et al., 2019) was that it discovered an extremely simple connection $f(\rho)$ between certified radius $\rho$ and accuracy $A$ of the smoothed classifier. Taking Gaussian distribution as the noise, they derived

$$\Phi(\frac{\rho}{\sigma}) = A, \tag{117}$$

by Neyman-Pearson lemma, where $\sigma$ is the variance of Gaussian and $\Phi(\cdot)$ is the CDF of Gaussian. However, the simple mapping between $r$ and $A$ does not universally present in other distributions. Taking EGG as the example, we have

$$\rho = \max \quad r, \tag{118a}$$

$$\text{s.t.} \quad \mathbb{E}_{u \sim \Gamma(\frac{d-2k}{\eta}, 1)} \Psi_{\frac{d-1}{2}} \left( \frac{\sigma_g^2 (\frac{4kW(\frac{\eta u}{2k} \chi^{\frac{\eta}{2k}} \exp(\frac{\eta u}{2k}))}{\eta})^{\frac{2}{\eta}} - (\sigma_g(2u)^{\frac{1}{\eta}} - r)^2}{4r\sigma_g(2u)^{\frac{1}{\eta}}} \right) > \frac{1}{2}, \tag{118b}$$

$$\mathbb{E}_{u \sim \Gamma(\frac{d-2k}{\eta}, 1)} \Psi_{\frac{d-1}{2}} \left( \frac{(r + \sigma_g(2u)^{\frac{1}{\eta}})^2 - \left( \frac{4kW(\frac{\eta u}{2k}(-\chi)^{-\frac{\eta}{2k}} \exp(\frac{\eta u}{2k}))\sigma_g^{\eta}}{\eta} \right)^{\frac{2}{\eta}}}{4r\sigma_g(2u)^{\frac{1}{\eta}}} \right) = A. \tag{118c}$$

Same as the setting in Cohen et al. (2019), $A$ is known. To solve the problem, we firstly perform a binary search on $\chi$, then substitute $\chi$ into Equation (118b), and perform another binary search on $r$. Finally, the maximum $r$ satisfying the subections is the certified radius we need. Through solving Equation (118), we learn that for some distributions, it is available to calculate the certified radius $\rho$ only depending on $A$, but the process can be much more complicated than that of Gaussian.

The DSRS framework shares the identical thinking of Equation (118), where the only difference is that DSRS introduces another subjection. To solve the DSRS problem, we only need to perform another binary search to obtain both $\nu_1$ and $\nu_2$. In other words, DSRS has two intermediate variables like $\chi$, which can be solved by performing two binary searches. Finally, since DSRS has a counterpart of Equation (118b), by substituting both the intermediate variables into the counterpart equation, and performing a binary search on $r$ can finally get the certified radius of DSRS. Given its complexity, we refer the reader to Appendix E and Appendix G of (Li et al., 2022) for further details on solving the DSRS problem.

# G   SUPPLEMENTARY FOR EXPERIMENTAL METHODS

## G.1   BASE CLASSIFIERS

To see the effects of $\eta$ clearly, we take pre-trained models in our real-world datasets experiments. Concretely, we use Gaussian-augmented models from Cohen et al. (2019) and General-Gaussian-augmented models from Li et al. (2022), including Consistency and SmoothMix models. For the convenience of comparison, we fix the base classifier for each group of experiments, and all the sampling distributions keep $\mathbb{E}r^2$ the same with the base classifier following the setting of the previous work (Yang et al., 2020; Li et al., 2022).

## G.2   CERTIFICATION DETAILS

**Hyperparameter settings.** In double-sampling process, the numbers of sampling $N_1, N_2$ are 50000, and significance levels $\alpha_1, \alpha_2$ are 0.0005 for Monte Carlo sampling, equal for $\mathcal{P}$ and $\mathcal{Q}$. For CIFAR-10 and ImageNet, we set $k = 1530$ and $k = 75260$, respectively, in consistent with base classifiers. The threshold parameter for $\mathcal{Q}$ is determined by a heuristic algorithm from Li et al. (2022). Specifically, we set

$$T_{\mathcal{S}} = \sigma_s (2\Lambda_{\frac{d}{\eta}}^{-1}(\kappa))^{\frac{1}{\eta}} = \sigma \sqrt{\frac{d\Gamma(\frac{d}{\eta})}{\Gamma(\frac{d+2}{\eta})}} (\Lambda_{\frac{d}{\eta}}^{-1}(\kappa))^{\frac{1}{\eta}},$$

$$T_{\mathcal{G}} = \sigma_g (2\Lambda_{\frac{d-2k}{\eta}}^{-1}(\kappa))^{\frac{1}{\eta}} = \sigma \sqrt{\frac{d\Gamma(\frac{d-2k}{\eta})}{\Gamma(\frac{d-2k+2}{\eta})}} (\Lambda_{\frac{d-2k}{\eta}}^{-1}(\kappa))^{\frac{1}{\eta}}, \tag{119}$$

where $\kappa$ is determined by the heuristic algorithm (Li et al., 2022), a simple function of Monte Carlo sampling probability from $\mathcal{P}$.

We choose NP certification as the baseline since it is the state-of-the-art method for single distribution certification. The sampling distribution for the NP method is $\mathcal{P} = \mathcal{S}(\sigma, \eta)$ for the ESG distribution, and $\mathcal{P} = \mathcal{G}(\sigma, \eta, k)$ for the EGG distribution. For fairness, the sampling number $N$ is set to 100000, with the significance level $\alpha = 0.001$ for Monte Carlo sampling. The setting for exponents $\eta$ in the NP method is the same as DSRS. We notice when $k = 0$, EGG distributions become their corresponding ESG distributions. The ESG distribution degenerates to the Standard Gaussian distribution when $\eta = 2$. We do not combine the cases for EGG and ESG since $k$ appears as the denominator in some equations.

We name the base classifiers by {training method}-{distribution}-{substitution variance} in experimental results for ACR. For example, model StdAug-GGS-1.00 denotes the base classifier is trained using Standard General Gaussian augmentation with $\sigma = 1.00$. In all the experiments, we set training method $\in$ {StdAug (i.e., standard augmentation), Consistency, SmoothMix}, distribution $\in$ {GS (i.e., Gaussian), GGS (i.e., General Gaussian)} and $\sigma \in \{0.25, 0.50, 1.00\}$. See details of the pre-trained models in Appendix G.1.

Overall, we provide Algorithm 2 as follows, where the error bound for certified radius $e$ is set $1 \times 10^{-6}$. We refer to readers to Algortihm 1, Algorithm 3 in Li et al. (2022) for the conservative algorithm $C$ and DualBinarySearch algorithm $D$. for Our code for experiments is modified from Li et al. (2022).

---

**Algorithm 2:** Standard algorithm for Double Sampling Randomized Smoothing by Exponential General Gaussian (EGG) distributions on real-world datasets

---

**Input:** base classifier $f$, substitution variance $\sigma$, exponent $\eta$, hyperparameter $k$, significance
       level $\alpha$, error bound for certified radius $e$, heuristic algorithm $H$, conservative algorithm
       $C$, DualBinarySearch algorithm $D$ (Algorithms $H, C, D$ from Li et al. (2022)).

1   Initialize the noise distribution as EGG: $\mathcal{P} = \mathcal{G}(\sigma, \eta, k)$
2   $A_1 \leftarrow$ SampleUnderNoise $(f, \mathcal{P}, \alpha)$
3      $\triangleright A_1$ is the Clopper-Pearson lower bound for the sampling result (Clopper & Pearson, 1934; Cohen et al., 2019)
4   $T \leftarrow H(A_1)$                    $\triangleright$ Threshhold $T$ is the hyperparameter for truncated distribution $\mathcal{Q}$
5   Initialize the supplementary distribution as TEGG: $\mathcal{Q} = \mathcal{G}_t(\sigma, \eta, k, T)$
6   $B_1 \leftarrow$ SampleUnderNoise $(f, \mathcal{Q}, \alpha)$
7   $A, B \leftarrow C(A_1, B_1)$          $\triangleright A, B$ for Problem (9) is determined by a conservative algorithm
8   $r_l \leftarrow 0, r_r \leftarrow I$          $\triangleright$ Initialization for the binary search on $r$, where $I$ is a big number
9   **while** $r_r - r_l > e$ **do**
10     |   $r_m \leftarrow (r_r + r_l)/2$
11     |   $p_m \leftarrow D(A, B)$               $\triangleright$ Problem (9) can be solved by $D$ by given $A, B$
12     |   **if** $p_m > 1/2$ **then**
13     |     |   $r_l \leftarrow r_m$
14     |   **else**
15     |     |   $r_r \leftarrow r_m$
16     |   **end if**
17   **end while**
**Output:** certified radius $r_l$

---

### G.3 COMPUTATIONAL OVERHEAD

All of our experiments on real-world datasets are composed of sampling and certification which are finished with 4 NVIDIA 3080 GPUs and CPUs. The most computationally intensive procedure is sampling. For $\sigma = 0.50$ base classifiers, it takes about 5s, 200s to sample 50000 times under noise distributions on CIFAR10, ImageNet with one GPU. Given that we uniformly pick 1000 data points from each dataset, one sampling procedure takes around 1 hour, 1 day for 50000 noises on CIFAR10, ImageNet respectively with one GPU. Naturally, the sampling time almost doubles for 100000 noises.

The computation for certification only relies on CPUs. Running time for certification is basically constant for different datasets, including NP certification and DSRS certification. Usually, it takes 1-2 days to complete. The overall computational time for standard DSRS certification is strictly larger than the pure NP certification, with respect to a specific number of samples. For instance, if we compute a certificate for 100000 noises, we need one NP certification under the pure NP certification, while we need one NP certification for 50000 noises and one DSRS certification for the remaining 50000 noises under the DSRS certification. Generally, the computational time for standard DSRS certification is one to two times that of pure NP certification.

### G.4 TRANSFERABILITY OF THE INCREMENTAL EFFECT

We show results for real-world datasets on different base classifiers in this section. In Table 6, we observe $\eta = 8.0$ shows better performance than General Gaussian ($\eta$=2.0) overall. This alludes to some defects in the current training method for General Gaussian because intuitively the model trained by General Gaussian should have provided the best certified results among all $\eta$, but we find the incremental effect with $\eta$ still exists in General-Gaussian-augmented models.

Table 6: The incremental effect with $\eta$ on different base classifiers.

| Dataset | Model | $\eta$ | Certified Accuracy at $r$ | | | | | | | | | | $ACR$ |
|---------|-------|--------|------|------|------|------|------|------|------|------|------|------|------|
| | | | 0.15 | 0.30 | 0.45 | 0.60 | 0.75 | 0.90 | 1.05 | 1.20 | 1.35 | 1.50 | |
| CIFAR-10 | StdAug-GS-0.50 | 2.0((Li et al., 2022), the SOTA) | 56.9% | 48.0% | 40.9% | 33.5% | 26.4% | 19.5% | 13.8% | 10.6% | 7.4% | 3.3% | 0.437 |
| | | 8.0 | 58.1% | 50.8% | 42.8% | 35.9% | 30.6% | 23.2% | 18.1% | 13.7% | 10.1% | 7.3% | 0.489 |
| | StdAug-GGS-0.50 | 2.0((Li et al., 2022), the SOTA) | 58.3% | 51.7% | 44.5% | 38.1% | 29.3% | 22.7% | 17.7% | 13.1% | 8.3% | 3.8% | 0.480 |
| | | 8.0 | 58.2% | 52.1% | 44.4% | 38.8% | 30.9% | 24.1% | 19.1% | 14.4% | 10.8% | 6.8% | 0.502 |
| | Consistency-GGS-0.50 | 2.0((Li et al., 2022), the SOTA) | 52.0% | 48.8% | 44.7% | 42.1% | 38.5% | 36.0% | 32.9% | 28.6% | 24.0% | 19.6% | 0.618 |
| | | 8.0 | 51.7% | 48.6% | 44.7% | 42.2% | 38.8% | 36.1% | 34.1% | 29.9% | 26.8% | 22.3% | 0.650 |
| | SmoothMix-GGS-0.50 | 2.0((Li et al., 2022), the SOTA) | 55.8% | 52.3% | 49.1% | 45.3% | 41.7% | 37.7% | 34.6% | 30.0% | 26.3% | 21.2% | 0.662 |
| | | 8.0 | 55.5% | 52.1% | 49.1% | 45.6% | 42.0% | 38.2% | 35.2% | 31.7% | 27.8% | 24.3% | 0.695 |
| ImageNet | StdAug-GGS-0.50 | 2.0((Li et al., 2022), the SOTA) | 56.7% | 52.8% | 49.4% | 45.7% | 41.5% | 38.1% | 34.0% | 30.6% | 25.4% | 19.4% | 0.654 |
| | | 8.0 | 56.7% | 52.9% | 49.8% | 46.3% | 42.9% | 39.3% | 35.8% | 32.7% | 29.1% | 24.6% | 0.703 |

### G.5 INTEGRATION METHOD FOR ESG DISTRIBUTIONS

The `scipy` package loses precision when calculating integrals for the $\Gamma(\alpha, 1)$ distribution with large parameters (say, $\alpha > 500$ ) on infinite intervals. To solve this problem, we propose a Linear Numerical Integration (LNI) method to compute the expectations fast and accurately. LNI is based on Lemma 5.1, here we provide the proof for it:

Lemma 5.1 demonstrates the mass of the gamma distribution highly concentrates to its expectation for large $d$ and small $\eta$. With this great property, let $\iota$ be a small positive number, we can compute the integral for the gamma distribution by considering $1 - \iota$ total mass. We find the most primitive method that uniformly segments the integration interval provides good precision for certifications on CIFAR-10 and ImageNet. In our experiments for ESG, we set the number of segments to 256, and $\iota = 10^{-4}$. We illustrate the effect of the segment number on CIFAR-10 and ImageNet in Figure 6.

### G.6 ERROR ESTIMATION FOR NUMERICAL INTEGRATION

To evaluate the accuracy of numerical simulation for our work, we follow the method from Yang et al. (2020), which calculates relative errors for the certified radius obtained by numerical integration (NI) and Monte Carlo (MC) simulation respectively. In our experiments, we sample 100000 times for Monte Carlo simulations. For each combination of $\sigma$ and dataset, we test 1000 probability lower bounds $p_l$ that are uniformly sampled from the interval $(0.5, 1)$. See details in Table 7 and Figure 7-8. Overall, the results for numerical integration of ESG have lower error levels than that of EGG.

## H SUPPLEMENTARY FOR NUMERICAL SIMULATION

We also conduct a numerical simulation to explore the effects of EGG distributions on currently unattainable $A, B$ pairs from Problem (9). We only show simulative experiments for the EGG distribution since we observe great monotonicity for certified robustness (and/or certified radius) w.r.t. $\eta$ from EGG, which does not occur in ESG. We consider two cases: $B = 1$ and $B < 1$ for $B$ in Problem (9). See details in Appendix H.

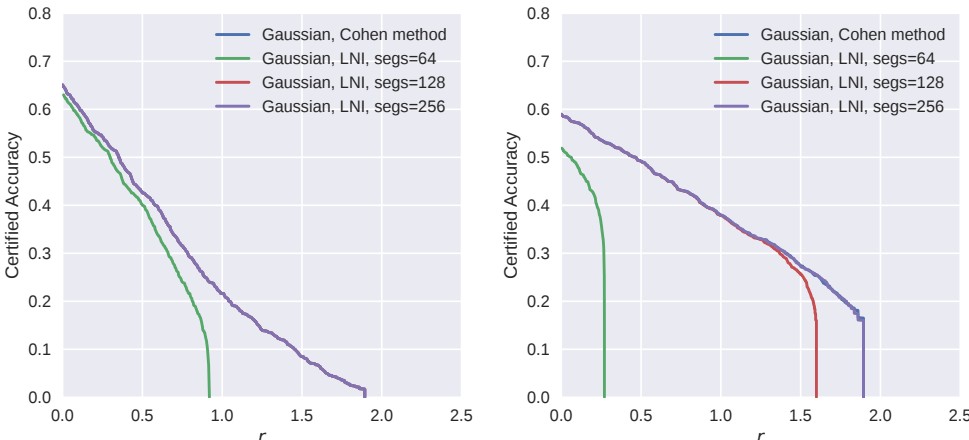

Figure 6: Results for the $\ell_2$ certified radius under different numbers of segments setting of LNI, $\iota$ is set to $10^{-4}$. **Left:** on StdAug-GGS-0.50 (CIFAR-10). The curves of StdAug method and segs = 128, 256 are overlapped, showing segs $\geq 128$ is enough for CIFAR-10. **Right:** on StdAug-GGS-0.50 (ImageNet). The curves of Cohen's method and segs = 256 are almost overlapped. Since the error here is tolerable for both CIFAR-10 and ImageNet, we pick 256 as the number of segments in all the ESG experiments.

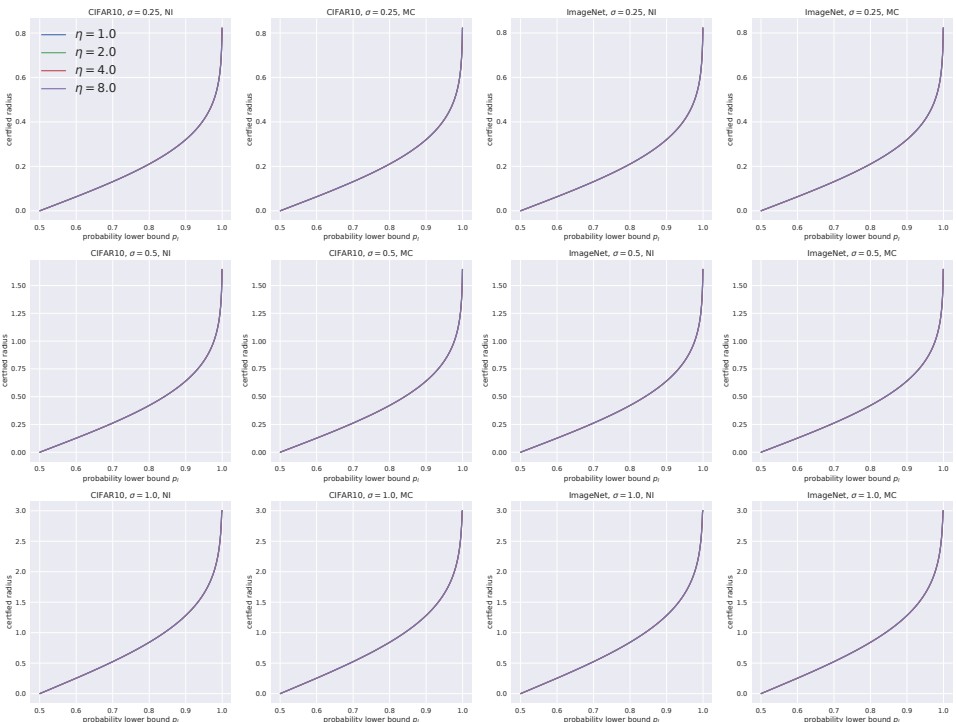

Figure 7: Comparisons of numerical integration and Monte Carlo simulation for ESG distributions.

**Results and analyses.** (1) $B = 1$. This is actually an ideal case since it is impossible to train such a base classifier. Here we still present the results, as it demonstrates the theoretical performance of EGG distributions. From Figure 1a, we observe there is a monotonically decreasing tendency in the certified radius w.r.t. $\eta$, which seems contradictory to the results on real-world datasets. This is partly because the concentration assumption (i.e., $B = 1$) is more friendly to the smaller $\eta$. In fact, the major mass of the EGG distribution with smaller $\eta$ gathers near 0. This makes smaller

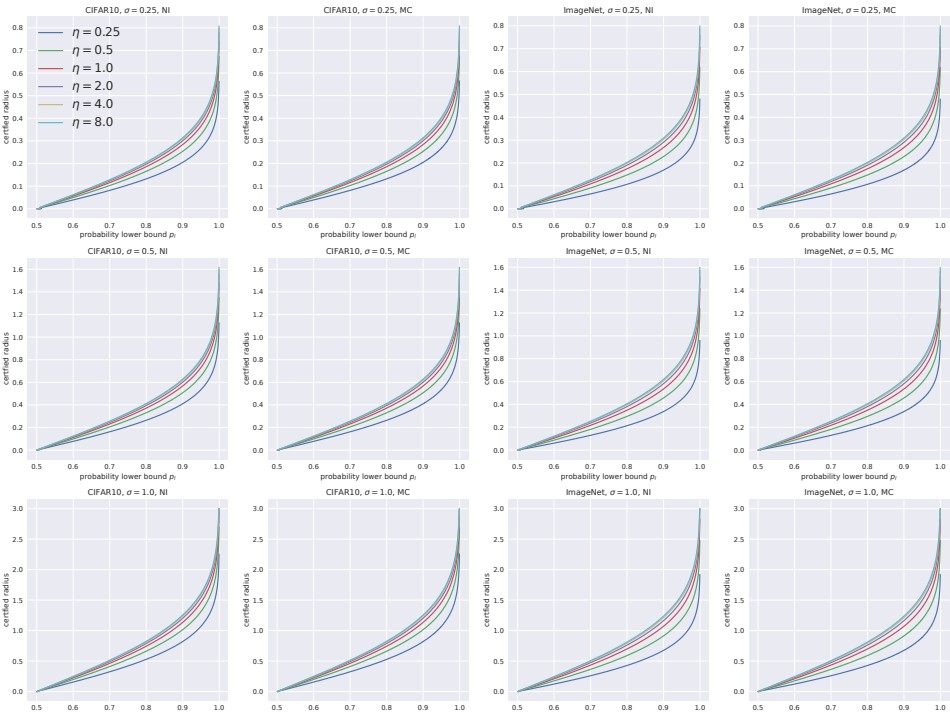

Figure 8: Comparisons of numerical integration and Monte Carlo simulation for EGG distributions.

Table 7: Maximum point-to-point error w.r.t. $\eta$ for numerical integration.

| dataset | $\sigma$ | $\eta$ (ESG) | | | | $\eta$ (EGG) | | | | | |
|---------|----------|------|------|------|------|------|------|------|------|------|------|
| | | 1.0 | 2.0 | 4.0 | 8.0 | 0.25 | 0.5 | 1.0 | 2.0 | 4.0 | 8.0 |
| | 0.25 | 3.4e-5 | 2.5e-5 | 5.6e-5 | 1.0e-4 | 1.7e-3 | 1.6e-3 | 1.0e-3 | 5.2e-4 | 4.7e-4 | 3.7e-4 |
| CIFAR10 | 0.5 | 5.0e-6 | 5.0e-6 | 9.3e-6 | 1.1e-5 | 1.9e-3 | 1.0e-3 | 9.6e-4 | 9.1e-4 | 1.0e-3 | 6.7e-4 |
| | 1.0 | 8.9e-5 | 7.0e-5 | 1.2e-4 | 1.9e-4 | 2.5e-3 | 2.2e-3 | 2.2e-3 | 1.3e-3 | 1.0e-3 | 1.1e-3 |
| | 0.25 | 1.2e-5 | 7.9e-6 | 1.5e-5 | 2.2e-5 | 2.5e-3 | 3.0e-3 | 2.6e-3 | 1.6e-3 | 1.3e-3 | 7.8e-4 |
| ImageNet | 0.5 | 1.5e-4 | 1.3e-4 | 1.9e-4 | 3.4e-4 | 4.6e-3 | 7.4e-3 | 2.9e-3 | 3.4e-3 | 1.8e-3 | 1.8e-3 |
| | 1.0 | 2.4e-5 | 1.5e-5 | 3.1e-5 | 4.8e-5 | 6.5e-3 | 5.1e-3 | 3.2e-3 | 2.5e-3 | 2.1e-3 | 2.0e-3 |

$\eta$ more sensitive to relaxation of concentration assumption (see Figure 10), which leads to worse performance on real-world datasets compared to larger $\eta$. (2) $B < 1$. Shown in Table 8, the monotonically increasing tendency w.r.t. $\eta$ continues to exist, manifesting the essential superiority of large $\eta$ EGG distribution in the DSRS framework. We also notice that the increase is not endless, since $\eta = 64.0$ shows a marginal increment to that of $\eta = 32.0$ (see also Figure 1b). Though hard to prove, this may imply some convergence due to the extremely slow growth. We do not show results for larger $\eta$ due to floating-point limitations and low necessity based on our observation. For all the distributions used for numerical simulation, we set $\sigma = 1.0$ and $k = \frac{d}{2} - 5$ for fair comparison.

## H.1  $B = 1$ SETTINGS

We need to modify Theorem 2 slightly to get certified results. We see

$$B = 1 \iff C_g \mathbb{E}_{u \sim \Gamma(\frac{d-2k}{\eta}, 1)} \omega_1(u, \nu_1 + C_g \nu_2) \cdot \mathbb{1}_{u \le \frac{T\eta}{2\sigma_g^\eta}} = 1 \tag{120}$$

$$\iff \nu_1 + C_g \nu_2 \to -\infty,$$

which makes

$$\omega_2(u) = \Psi_{\frac{d-1}{2}}\left(\frac{T^2 - (t-\rho)^2}{4\rho t}\right). \tag{121}$$

Injecting $\omega_2(u)$ into Theorem 2, we can compute the certified radius as general cases. In $B = 1$ experiments, $T$ is set to $\sqrt{2\Lambda_{\frac{d}{2}}(0.5)}$ due to the assumption of $(1, 0.5, 2)$-concentration property.

Table 8: Numerical simulation for EGG distributions (metric: certified radius).

| $\eta$ | A | 0.6 | | | | 0.7 | | | | 0.8 | | |
| --- | --- | --- | --- | --- | --- | --- | --- | --- | --- | --- | --- | --- |
| | B | 0.6 | 0.7 | 0.8 | 0.9 | 0.6 | 0.7 | 0.8 | 0.9 | 0.7 | 0.8 | 0.9 |
| 0.5 | | 0.188 | 0.194 | 0.216 | 0.273 | 0.408 | 0.391 | 0.407 | 0.471 | 0.678 | 0.632 | 0.675 |
| 1.0 | | 0.218 | 0.225 | 0.251 | 0.320 | 0.471 | 0.451 | 0.470 | 0.546 | 0.778 | 0.726 | 0.776 |
| 2.0 | | 0.234 | 0.242 | 0.271 | 0.346 | 0.506 | 0.485 | 0.505 | 0.589 | 0.836 | 0.779 | 0.833 |
| 4.0 | | 0.243 | 0.251 | 0.281 | 0.360 | 0.525 | 0.502 | 0.524 | 0.611 | 0.867 | 0.807 | 0.863 |
| 8.0 | | 0.247 | 0.255 | 0.286 | 0.367 | 0.534 | 0.511 | 0.533 | 0.622 | 0.882 | 0.821 | 0.878 |
| 16.0 | | 0.249 | 0.257 | 0.288 | 0.370 | 0.538 | 0.515 | 0.537 | 0.627 | 0.889 | 0.827 | 0.885 |
| 32.0 | | 0.250 | 0.258 | 0.289 | 0.371 | 0.540 | 0.517 | 0.539 | 0.629 | 0.893 | 0.830 | 0.887 |
| 64.0 | | 0.250 | 0.258 | 0.290 | 0.371 | 0.541 | 0.518 | 0.539 | 0.629 | 0.894 | 0.831 | 0.888 |
| Increase (64.0 to 2.0) | | 6.8% | 6.6% | 7.0% | 7.2% | 6.9% | 6.8% | 6.7% | 6.8% | 6.9% | 6.7% | 6.6% |

## H.2 $B < 1$ SETTINGS

. Theorem 2 can be directly used in computing the certified radius for this case. The values set for $A$ and $B$ are not completely random as there is an inherent constraint (Li et al., 2022):

$$\begin{cases} \dfrac{B}{C_g} \leq A \leq 1 - \dfrac{1-B}{C_g}, \\ 0 \leq B \leq 1. \end{cases} \tag{122}$$

This constraint should also be considered when setting $A$ in case $B = 1$. For each EGG distribution, we set $C_g = 2$, meaning $T = \sigma_g (2\Lambda^{-1}_{\frac{d-2k}{\eta}}(0.5))^{\frac{1}{\eta}}$ for Figure 1b and Table 8.

## H.3 NUMERICAL SIMULATION RESULTS FOR OTHER DATASETS

The results in Table 8 almost do not change on large $d$. That is, our results that certified radius increases monotonically with $\eta$ is general for common datasets like CIFAR-10 and ImageNet. See Figure 9.

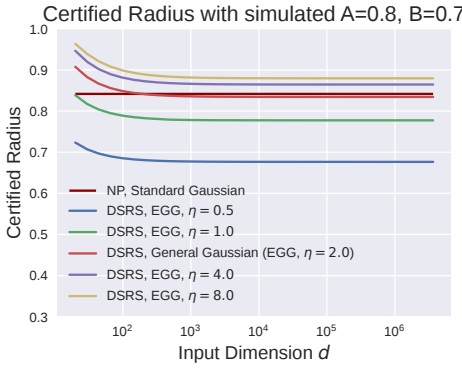

Figure 9: Certified radius vs. $d$, $A = 0.8, B = 0.7$.

## H.4 DIFFERENT SENSITIVITY TO RELAXATION FOR $\eta$

We observe contrary monotonicity for certified radius $w.r.t.$ $\eta$ under $B = 1$ and $B < 1$. A direct reason is small $\eta$ in EGG is more susceptible to relaxation of $B = 1$, see the following figure.

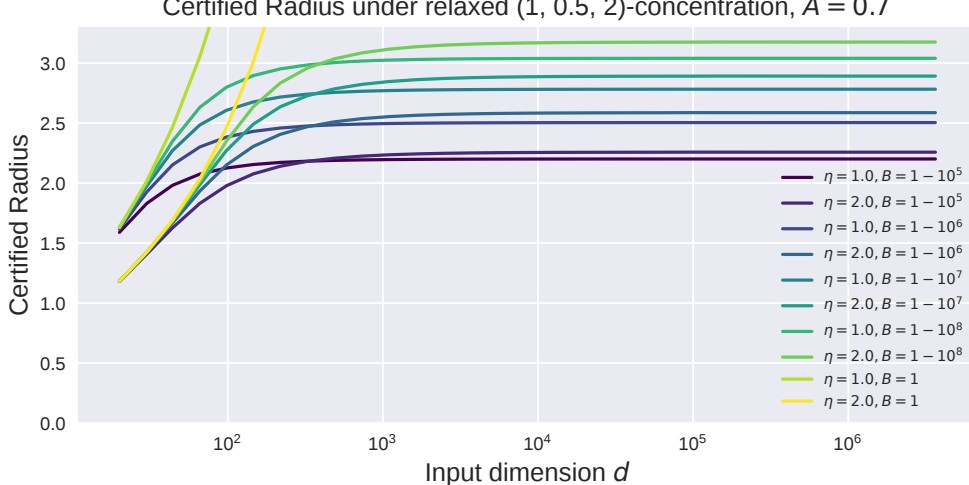

Figure 10: Certified radius with $\eta = 1$ vs. $\eta = 2$ under relaxed concentration assumption.

We set $T = \sqrt{2\Lambda_{\frac{d}{2}}(0.5)}$ for $B < 1$ curves in this figure for fair comparison with $B = 1$ ones. Figure 10 demonstrates that the smaller $\eta$ perform better when $B = 1$, while perform worse when $B < 1$ than the larger $\eta$. This indicates that the smaller $\eta$ suffer more from the relaxation of concentration assumption than the larger $\eta$.

## I    COMPARISON TO OTHER WORK

**Li et al. (2022).** Li et al. (2022)'s work is one of the cornerstones of this work. To the best of our knowledge, we are the first to generalize the DSRS framework systematically. Our theorems show in the DSRS framework, Exponential General Gaussian (EGG) distribution with exponent $\eta \in (0, 2)$ can provide tighter lower bounds than Li et al. 's results under concentration assumptions, and our experiments illustrates that EGG with exponent $\eta > 2$ provides better certification results than the General Gaussian distribution ($\eta$=2 in EGG) under practical settings (i.e., without concentration assumptions). Our proof for EGG removes some details from Li et al. (2022) 's, and adds other details such as the derivation for PDFs and for the Lambert W function. We also provide proof for ESG, which further uses the DSRS framework and contains nontrivial branches.

**Others.** For NP certification, though many attempts in the community have researched the inter-relationship between the smoothing distribution and certified radius, nobody has shown results as we do. Yang et al. (2020) trained models for each smoothing distribution, while we fixed the base classifier for the convenience of comparison. They also did not fully consider the distributions we use. Kumar et al. (2020) showed similar results to us for ESG in their Figure 5, but they only showed sampling results due to the lack of computing method, and they also trained base models for each distribution like Yang et al. (2020). Zhang et al. (2020) showed results for $\eta = 2$ EGG distribution under a fixed-model setting, but they didn't consider $\eta \neq 2$ cases.

## J    SUPPLEMENTARY FOR EXPERIMENTAL RESULTS

### J.1    CERTIFIED RADIUS AT $r$ FOR CONSISTENCY (JEONG & SHIN, 2020) AND SMOOTHMIX (JEONG ET AL., 2021) MODELS, MAXIMUM RESULTS

Table 9 and Table 10 shows the experimental results for certified accuracy at radius $r$. Each data is the **maximum** one among base classifiers with $\sigma \in \{0.25, 0.50, 1.00\}$. All the base classifiers in both the tables are trained under EGG distribution with $\eta = 2$ (the General Gaussian distribution used in DSRS (Li et al., 2022)), by Consistency (Jeong & Shin, 2020) and SmoothMix (Jeong et al., 2021) respectively. From the tables, we can see the rule observed on classifiers augmented

Table 9: Maximum certified accuracy w.r.t. $\sigma$, Consistency models

| Dataset | Method | Certified accuracy at $r$ | | | | | | | | | | | | | |
|---|---|---|---|---|---|---|---|---|---|---|---|---|---|---|---|
| | | 0.25 | 0.50 | 0.75 | 1.00 | 1.25 | 1.50 | 1.75 | 2.00 | 2.25 | 2.50 | 2.75 | 3.00 | 3.25 | 3.50 |
| CIFAR10 | EGG, $\eta = 1.0$ | 62.1% | 50.7% | 38.2% | 33.1% | 24.2% | 19.2% | 16.7% | 14.3% | 11.3% | 9.2% | 6.6% | 4.1% | 1.4% | 0.0% |
| | EGG, $\eta = 2.0$ | 62.5% | 52.0% | 38.5% | 34.4% | 27.4% | 20.6% | 17.0% | 14.7% | 12.7% | 10.5% | 8.5% | 6.3% | 3.9% | 2.5% |
| | EGG, $\eta = 4.0$ | 62.5% | 52.2% | 39.1% | 35.4% | 28.3% | 21.1% | 17.5% | 15.3% | 13.0% | 10.9% | 9.2% | 7.0% | 5.2% | 3.1% |
| | EGG, $\eta = 8.0$ | 62.5% | 52.6% | 40.4% | 35.3% | 28.6% | 22.3% | 17.6% | 15.5% | 13.2% | 11.3% | 9.6% | 7.8% | 5.8% | 3.9% |
| CIFAR10 | ESG, $\eta = 1.0$ | 62.6% | 52.9% | 41.6% | 35.5% | 29.3% | 23.7% | 17.7% | 15.8% | 13.7% | 11.8% | 10.0% | 8.8% | 6.8% | 4.6% |
| | ESG, $\eta = 2.0$ | 62.7% | 53.0% | 41.4% | 35.5% | 29.4% | 24.0% | 17.7% | 16.0% | 13.8% | 11.9% | 10.1% | 8.7% | 6.7% | 4.4% |
| | ESG, $\eta = 4.0$ | 62.7% | 52.9% | 41.6% | 35.5% | 29.5% | 23.8% | 17.8% | 15.7% | 13.9% | 11.9% | 10.2% | 8.8% | 6.7% | 4.8% |
| | ESG, $\eta = 8.0$ | 62.7% | 52.9% | 41.7% | 35.5% | 29.1% | 23.8% | 17.6% | 15.8% | 13.6% | 11.8% | 10.1% | 8.3% | 6.7% | 4.8% |

Table 10: Maximum certified accuracy w.r.t. $\sigma$, SmoothMix models

| Dataset | Method | Certified accuracy at $r$ | | | | | | | | | | | | | |
|---|---|---|---|---|---|---|---|---|---|---|---|---|---|---|---|
| | | 0.25 | 0.50 | 0.75 | 1.00 | 1.25 | 1.50 | 1.75 | 2.00 | 2.25 | 2.50 | 2.75 | 3.00 | 3.25 | 3.50 |
| CIFAR10 | EGG, $\eta = 1.0$ | 63.7% | 53.8% | 40.9% | 34.3% | 26.6% | 21.1% | 17.0% | 14.2% | 10.5% | 7.7% | 4.0% | 1.5% | 0.1% | 0.0% |
| | EGG, $\eta = 2.0$ | 64.5% | 55.0% | 41.7% | 35.6% | 28.9% | 21.3% | 18.0% | 15.2% | 12.3% | 9.7% | 6.4% | 3.7% | 1.4% | 0.4% |
| | EGG, $\eta = 4.0$ | 64.4% | 55.5% | 43.0% | 35.9% | 29.5% | 23.4% | 18.3% | 15.8% | 12.8% | 10.2% | 7.7% | 4.6% | 2.1% | 0.9% |
| | EGG, $\eta = 8.0$ | 64.7% | 55.7% | 43.9% | 36.2% | 30.1% | 24.3% | 18.6% | 15.8% | 13.2% | 10.6% | 8.0% | 5.4% | 2.7% | 1.3% |
| CIFAR10 | ESG, $\eta = 1.0$ | 64.6% | 56.5% | 46.5% | 36.6% | 31.3% | 25.6% | 19.1% | 16.2% | 13.4% | 11.4% | 8.9% | 6.3% | 4.0% | 1.8% |
| | ESG, $\eta = 2.0$ | 64.7% | 56.3% | 45.8% | 36.6% | 31.2% | 25.6% | 18.9% | 16.3% | 13.4% | 11.6% | 8.8% | 6.2% | 4.0% | 1.7% |
| | ESG, $\eta = 4.0$ | 64.6% | 56.3% | 46.0% | 36.5% | 31.0% | 25.6% | 19.0% | 16.2% | 13.6% | 11.5% | 9.2% | 6.0% | 3.9% | 1.8% |
| | ESG, $\eta = 8.0$ | 64.6% | 56.1% | 46.5% | 36.4% | 31.1% | 25.5% | 18.9% | 16.3% | 13.5% | 11.3% | 9.0% | 6.2% | 4.1% | 1.6% |

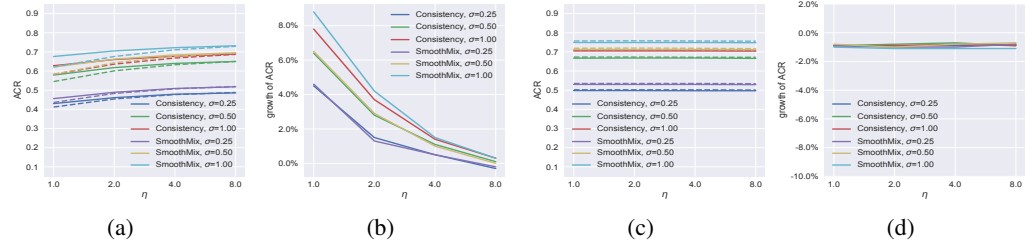

(a)      (b)      (c)      (d)

Figure 11: ACR results on Consistency and SmoothMix models. **(a).** ACR monotonically increases with $\eta$ in EGG. **(b).** The ACR growth gain by DSRS relative to NP shrinks with $\eta$ in EGG. **(c).** ACR keeps almost constant in ESG. **(d).** The ACR growth gain by DSRS remains almost constant in ESG. For (a) and (c), solid lines represent results from DSRS, and dotted lines represent results from NP.

standardly continues to exist, that certified accuracy increases monotonically with the $\eta$ of EGG, and keeps almost constant with the $\eta$ of ESG. We also show results for ACR in Table 11.

## J.2 FULL EXPERIMENTAL RESULTS FOR CERTIFICATIONS

We show full experimental results in this section, where the maximum results for certified accuracy in Table 2, Table 3, Table 9 and Table 10 are originated from. All the base classifiers in this section are trained under EGG distribution with $\eta = 2$ (the General Gaussian distribution used in DSRS (Li et al., 2022)). Our base classifiers are only affected by the dataset and the substitution variance. For example, in Table 11, all data under $\sigma = 0.25$ use the same base classifier, no matter what $\eta$ is. In addition, for $\sigma = 0.25$, the base classifiers for EGG and ESG are the same, which guarantees fair comparisons between distinctive distributions for certification.

Table 11: Full experimental results for certified accuracy, standard augmentation, EGG, CIFAR10

| $\sigma$ | $\eta$ | Certification method | 0.25 | 0.50 | 0.75 | 1.00 | 1.25 | 1.50 | 1.75 | 2.00 | 2.25 | 2.50 | 2.75 | 3.00 | 3.25 | 3.50 |
|---|---|---|---|---|---|---|---|---|---|---|---|---|---|---|---|---|
| 0.25 | 0.25 | NP | 41.4% | 0.5% | | | | | | | | | | | | |
| | | DSRS | 52.6% | 7.9% | | | | | | | | | | | | |
| | | (Growth) | 11.2% | 7.4% | | | | | | | | | | | | |
| | 0.5 | NP | 51.6% | 14.5% | 0.1% | | | | | | | | | | | |
| | | DSRS | 55.5% | 29.5% | | | | | | | | | | | | |
| | | (Growth) | 3.9% | 15.0% | | | | | | | | | | | | |
| | 1.0 | NP | 55.6% | 29.1% | 5.5% | | | | | | | | | | | |
| | | DSRS | 56.3% | 36.5% | 9.3% | | | | | | | | | | | |
| | | (Growth) | 0.7% | 7.4% | 3.8% | | | | | | | | | | | |
| | 2.0 | NP | 56.2% | 35.7% | 13.4% | | | | | | | | | | | |
| | | DSRS | 56.7% | 38.4% | 16.9% | | | | | | | | | | | |
| | | (Growth) | 0.5% | 2.7% | 3.5% | | | | | | | | | | | |
| | 4.0 | NP | 57.3% | 38.5% | 18.5% | | | | | | | | | | | |
| | | DSRS | 57.5% | 39.3% | 20.0% | | | | | | | | | | | |
| | | (Growth) | 0.2% | 0.8% | 1.5% | | | | | | | | | | | |
| | 8.0 | NP | 57.5% | 39.2% | 21.2% | | | | | | | | | | | |
| | | DSRS | 57.6% | 40.1% | 22.0% | | | | | | | | | | | |
| | | (Growth) | 0.1% | 0.9% | 0.8% | | | | | | | | | | | |
| 0.50 | 0.25 | NP | 50.9% | 23.7% | 3.9% | 0.1% | | | | | | | | | | |
| | | DSRS | 54.2% | 37.6% | 16.4% | 2.0% | | | | | | | | | | |
| | | (Growth) | 3.3% | 13.9% | 12.5% | 1.9% | | | | | | | | | | |
| | 0.5 | NP | 53.0% | 34.6% | 16.6% | 4.5% | 0.3% | | | | | | | | | |
| | | DSRS | 53.6% | 40.4% | 25.2% | 12.9% | 2.5% | | | | | | | | | |
| | | (Growth) | 0.6% | 5.8% | 8.6% | 8.4% | 2.2% | | | | | | | | | |
| | 1.0 | NP | 53.7% | 39.8% | 23.1% | 12.4% | 3.6% | 0.7% | | | | | | | | |
| | | DSRS | 54.1% | 41.7% | 28.2% | 17.3% | 9.2% | 1.8% | | | | | | | | |
| | | (Growth) | 0.4% | 1.9% | 5.1% | 4.9% | 5.6% | 1.1% | | | | | | | | |
| | 2.0 | NP | 53.8% | 41.2% | 27.9% | 17.0% | 8.9% | 2.9% | 0.1% | | | | | | | |
| | | DSRS | 54.0% | 42.4% | 29.3% | 19.4% | 11.6% | 3.8% | 0.6% | | | | | | | |
| | | (Growth) | 0.2% | 1.2% | 1.4% | 2.4% | 2.7% | 0.9% | 0.5% | | | | | | | |
| | 4.0 | NP | 53.9% | 42.2% | 29.7% | 19.0% | 11.3% | 4.7% | 1.5% | | | | | | | |
| | | DSRS | 54.0% | 42.5% | 30.0% | 20.2% | 12.8% | 5.8% | 1.5% | | | | | | | |
| | | (Growth) | 0.1% | 0.3% | 0.3% | 1.2% | 1.5% | 1.1% | 0.0% | | | | | | | |
| | 8.0 | NP | 54.0% | 42.6% | 30.0% | 20.0% | 12.7% | 6.8% | 2.3% | | | | | | | |
| | | DSRS | 54.2% | 42.5% | 30.9% | 20.6% | 13.4% | 6.8% | 1.8% | | | | | | | |
| | | (Growth) | 0.2% | -0.1% | 0.9% | 0.6% | 0.7% | 0.0% | -0.5% | | | | | | | |
| 1.00 | 0.25 | NP | 40.6% | 28.7% | 18.0% | 9.0% | 3.1% | 0.4% | | | | | | | | |
| | | DSRS | 41.8% | 32.6% | 23.5% | 16.5% | 9.4% | 4.5% | 0.5% | 0.1% | | | | | | |
| | | (Growth) | 1.2% | 3.9% | 5.5% | 7.5% | 6.3% | 4.1% | | | | | | | | |
| | 0.5 | NP | 40.9% | 31.8% | 21.8% | 15.7% | 9.1% | 4.9% | 1.6% | 0.2% | 0.1% | | | | | |
| | | DSRS | 41.1% | 33.6% | 24.7% | 19.1% | 13.4% | 8.5% | 5.5% | 2.0% | 0.4% | 0.1% | | | | |
| | | (Growth) | 0.2% | 1.8% | 2.9% | 3.4% | 4.3% | 3.6% | 3.9% | 1.8% | 0.3% | | | | | |
| | 1.0 | NP | 40.4% | 32.4% | 24.0% | 17.7% | 12.6% | 8.5% | 5.0% | 2.2% | 0.5% | 0.1% | 0.1% | | | |
| | | DSRS | 40.2% | 33.2% | 25.5% | 20.0% | 15.1% | 10.5% | 7.1% | 4.2% | 1.9% | 0.9% | 0.1% | | | |
| | | (Growth) | -0.2% | 0.8% | 1.5% | 2.3% | 2.5% | 2.0% | 2.1% | 2.0% | 1.4% | 0.8% | 0.0% | | | |
| | 2.0 | NP | 40.2% | 32.6% | 24.6% | 18.9% | 15.0% | 10.1% | 7.4% | 4.0% | 2.0% | 0.7% | 0.1% | 0.1% | | |
| | | DSRS | 40.2% | 32.8% | 25.5% | 20.2% | 15.7% | 11.5% | 8.0% | 5.5% | 2.6% | 1.5% | 0.6% | 0.1% | | |
| | | (Growth) | 0.0% | 0.2% | 0.9% | 1.3% | 0.7% | 1.4% | 0.6% | 1.5% | 0.6% | 0.8% | 0.5% | 0.0% | | |
| | 4.0 | NP | 40.0% | 32.7% | 25.2% | 19.7% | 15.5% | 11.4% | 8.2% | 5.3% | 3.0% | 1.5% | 0.6% | 0.1% | 0.1% | |
| | | DSRS | 39.6% | 32.7% | 25.7% | 20.2% | 15.9% | 12.2% | 8.5% | 6.5% | 3.4% | 1.8% | 0.9% | 0.4% | | |
| | | (Growth) | -0.4% | 0.0% | 0.5% | 0.5% | 0.4% | 0.8% | 0.3% | 1.2% | 0.4% | 0.3% | 0.3% | 0.3% | | |
| | 8.0 | NP | 39.7% | 32.5% | 25.5% | 20.2% | 15.7% | 12.1% | 8.5% | 6.3% | 3.4% | 2.0% | 0.9% | 0.5% | 0.2% | |
| | | DSRS | 39.5% | 32.6% | 25.5% | 20.2% | 15.8% | 12.3% | 8.6% | 6.6% | 3.7% | 2.1% | 1.1% | 0.5% | 0.2% | |
| | | (Growth) | -0.2% | 0.1% | 0.0% | 0.0% | 0.1% | 0.2% | 0.1% | 0.3% | 0.3% | 0.1% | 0.2% | 0.0% | 0.0% | |

Table 12: Full experimental results for certified accuracy, standard augmentation, EGG, ImageNet

| σ | η | Certification method | Certified accuracy at $r$ | | | | | | | | | | | | | |
|---|---|---|---|---|---|---|---|---|---|---|---|---|---|---|---|---|
| | | | 0.25 | 0.50 | 0.75 | 1.00 | 1.25 | 1.50 | 1.75 | 2.00 | 2.25 | 2.50 | 2.75 | 3.00 | 3.25 | 3.50 |
| 0.25 | 0.25 | NP | 9.9% | | | | | | | | | | | | | |
| | | DSRS | 45.8% | | | | | | | | | | | | | |
| | | (Growth) | 35.9% | | | | | | | | | | | | | |
| | 0.5 | NP | 44.6% | 0.6% | | | | | | | | | | | | |
| | | DSRS | 54.9% | 9.7% | | | | | | | | | | | | |
| | | (Growth) | 10.3% | 9.1% | | | | | | | | | | | | |
| | 1.0 | NP | 54.0% | 21.1% | 0.6% | | | | | | | | | | | |
| | | DSRS | 57.0% | 39.3% | 1.4% | | | | | | | | | | | |
| | | (Growth) | 3.0% | 18.2% | 0.8% | | | | | | | | | | | |
| | 2.0 | NP | 57.1% | 41.7% | 17.6% | | | | | | | | | | | |
| | | DSRS | 58.4% | 47.9% | 24.1% | | | | | | | | | | | |
| | | (Growth) | 1.3% | 6.2% | 6.5% | | | | | | | | | | | |
| | 4.0 | NP | 58.4% | 48.0% | 33.6% | | | | | | | | | | | |
| | | DSRS | 58.7% | 49.9% | 36.2% | | | | | | | | | | | |
| | | (Growth) | 0.3% | 1.9% | 2.6% | | | | | | | | | | | |
| | 8.0 | NP | 58.7% | 50.0% | 37.8% | | | | | | | | | | | |
| | | DSRS | 59.1% | 50.8% | 38.7% | | | | | | | | | | | |
| | | (Growth) | 0.4% | 0.8% | 0.9% | | | | | | | | | | | |
| 0.50 | 0.25 | NP | 49.7% | 26.7% | 0.1% | | | | | | | | | | | |
| | | DSRS | 53.8% | 41.4% | 14.8% | | | | | | | | | | | |
| | | (Growth) | 4.1% | 14.7% | 14.7% | | | | | | | | | | | |
| | 0.5 | NP | 52.1% | 40.6% | 26.4% | 8.3% | 0.1% | | | | | | | | | |
| | | DSRS | 54.4% | 46.3% | 36.4% | 22.5% | 2.9% | | | | | | | | | |
| | | (Growth) | 2.3% | 5.7% | 10.0% | 14.2% | 2.8% | | | | | | | | | |
| | 1.0 | NP | 52.8% | 44.8% | 35.7% | 26.2% | 16.1% | 6.3% | | | | | | | | |
| | | DSRS | 54.2% | 47.8% | 39.9% | 32.8% | 22.9% | 8.9% | | | | | | | | |
| | | (Growth) | 1.4% | 3.0% | 4.2% | 6.6% | 6.8% | 2.6% | | | | | | | | |
| | 2.0 | NP | 53.4% | 47.0% | 39.4% | 33.3% | 24.5% | 17.4% | 8.4% | | | | | | | |
| | | DSRS | 53.8% | 48.5% | 41.5% | 35.2% | 28.9% | 19.4% | 11.3% | | | | | | | |
| | | (Growth) | 0.4% | 1.5% | 2.1% | 1.9% | 4.4% | 2.0% | 2.9% | | | | | | | |
| | 4.0 | NP | 53.3% | 47.7% | 41.3% | 35.2% | 29.3% | 21.3% | 14.0% | | | | | | | |
| | | DSRS | 53.6% | 48.6% | 42.6% | 36.4% | 31.0% | 22.8% | 14.4% | | | | | | | |
| | | (Growth) | 0.3% | 0.9% | 1.3% | 1.2% | 1.7% | 1.5% | 0.4% | | | | | | | |
| | 8.0 | NP | 53.3% | 48.3% | 42.2% | 36.5% | 31.1% | 24.0% | 16.9% | | | | | | | |
| | | DSRS | 53.6% | 48.8% | 42.9% | 36.8% | 31.8% | 24.6% | 16.5% | | | | | | | |
| | | (Growth) | 0.3% | 0.5% | 0.7% | 0.3% | 0.7% | 0.6% | -0.4% | | | | | | | |
| 1.00 | 0.25 | NP | 38.8% | 29.8% | 15.7% | 3.3% | | | | | | | | | | |
| | | DSRS | 41.0% | 35.3% | 28.4% | 20.1% | 7.1% | 0.8% | | | | | | | | |
| | | (Growth) | 2.2% | 5.5% | 12.7% | 16.8% | | | | | | | | | | |
| | 0.5 | NP | 40.3% | 34.1% | 26.4% | 19.0% | 10.7% | 4.1% | 1.4% | 0.1% | | | | | | |
| | | DSRS | 40.9% | 37.0% | 31.6% | 26.3% | 22.1% | 15.2% | 8.7% | 3.1% | 0.8% | | | | | |
| | | (Growth) | 0.6% | 2.9% | 5.2% | 7.3% | 11.4% | 11.1% | 7.3% | 3.0% | | | | | | |
| | 1.0 | NP | 41.7% | 36.4% | 30.7% | 25.5% | 20.9% | 15.3% | 10.5% | 6.3% | 3.3% | 1.7% | 0.7% | 0.2% | | |
| | | DSRS | 42.0% | 37.9% | 33.4% | 29.3% | 24.9% | 22.0% | 18.5% | 13.1% | 9.2% | 5.0% | 2.1% | 0.5% | | |
| | | (Growth) | 0.3% | 1.5% | 2.7% | 3.8% | 4.0% | 6.7% | 8.0% | 6.8% | 5.9% | 3.3% | 1.4% | 0.3% | | |
| | 2.0 | NP | 42.5% | 37.2% | 32.8% | 29.2% | 24.7% | 21.4% | 17.4% | 13.8% | 10.1% | 7.8% | 5.5% | 3.3% | 2.2% | 1.1% |
| | | DSRS | 42.9% | 38.2% | 34.3% | 30.2% | 26.8% | 23.3% | 21.3% | 18.8% | 14.1% | 11.1% | 8.9% | 6.1% | 2.2% | 1.4% |
| | | (Growth) | 0.4% | 1.0% | 1.5% | 1.0% | 2.1% | 1.9% | 3.9% | 5.0% | 4.0% | 3.3% | 3.4% | 2.8% | 0.0% | 0.3% |
| | 4.0 | NP | 42.2% | 38.0% | 33.8% | 30.8% | 26.1% | 23.3% | 21.1% | 18.2% | 13.9% | 11.2% | 9.3% | 8.1% | 6.0% | 4.2% |
| | | DSRS | 42.5% | 38.5% | 34.6% | 31.3% | 27.5% | 23.9% | 22.3% | 20.2% | 17.3% | 13.2% | 10.7% | 9.2% | 6.8% | 4.0% |
| | | (Growth) | 0.3% | 0.5% | 0.8% | 0.5% | 1.4% | 0.6% | 1.2% | 2.0% | 3.4% | 2.0% | 1.4% | 1.1% | 0.8% | -0.2% |
| | 8.0 | NP | 42.4% | 38.1% | 34.5% | 31.1% | 26.9% | 24.0% | 22.2% | 19.9% | 16.8% | 12.8% | 11.0% | 9.5% | 8.0% | 6.1% |
| | | DSRS | 42.4% | 38.4% | 35.0% | 31.4% | 28.0% | 24.4% | 22.6% | 20.7% | 18.9% | 14.5% | 11.7% | 10.1% | 8.6% | 5.2% |
| | | (Growth) | 0.0% | 0.3% | 0.5% | 0.3% | 1.1% | 0.4% | 0.4% | 0.8% | 2.1% | 1.7% | 0.7% | 0.6% | 0.6% | -0.9% |

Table 13: Full experimental results for certified accuracy, standard augmentation, ESG, CIFAR10

| $\sigma$ | $\eta$ | Certification method | \multicolumn{14}{c|}{Certified accuracy at $r$} |
| | | | 0.25 | 0.50 | 0.75 | 1.00 | 1.25 | 1.50 | 1.75 | 2.00 | 2.25 | 2.50 | 2.75 | 3.00 | 3.25 | 3.50 |
|---|---|---|---|---|---|---|---|---|---|---|---|---|---|---|---|---|
| 0.25 | 1.0 | NP | 57.8% | 40.7% | 25.6% | | | | | | | | | | | |
| | | DSRS | 57.6% | 40.7% | 25.1% | | | | | | | | | | | |
| | | (Growth) | -0.2% | 0.0% | -0.5% | | | | | | | | | | | |
| | 2.0 | NP | 57.8% | 40.8% | 25.7% | | | | | | | | | | | |
| | | DSRS | 57.6% | 40.6% | 25.1% | | | | | | | | | | | |
| | | (Growth) | -0.2% | -0.2% | -0.6% | | | | | | | | | | | |
| | 4.0 | NP | 57.9% | 40.9% | 26.0% | | | | | | | | | | | |
| | | DSRS | 57.6% | 40.6% | 25.0% | | | | | | | | | | | |
| | | (Growth) | -0.3% | -0.3% | -1.0% | | | | | | | | | | | |
| | 8.0 | NP | 57.9% | 40.8% | 25.6% | | | | | | | | | | | |
| | | DSRS | 57.8% | 40.6% | 24.9% | | | | | | | | | | | |
| | | (Growth) | -0.1% | -0.2% | -0.7% | | | | | | | | | | | |
| 0.50 | 1.0 | NP | 54.3% | 42.7% | 31.7% | 21.8% | 14.0% | 8.7% | 3.5% | | | | | | | |
| | | DSRS | 54.2% | 42.6% | 31.3% | 21.5% | 14.1% | 8.3% | 2.7% | | | | | | | |
| | | (Growth) | -0.1% | -0.1% | -0.4% | -0.3% | 0.1% | -0.4% | -0.8% | | | | | | | |
| | 2.0 | NP | 54.3% | 42.6% | 31.6% | 21.7% | 14.0% | 8.5% | 3.6% | | | | | | | |
| | | DSRS | 54.3% | 42.6% | 31.6% | 21.5% | 13.9% | 8.7% | 2.8% | | | | | | | |
| | | (Growth) | 0.0% | 0.0% | 0.0% | -0.2% | -0.1% | 0.2% | -0.8% | | | | | | | |
| | 4.0 | NP | 54.3% | 42.7% | 31.5% | 21.6% | 14.3% | 8.6% | 3.6% | | | | | | | |
| | | DSRS | 54.3% | 42.6% | 31.3% | 21.5% | 13.9% | 8.2% | 3.0% | | | | | | | |
| | | (Growth) | 0.0% | -0.1% | -0.2% | -0.1% | -0.4% | -0.4% | -0.6% | | | | | | | |
| | 8.0 | NP | 54.3% | 42.6% | 31.7% | 21.7% | 14.1% | 8.7% | 3.5% | | | | | | | |
| | | DSRS | 54.4% | 42.6% | 31.6% | 21.6% | 14.0% | 8.1% | 2.4% | | | | | | | |
| | | (Growth) | 0.1% | 0.0% | -0.1% | -0.1% | -0.1% | -0.6% | -1.1% | | | | | | | |
| 1.00 | 1.0 | NP | 39.6% | 32.6% | 26.0% | 20.5% | 15.9% | 13.0% | 9.2% | 7.0% | 4.5% | 2.5% | 1.5% | 0.9% | 0.5% | 0.2% |
| | | DSRS | 39.6% | 32.5% | 25.7% | 20.4% | 15.8% | 12.8% | 8.6% | 6.8% | 4.3% | 2.3% | 1.3% | 0.8% | 0.3% | 0.1% |
| | | (Growth) | 0.0% | -0.1% | -0.3% | -0.1% | -0.1% | -0.2% | -0.6% | -0.2% | -0.2% | -0.2% | -0.2% | -0.1% | -0.2% | -0.1% |
| | 2.0 | NP | 39.6% | 32.6% | 25.9% | 20.4% | 15.9% | 13.0% | 9.2% | 7.0% | 4.6% | 2.5% | 1.3% | 0.8% | 0.3% | 0.2% |
| | | DSRS | 39.5% | 32.6% | 25.8% | 20.4% | 15.8% | 12.7% | 8.8% | 6.8% | 4.5% | 2.4% | 1.3% | 0.7% | 0.2% | 0.2% |
| | | (Growth) | -0.1% | 0.0% | -0.1% | 0.0% | -0.1% | -0.3% | -0.4% | -0.2% | -0.1% | -0.1% | 0.0% | -0.1% | -0.1% | 0.0% |
| | 4.0 | NP | 39.7% | 32.6% | 25.9% | 20.4% | 15.9% | 12.9% | 9.0% | 7.0% | 4.6% | 2.5% | 1.5% | 0.8% | 0.3% | 0.2% |
| | | DSRS | 39.5% | 32.6% | 25.8% | 20.3% | 15.9% | 12.9% | 8.6% | 6.9% | 4.3% | 2.4% | 1.3% | 0.8% | 0.2% | 0.1% |
| | | (Growth) | -0.2% | 0.0% | -0.1% | -0.1% | 0.0% | 0.0% | -0.4% | -0.1% | -0.3% | -0.1% | -0.2% | 0.0% | -0.1% | -0.1% |
| | 8.0 | NP | 39.6% | 32.6% | 25.8% | 20.3% | 15.9% | 12.9% | 9.2% | 7.0% | 4.5% | 2.5% | 1.3% | 0.8% | 0.3% | 0.2% |
| | | DSRS | 39.7% | 32.4% | 25.8% | 20.4% | 15.9% | 12.9% | 8.9% | 6.7% | 4.2% | 2.4% | 1.3% | 0.9% | 0.2% | 0.1% |
| | | (Growth) | 0.1% | -0.2% | 0.0% | 0.1% | 0.0% | 0.0% | -0.3% | -0.3% | -0.3% | -0.1% | 0.0% | 0.1% | -0.1% | -0.1% |

Table 14: Full experimental results for certified accuracy, standard augmentation, ESG, ImageNet

| $\sigma$ | $\eta$ | Certification method | \multicolumn{14}{c|}{Certified accuracy at $r$} |
| | | | 0.25 | 0.50 | 0.75 | 1.00 | 1.25 | 1.50 | 1.75 | 2.00 | 2.25 | 2.50 | 2.75 | 3.00 | 3.25 | 3.50 |
|---|---|---|---|---|---|---|---|---|---|---|---|---|---|---|---|---|
| 0.25 | 1.0 | NP | 59.6% | 51.6% | 42.2% | | | | | | | | | | | |
| | | DSRS | 59.6% | 51.5% | 41.6% | | | | | | | | | | | |
| | | (Growth) | 0.0% | -0.1% | -0.6% | | | | | | | | | | | |
| | 2.0 | NP | 59.6% | 51.7% | 41.9% | | | | | | | | | | | |
| | | DSRS | 59.6% | 51.6% | 41.8% | | | | | | | | | | | |
| | | (Growth) | 0.0% | -0.1% | -0.1% | | | | | | | | | | | |
| | 4.0 | NP | 59.6% | 51.7% | 42.0% | | | | | | | | | | | |
| | | DSRS | 59.6% | 51.5% | 41.9% | | | | | | | | | | | |
| | | (Growth) | 0.0% | -0.2% | -0.1% | | | | | | | | | | | |
| | 8.0 | NP | 59.6% | 51.6% | 42.2% | | | | | | | | | | | |
| | | DSRS | 59.6% | 51.5% | 41.5% | | | | | | | | | | | |
| | | (Growth) | 0.0% | -0.1% | -0.7% | | | | | | | | | | | |
| 0.50 | 1.0 | NP | 53.6% | 49.3% | 43.2% | 38.2% | 33.0% | 27.2% | 21.0% | | | | | | | |
| | | DSRS | 53.6% | 49.1% | 43.2% | 37.9% | 33.0% | 26.8% | 19.1% | | | | | | | |
| | | (Growth) | 0.0% | -0.2% | 0.0% | -0.3% | 0.0% | -0.4% | -1.9% | | | | | | | |
| | 2.0 | NP | 53.7% | 49.2% | 43.2% | 38.1% | 33.1% | 27.3% | 20.7% | | | | | | | |
| | | DSRS | 53.6% | 49.2% | 43.1% | 38.0% | 32.9% | 26.9% | 19.2% | | | | | | | |
| | | (Growth) | -0.1% | 0.0% | -0.1% | -0.1% | -0.2% | -0.4% | -1.5% | | | | | | | |
| | 4.0 | NP | 53.6% | 49.2% | 43.2% | 38.1% | 33.0% | 27.5% | 20.8% | | | | | | | |
| | | DSRS | 53.6% | 49.2% | 43.2% | 38.0% | 32.9% | 27.2% | 19.4% | | | | | | | |
| | | (Growth) | 0.0% | 0.0% | 0.0% | -0.1% | -0.1% | -0.3% | -1.4% | | | | | | | |
| | 8.0 | NP | 53.7% | 49.2% | 43.2% | 38.1% | 33.1% | 27.4% | 21.0% | | | | | | | |
| | | DSRS | 53.6% | 49.1% | 43.2% | 38.0% | 33.0% | 26.8% | 19.4% | | | | | | | |
| | | (Growth) | -0.1% | -0.1% | 0.0% | -0.1% | -0.1% | -0.6% | -1.6% | | | | | | | |
| 1.00 | 1.0 | NP | 42.7% | 39.1% | 35.4% | 32.0% | 29.5% | 25.3% | 23.1% | 21.6% | 20.0% | 17.6% | 13.9% | 11.7% | 10.5% | 9.1% |
| | | DSRS | 42.6% | 38.8% | 35.3% | 31.9% | 28.9% | 25.3% | 23.1% | 21.5% | 19.9% | 17.4% | 13.8% | 11.5% | 10.3% | 7.7% |
| | | (Growth) | -0.1% | -0.3% | -0.1% | -0.1% | -0.6% | 0.0% | 0.0% | -0.1% | -0.1% | -0.2% | -0.1% | -0.2% | -0.2% | -1.4% |
| | 2.0 | NP | 42.6% | 39.1% | 35.3% | 32.1% | 29.2% | 25.3% | 23.2% | 21.6% | 20.0% | 17.6% | 14.1% | 11.7% | 10.5% | 9.1% |
| | | DSRS | 42.5% | 39.0% | 35.2% | 31.7% | 29.0% | 25.2% | 23.1% | 21.5% | 19.7% | 17.4% | 13.6% | 11.4% | 10.1% | 8.3% |
| | | (Growth) | -0.1% | -0.1% | -0.1% | -0.4% | -0.2% | -0.1% | -0.1% | -0.1% | -0.3% | -0.2% | -0.5% | -0.3% | -0.4% | -0.8% |
| | 4.0 | NP | 42.6% | 39.1% | 35.4% | 32.0% | 29.1% | 25.3% | 23.1% | 21.6% | 19.9% | 17.9% | 14.5% | 11.7% | 10.5% | 9.1% |
| | | DSRS | 42.5% | 38.8% | 35.2% | 31.9% | 29.3% | 25.3% | 23.1% | 21.6% | 19.9% | 17.2% | 13.6% | 11.4% | 10.2% | 8.0% |
| | | (Growth) | -0.1% | -0.3% | -0.2% | -0.1% | 0.2% | 0.0% | 0.0% | 0.0% | 0.0% | -0.7% | -0.9% | -0.3% | -0.3% | -1.1% |
| | 8.0 | NP | 42.7% | 39.0% | 35.3% | 31.9% | 29.4% | 25.3% | 23.1% | 21.7% | 20.0% | 17.5% | 13.9% | 11.5% | 10.4% | 9.3% |
| | | DSRS | 42.5% | 38.8% | 35.3% | 32.0% | 29.0% | 25.2% | 23.1% | 21.6% | 19.7% | 17.3% | 13.6% | 11.5% | 10.1% | 8.4% |
| | | (Growth) | -0.2% | -0.2% | 0.0% | 0.1% | -0.4% | -0.1% | 0.0% | -0.1% | -0.3% | -0.2% | -0.3% | 0.0% | -0.3% | -0.9% |

Table 15: Full experimental results for certified accuracy, Consistency, EGG, CIFAR10

| $\sigma$ | $\eta$ | Certification method | 0.25 | 0.50 | 0.75 | 1.00 | 1.25 | 1.50 | 1.75 | 2.00 | 2.25 | 2.50 | 2.75 | 3.00 | 3.25 | 3.50 |
|---|---|---|---|---|---|---|---|---|---|---|---|---|---|---|---|---|
| | | | | | | | | | | | | | | Certified accuracy at $r$ | | |
| 0.25 | 1.0 | NP | 61.4% | 47.0% | 25.2% | | | | | | | | | | | |
| | | DSRS | 62.1% | 50.7% | 30.1% | | | | | | | | | | | |
| | | (Growth) | 0.7% | 3.7% | 4.9% | | | | | | | | | | | |
| | 2.0 | NP | 61.8% | 50.7% | 35.1% | | | | | | | | | | | |
| | | DSRS | 62.5% | 52.0% | 37.2% | | | | | | | | | | | |
| | | (Growth) | 0.7% | 1.3% | 2.1% | | | | | | | | | | | |
| | 4.0 | NP | 62.3% | 51.7% | 38.2% | | | | | | | | | | | |
| | | DSRS | 62.5% | 52.2% | 39.1% | | | | | | | | | | | |
| | | (Growth) | 0.2% | 0.5% | 0.9% | | | | | | | | | | | |
| | 8.0 | NP | 62.5% | 52.2% | 40.2% | | | | | | | | | | | |
| | | DSRS | 62.5% | 52.6% | 40.4% | | | | | | | | | | | |
| | | (Growth) | 0.0% | 0.4% | 0.2% | | | | | | | | | | | |
| 0.50 | 1.0 | NP | 49.2% | 43.1% | 36.3% | 28.7% | 19.6% | 11.6% | | | | | | | | |
| | | DSRS | 49.5% | 43.7% | 38.2% | 33.1% | 24.2% | 15.4% | | | | | | | | |
| | | (Growth) | 0.3% | 0.6% | 1.9% | 4.4% | 4.6% | 3.8% | | | | | | | | |
| | 2.0 | NP | 49.3% | 43.8% | 37.8% | 32.3% | 23.5% | 18.0% | 9.6% | | | | | | | |
| | | DSRS | 49.4% | 44.1% | 38.5% | 34.4% | 27.4% | 19.6% | 11.5% | | | | | | | |
| | | (Growth) | 0.1% | 0.3% | 0.7% | 2.1% | 3.9% | 1.6% | 1.9% | | | | | | | |
| | 4.0 | NP | 49.3% | 44.0% | 38.3% | 34.1% | 27.4% | 20.4% | 14.6% | | | | | | | |
| | | DSRS | 49.3% | 44.0% | 38.7% | 35.4% | 28.3% | 21.1% | 14.1% | | | | | | | |
| | | (Growth) | 0.0% | 0.0% | 0.4% | 1.3% | 0.9% | 0.7% | -0.5% | | | | | | | |
| | 8.0 | NP | 49.3% | 44.1% | 38.7% | 35.0% | 28.4% | 21.9% | 15.8% | | | | | | | |
| | | DSRS | 49.3% | 44.1% | 38.8% | 35.3% | 28.6% | 22.3% | 15.2% | | | | | | | |
| | | (Growth) | 0.0% | 0.0% | 0.1% | 0.3% | 0.2% | 0.4% | -0.6% | | | | | | | |
| 1.00 | 1.0 | NP | 37.2% | 32.5% | 29.4% | 25.2% | 21.8% | 17.6% | 14.4% | 11.7% | 8.8% | 6.3% | 4.5% | 2.6% | 1.8% | |
| | | DSRS | 37.4% | 33.0% | 29.5% | 26.2% | 22.8% | 19.2% | 16.7% | 14.3% | 11.3% | 9.2% | 6.6% | 4.1% | 1.4% | |
| | | (Growth) | 0.2% | 0.5% | 0.1% | 1.0% | 1.0% | 1.6% | 2.3% | 2.6% | 2.5% | 2.9% | 2.1% | 1.5% | -0.4% | |
| | 2.0 | NP | 37.2% | 32.6% | 29.7% | 25.9% | 22.4% | 19.0% | 16.3% | 13.9% | 11.3% | 8.9% | 7.2% | 5.1% | 3.5% | 2.2% |
| | | DSRS | 37.1% | 32.3% | 29.8% | 26.5% | 23.0% | 20.6% | 17.0% | 14.7% | 12.7% | 10.5% | 8.5% | 6.3% | 3.9% | 2.5% |
| | | (Growth) | -0.1% | -0.3% | 0.1% | 0.6% | 0.6% | 1.6% | 0.7% | 0.8% | 1.4% | 1.6% | 1.3% | 1.2% | 0.4% | 0.3% |
| | 4.0 | NP | 37.1% | 32.5% | 29.8% | 26.2% | 22.7% | 20.3% | 16.9% | 14.9% | 12.4% | 10.6% | 8.7% | 6.7% | 5.1% | 3.2% |
| | | DSRS | 37.0% | 32.4% | 29.8% | 26.7% | 23.1% | 20.9% | 17.5% | 15.3% | 13.0% | 10.9% | 9.2% | 7.0% | 5.2% | 3.1% |
| | | (Growth) | -0.1% | -0.1% | 0.0% | 0.5% | 0.4% | 0.6% | 0.6% | 0.4% | 0.6% | 0.3% | 0.5% | 0.3% | 0.1% | -0.1% |
| | 8.0 | NP | 37.1% | 32.5% | 29.9% | 26.4% | 23.0% | 20.7% | 17.2% | 15.2% | 13.2% | 10.9% | 9.6% | 7.5% | 5.9% | 4.2% |
| | | DSRS | 36.7% | 32.5% | 29.8% | 26.6% | 23.2% | 20.9% | 17.6% | 15.5% | 13.2% | 11.3% | 9.6% | 7.8% | 5.8% | 3.9% |
| | | (Growth) | -0.4% | 0.0% | -0.1% | 0.2% | 0.2% | 0.2% | 0.4% | 0.3% | 0.0% | 0.4% | 0.0% | 0.3% | -0.1% | -0.3% |

Table 16: Full experimental results for certified accuracy, Consistency, ESG, CIFAR10

| $\sigma$ | $\eta$ | Certification method | 0.25 | 0.50 | 0.75 | 1.00 | 1.25 | 1.50 | 1.75 | 2.00 | 2.25 | 2.50 | 2.75 | 3.00 | 3.25 | 3.50 |
|---|---|---|---|---|---|---|---|---|---|---|---|---|---|---|---|---|
| | | | | | | | | | | | | | | Certified accuracy at $r$ | | |
| 0.25 | 1.0 | NP | 62.7% | 52.9% | 41.8% | | | | | | | | | | | |
| | | DSRS | 62.6% | 52.9% | 41.6% | | | | | | | | | | | |
| | | (Growth) | -0.1% | 0.0% | -0.2% | | | | | | | | | | | |
| | 2.0 | NP | 62.7% | 53.0% | 42.1% | | | | | | | | | | | |
| | | DSRS | 62.7% | 53.0% | 41.4% | | | | | | | | | | | |
| | | (Growth) | 0.0% | 0.0% | -0.7% | | | | | | | | | | | |
| | 4.0 | NP | 62.7% | 53.0% | 42.0% | | | | | | | | | | | |
| | | DSRS | 62.7% | 52.9% | 41.6% | | | | | | | | | | | |
| | | (Growth) | 0.0% | -0.1% | -0.4% | | | | | | | | | | | |
| | 8.0 | NP | 62.7% | 53.0% | 42.0% | | | | | | | | | | | |
| | | DSRS | 62.7% | 52.9% | 41.7% | | | | | | | | | | | |
| | | (Growth) | 0.0% | -0.1% | -0.3% | | | | | | | | | | | |
| 0.50 | 1.0 | NP | 49.3% | 44.1% | 39.2% | 35.5% | 29.7% | 24.3% | 18.7% | | | | | | | |
| | | DSRS | 49.3% | 44.1% | 38.9% | 35.5% | 29.3% | 23.7% | 16.9% | | | | | | | |
| | | (Growth) | 0.0% | 0.0% | -0.3% | 0.0% | -0.4% | -0.6% | -1.8% | | | | | | | |
| | 2.0 | NP | 49.3% | 44.1% | 39.2% | 35.5% | 29.7% | 24.1% | 18.7% | | | | | | | |
| | | DSRS | 49.3% | 44.1% | 38.9% | 35.5% | 29.4% | 24.0% | 17.1% | | | | | | | |
| | | (Growth) | 0.0% | 0.0% | -0.3% | 0.0% | -0.3% | -0.1% | -1.6% | | | | | | | |
| | 4.0 | NP | 49.3% | 44.1% | 39.1% | 35.5% | 29.9% | 24.1% | 18.6% | | | | | | | |
| | | DSRS | 49.3% | 44.0% | 39.0% | 35.5% | 29.5% | 23.8% | 17.6% | | | | | | | |
| | | (Growth) | 0.0% | -0.1% | -0.1% | 0.0% | -0.4% | -0.3% | -1.0% | | | | | | | |
| | 8.0 | NP | 49.3% | 44.1% | 39.3% | 35.5% | 29.6% | 24.2% | 19.0% | | | | | | | |
| | | DSRS | 49.3% | 44.1% | 38.9% | 35.5% | 29.1% | 23.8% | 17.3% | | | | | | | |
| | | (Growth) | 0.0% | 0.0% | -0.4% | 0.0% | -0.5% | -0.4% | -1.7% | | | | | | | |
| 1.00 | 1.0 | NP | 36.7% | 32.6% | 30.0% | 26.9% | 23.4% | 21.0% | 18.0% | 16.0% | 14.0% | 12.2% | 10.4% | 8.8% | 7.0% | 5.4% |
| | | DSRS | 36.5% | 32.3% | 29.8% | 27.0% | 23.2% | 20.9% | 17.7% | 15.8% | 13.7% | 11.8% | 10.0% | 8.8% | 6.8% | 4.6% |
| | | (Growth) | -0.2% | -0.3% | -0.2% | 0.1% | -0.2% | -0.1% | -0.3% | -0.2% | -0.3% | -0.4% | -0.4% | 0.0% | -0.2% | -0.8% |
| | 2.0 | NP | 36.8% | 32.5% | 29.9% | 27.1% | 23.3% | 21.0% | 18.1% | 16.0% | 14.0% | 12.2% | 10.3% | 8.8% | 7.1% | 5.5% |
| | | DSRS | 36.6% | 32.3% | 29.9% | 26.8% | 23.3% | 21.0% | 17.7% | 16.0% | 13.8% | 11.9% | 10.1% | 8.7% | 6.7% | 4.4% |
| | | (Growth) | -0.2% | -0.2% | 0.0% | -0.3% | 0.0% | 0.0% | -0.4% | 0.0% | -0.2% | -0.3% | -0.2% | -0.1% | -0.4% | -1.1% |
| | 4.0 | NP | 36.6% | 32.5% | 29.9% | 27.0% | 23.2% | 21.0% | 18.0% | 16.1% | 14.0% | 12.2% | 10.3% | 9.0% | 7.2% | 5.3% |
| | | DSRS | 36.6% | 32.3% | 29.8% | 26.9% | 23.3% | 20.9% | 17.8% | 15.7% | 13.9% | 11.9% | 10.2% | 8.8% | 6.7% | 4.8% |
| | | (Growth) | 0.0% | -0.2% | -0.1% | -0.1% | 0.1% | -0.1% | -0.2% | -0.4% | -0.1% | -0.3% | -0.1% | -0.2% | -0.5% | -0.5% |
| | 8.0 | NP | 36.7% | 32.6% | 29.8% | 27.0% | 23.3% | 21.0% | 17.9% | 16.1% | 13.8% | 12.2% | 10.3% | 9.0% | 7.1% | 5.1% |
| | | DSRS | 36.6% | 32.1% | 29.8% | 26.9% | 23.2% | 20.9% | 17.6% | 15.8% | 13.6% | 11.8% | 10.1% | 8.3% | 6.7% | 4.8% |
| | | (Growth) | -0.1% | -0.5% | 0.0% | -0.1% | -0.1% | -0.1% | -0.3% | -0.3% | -0.2% | -0.4% | -0.2% | -0.7% | -0.4% | -0.3% |

Table 17: Full experimental results for certified accuracy, SmoothMix, EGG, CIFAR10

| σ | η | Certification method | \multicolumn{14}{c}{Certified accuracy at r} |
|---|---|---|---|
| | | | 0.25 | 0.50 | 0.75 | 1.00 | 1.25 | 1.50 | 1.75 | 2.00 | 2.25 | 2.50 | 2.75 | 3.00 | 3.25 | 3.50 |
| 0.25 | 1.0 | NP | 63.3% | 49.8% | 27.4% | | | | | | | | | | | |
| | | DSRS | 63.7% | 53.8% | 32.4% | | | | | | | | | | | |
| | | (Growth) | 0.4% | 4.0% | 5.0% | | | | | | | | | | | |
| | 2.0 | NP | 63.8% | 53.3% | 38.1% | | | | | | | | | | | |
| | | DSRS | 64.5% | 55.0% | 40.8% | | | | | | | | | | | |
| | | (Growth) | 0.7% | 1.7% | 2.7% | | | | | | | | | | | |
| | 4.0 | NP | 64.2% | 55.1% | 42.0% | | | | | | | | | | | |
| | | DSRS | 64.4% | 55.5% | 43.0% | | | | | | | | | | | |
| | | (Growth) | 0.2% | 0.4% | 1.0% | | | | | | | | | | | |
| | 8.0 | NP | 64.6% | 55.5% | 43.7% | | | | | | | | | | | |
| | | DSRS | 64.7% | 55.7% | 43.9% | | | | | | | | | | | |
| | | (Growth) | 0.1% | 0.2% | 0.2% | | | | | | | | | | | |
| 0.50 | 1.0 | NP | 53.0% | 46.7% | 38.8% | 30.1% | 22.2% | 12.7% | | | | | | | | |
| | | DSRS | 53.3% | 47.7% | 40.9% | 34.3% | 26.6% | 16.5% | | | | | | | | |
| | | (Growth) | 0.3% | 1.0% | 2.1% | 4.2% | 4.4% | 3.8% | | | | | | | | |
| | 2.0 | NP | 53.2% | 47.5% | 40.3% | 34.0% | 26.6% | 19.4% | 9.6% | | | | | | | |
| | | DSRS | 53.3% | 48.1% | 41.7% | 35.6% | 28.9% | 21.2% | 11.4% | | | | | | | |
| | | (Growth) | 0.1% | 0.6% | 1.4% | 1.6% | 2.3% | 1.8% | 1.8% | | | | | | | |
| | 4.0 | NP | 53.3% | 48.0% | 41.3% | 35.1% | 29.0% | 22.6% | 15.5% | | | | | | | |
| | | DSRS | 53.3% | 48.1% | 41.9% | 35.9% | 29.5% | 23.4% | 14.3% | | | | | | | |
| | | (Growth) | 0.0% | 0.1% | 0.6% | 0.8% | 0.5% | 0.8% | -1.2% | | | | | | | |
| | 8.0 | NP | 53.3% | 48.2% | 41.6% | 35.8% | 29.5% | 23.9% | 17.5% | | | | | | | |
| | | DSRS | 53.3% | 48.3% | 42.0% | 36.2% | 30.1% | 24.3% | 15.6% | | | | | | | |
| | | (Growth) | 0.0% | 0.1% | 0.4% | 0.4% | 0.6% | 0.4% | -1.9% | | | | | | | |
| 1.00 | 1.0 | NP | 43.3% | 39.3% | 33.2% | 27.9% | 22.7% | 18.2% | 15.1% | 10.9% | 7.4% | 3.5% | 1.6% | 0.8% | 0.1% | |
| | | DSRS | 43.5% | 39.6% | 34.5% | 29.1% | 24.8% | 21.1% | 17.0% | 14.2% | 10.5% | 7.7% | 4.0% | 1.5% | 0.1% | |
| | | (Growth) | 0.2% | 0.3% | 1.3% | 1.2% | 2.1% | 2.9% | 1.9% | 3.3% | 3.1% | 4.2% | 2.4% | 0.7% | 0.0% | |
| | 2.0 | NP | 43.2% | 39.4% | 33.9% | 29.2% | 24.1% | 20.5% | 16.8% | 14.1% | 10.4% | 7.9% | 4.8% | 2.0% | 1.3% | 0.3% |
| | | DSRS | 43.1% | 39.6% | 34.3% | 29.3% | 24.8% | 21.3% | 18.0% | 15.2% | 12.3% | 9.7% | 6.4% | 3.7% | 1.4% | 0.4% |
| | | (Growth) | -0.1% | 0.2% | 0.4% | 0.1% | 0.7% | 0.8% | 1.2% | 1.1% | 1.9% | 1.8% | 1.6% | 1.7% | 0.1% | 0.1% |
| | 4.0 | NP | 43.1% | 39.5% | 34.3% | 29.6% | 24.5% | 21.3% | 17.7% | 15.0% | 12.2% | 9.6% | 6.7% | 4.0% | 2.1% | 1.3% |
| | | DSRS | 42.8% | 39.6% | 34.4% | 29.4% | 25.2% | 21.8% | 18.3% | 15.8% | 12.8% | 10.2% | 7.7% | 4.6% | 2.1% | 0.9% |
| | | (Growth) | -0.3% | 0.1% | 0.1% | -0.2% | 0.7% | 0.6% | 0.6% | 0.8% | 0.6% | 1.0% | 0.6% | 0.0% | 0.0% | -0.4% |
| | 8.0 | NP | 43.1% | 39.5% | 34.1% | 29.8% | 24.9% | 21.7% | 18.4% | 15.6% | 12.9% | 10.1% | 8.0% | 5.6% | 3.2% | 1.6% |
| | | DSRS | 42.9% | 39.5% | 34.3% | 29.8% | 25.2% | 21.9% | 18.6% | 15.8% | 13.2% | 10.6% | 8.0% | 5.4% | 2.7% | 1.3% |
| | | (Growth) | -0.2% | 0.0% | 0.2% | 0.0% | 0.3% | 0.2% | 0.2% | 0.2% | 0.3% | 0.5% | 0.0% | -0.2% | -0.5% | -0.3% |

Table 18: Full experimental results for certified accuracy, SmoothMix, ESG, CIFAR10

| σ | η | Certification method | \multicolumn{14}{c}{Certified accuracy at r} |
|---|---|---|---|
| | | | 0.25 | 0.50 | 0.75 | 1.00 | 1.25 | 1.50 | 1.75 | 2.00 | 2.25 | 2.50 | 2.75 | 3.00 | 3.25 | 3.50 |
| 0.25 | 1.0 | NP | 64.8% | 56.5% | 46.7% | | | | | | | | | | | |
| | | DSRS | 64.6% | 56.5% | 46.5% | | | | | | | | | | | |
| | | (Growth) | -0.2% | 0.0% | -0.2% | | | | | | | | | | | |
| | 2.0 | NP | 64.8% | 56.5% | 46.7% | | | | | | | | | | | |
| | | DSRS | 64.7% | 56.3% | 45.8% | | | | | | | | | | | |
| | | (Growth) | -0.1% | -0.2% | -0.9% | | | | | | | | | | | |
| | 4.0 | NP | 64.8% | 56.4% | 46.8% | | | | | | | | | | | |
| | | DSRS | 64.6% | 56.3% | 46.0% | | | | | | | | | | | |
| | | (Growth) | -0.2% | -0.1% | -0.8% | | | | | | | | | | | |
| | 8.0 | NP | 64.8% | 56.5% | 46.9% | | | | | | | | | | | |
| | | DSRS | 64.6% | 56.1% | 46.5% | | | | | | | | | | | |
| | | (Growth) | -0.2% | -0.4% | -0.4% | | | | | | | | | | | |
| 0.50 | 1.0 | NP | 53.3% | 48.3% | 42.1% | 36.7% | 31.6% | 26.2% | 20.1% | | | | | | | |
| | | DSRS | 53.3% | 48.3% | 42.0% | 36.6% | 31.3% | 25.6% | 18.1% | | | | | | | |
| | | (Growth) | 0.0% | 0.0% | -0.1% | -0.1% | -0.3% | -0.6% | -2.0% | | | | | | | |
| | 2.0 | NP | 53.2% | 48.3% | 42.2% | 36.7% | 31.7% | 26.0% | 20.4% | | | | | | | |
| | | DSRS | 53.3% | 48.3% | 42.0% | 36.6% | 31.2% | 25.6% | 17.6% | | | | | | | |
| | | (Growth) | 0.1% | 0.0% | -0.2% | -0.1% | -0.5% | -0.4% | -2.8% | | | | | | | |
| | 4.0 | NP | 53.3% | 48.3% | 42.1% | 36.6% | 31.6% | 26.1% | 20.3% | | | | | | | |
| | | DSRS | 53.1% | 48.3% | 42.0% | 36.5% | 31.0% | 25.6% | 18.3% | | | | | | | |
| | | (Growth) | -0.2% | 0.0% | -0.1% | -0.1% | -0.6% | -0.5% | -2.0% | | | | | | | |
| | 8.0 | NP | 53.3% | 48.3% | 42.0% | 36.5% | 31.6% | 26.3% | 20.2% | | | | | | | |
| | | DSRS | 53.3% | 48.3% | 42.0% | 36.4% | 31.1% | 25.5% | 18.5% | | | | | | | |
| | | (Growth) | 0.0% | 0.0% | 0.0% | -0.1% | -0.5% | -0.8% | -1.7% | | | | | | | |
| 1.00 | 1.0 | NP | 43.1% | 39.6% | 34.4% | 30.2% | 25.4% | 22.2% | 19.0% | 16.3% | 13.7% | 11.6% | 9.1% | 6.7% | 4.9% | 2.3% |
| | | DSRS | 42.8% | 39.5% | 34.4% | 29.9% | 25.3% | 22.0% | 19.1% | 16.2% | 13.4% | 11.4% | 8.9% | 6.3% | 4.0% | 1.8% |
| | | (Growth) | -0.3% | -0.1% | 0.0% | -0.3% | -0.1% | -0.2% | 0.1% | -0.1% | -0.3% | -0.2% | -0.2% | -0.4% | -0.9% | -0.5% |
| | 2.0 | NP | 43.0% | 39.7% | 34.4% | 30.0% | 25.5% | 22.5% | 19.0% | 16.2% | 13.6% | 11.5% | 9.3% | 6.5% | 4.6% | 2.1% |
| | | DSRS | 42.9% | 39.4% | 34.3% | 29.8% | 25.3% | 22.1% | 18.9% | 16.3% | 13.4% | 11.6% | 8.8% | 6.2% | 4.0% | 1.7% |
| | | (Growth) | -0.1% | -0.3% | -0.1% | -0.2% | -0.2% | -0.4% | -0.1% | 0.1% | -0.2% | 0.1% | -0.5% | -0.3% | -0.6% | -0.4% |
| | 4.0 | NP | 43.0% | 39.7% | 34.6% | 30.1% | 25.5% | 22.4% | 19.0% | 16.3% | 13.7% | 11.6% | 9.4% | 6.6% | 4.8% | 2.4% |
| | | DSRS | 42.9% | 39.5% | 34.3% | 29.8% | 25.4% | 22.1% | 19.0% | 16.2% | 13.6% | 11.5% | 9.2% | 6.0% | 3.9% | 1.8% |
| | | (Growth) | -0.1% | -0.2% | -0.3% | -0.3% | -0.1% | -0.3% | 0.0% | -0.1% | -0.1% | -0.1% | -0.2% | -0.6% | -0.9% | -0.6% |
| | 8.0 | NP | 43.0% | 39.7% | 34.3% | 30.0% | 25.5% | 22.3% | 19.1% | 16.2% | 13.5% | 11.5% | 9.2% | 6.5% | 4.7% | 2.5% |
| | | DSRS | 42.7% | 39.4% | 34.3% | 29.8% | 25.3% | 22.3% | 18.9% | 16.3% | 13.5% | 11.3% | 9.0% | 6.2% | 4.1% | 1.6% |
| | | (Growth) | -0.3% | -0.3% | 0.0% | -0.2% | -0.2% | 0.0% | -0.2% | 0.1% | 0.0% | -0.2% | -0.2% | -0.3% | -0.6% | -0.9% |

## J.3 SUPPLEMENTAL FIGURES FOR STDAUG-GGS MODELS

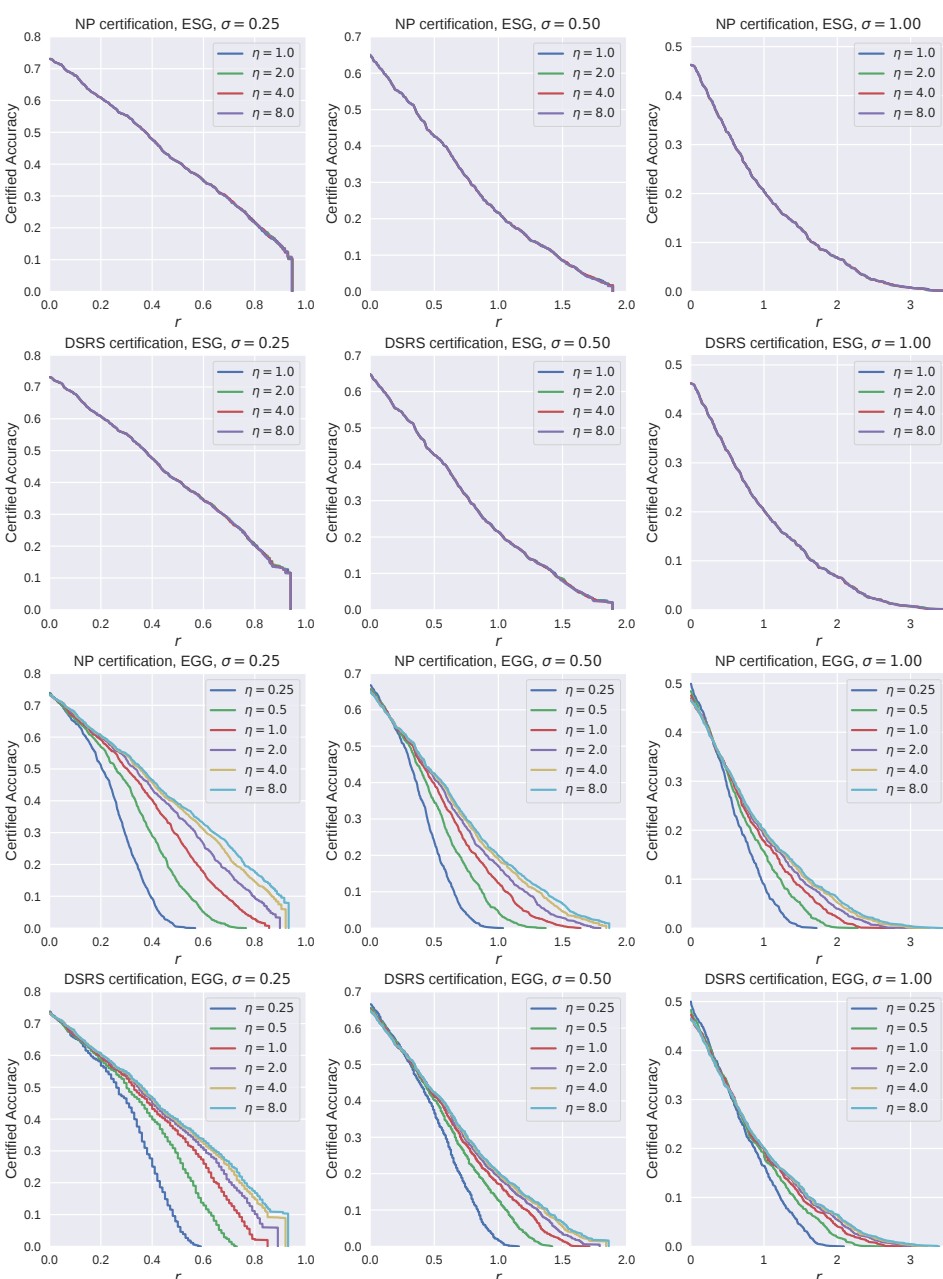

Figure 12: Certified accuracy of $\ell_2$ for standardly augmented models, on CIFAR-10 by General Gaussian, $k = 1530$.

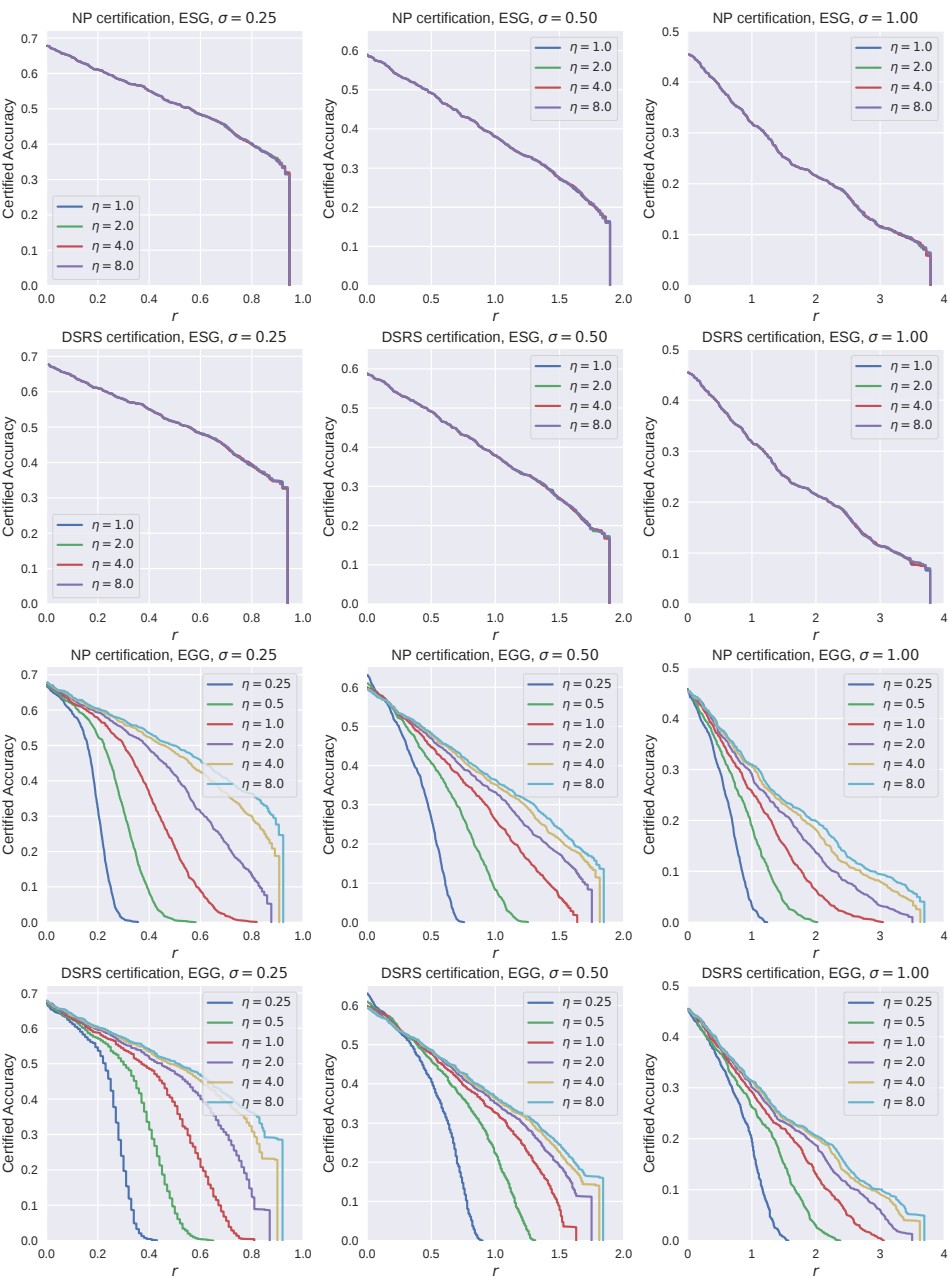

Figure 13: Certified accuracy of $\ell_2$ for standardly augmented models, on ImageNet by General Gaussian, $k = 75260$.

### J.4 SUPPLEMENTAL FIGURES FOR CONSISTENCY-GGS AND SMOOTHMIX-GGS MODELS

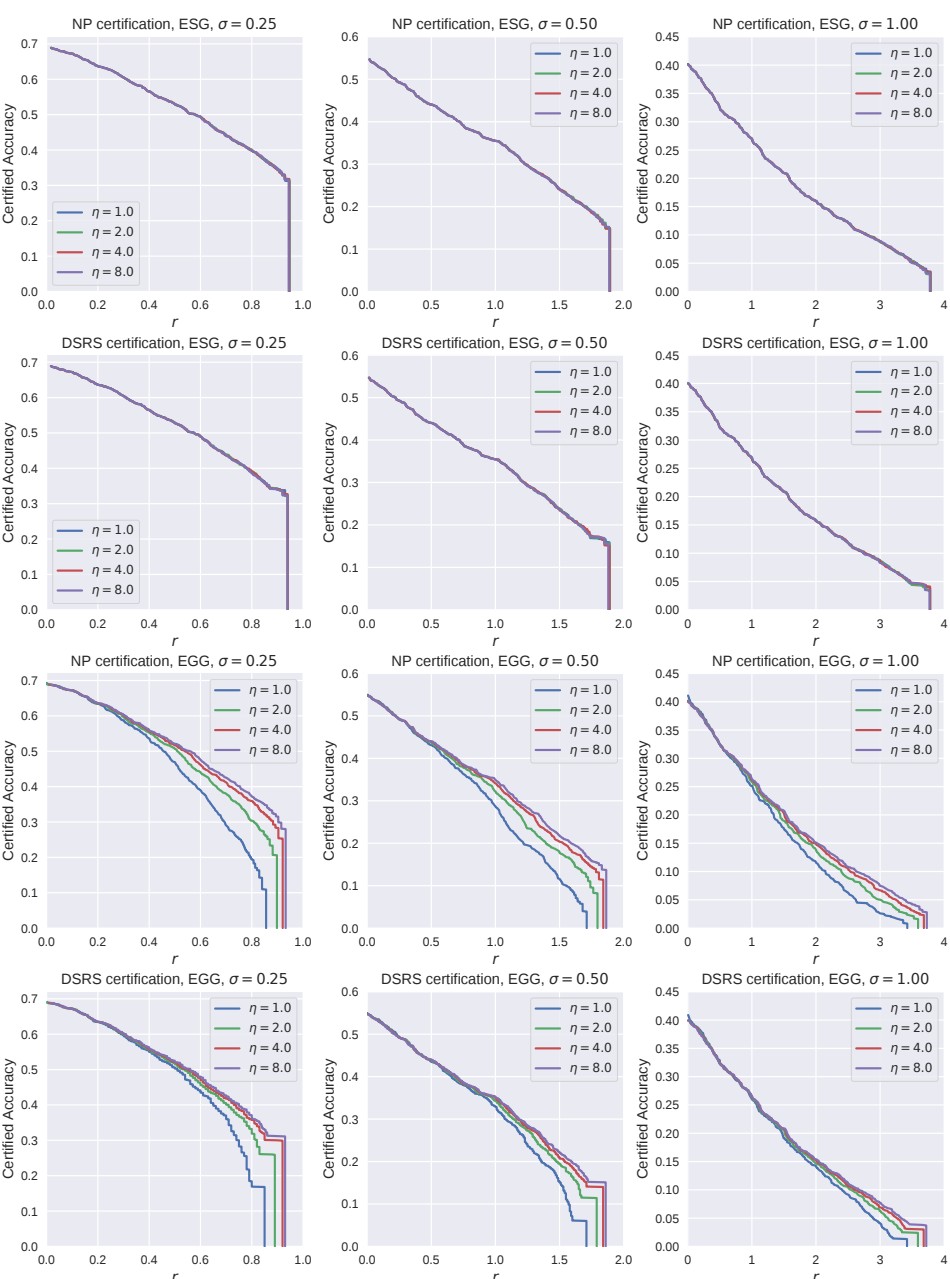

Figure 14: Certified accuracy of $\ell_2$ for Consistency models, augmented on CIFAR-10 by General Gaussian, $k = 1530$.

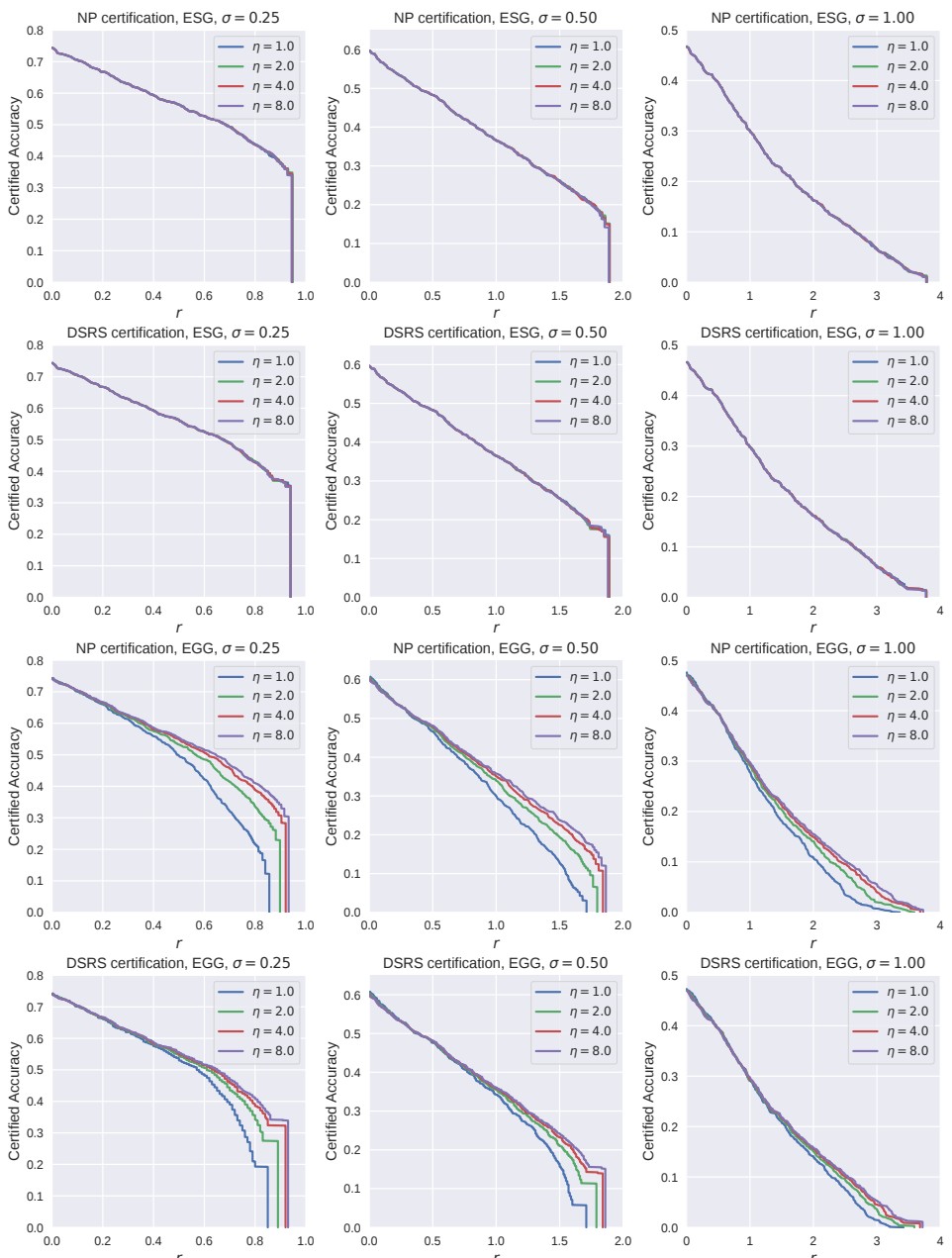

Figure 15: Certified accuracy of $\ell_2$ for SmoothMix models, augmented on CIFAR-10 by General Gaussian, $k = 1530$.

