# OpenReview forum: "Rethinking the Solution to Curse of Dimensionality on Randomized Smoothing"
_ICLR.cc/2024/Conference — Submitted to ICLR 2024_

### Official Review · Reviewer_FGgM · 2023-10-24

**Soundness:** 1 poor
**Presentation:** 1 poor
**Contribution:** 1 poor
**Rating:** 1
**Confidence:** 4

**Summary:**

This work is a follow-up work of Li et al., Double sampling randomized smoothing, in ICML 2022. Compared to the original randomized smoothing, DSRS leverages another random smoothing distribution $Q$, which could improve the certified radius of randomized smoothing. The main novelty of this work is using the "exponential general Gaussian" (EGG) distribution as $Q$, which compared to standard Gaussian changes the exponent from 2 to $\eta$. The authors claim that this can further improve the certified radius of DSRS.

**Strengths:**

N/A

**Weaknesses:**

I recommend rejecting this submission, because (a) this submission in a very large part is essentially the same as Li et al. (2022); (b) there are fundamental errors in the theoretical analysis as well as the experiments.

## (a) Comparing to Li et al. (2022)
Compared to Li et al. published in ICML 2022, the only novelty of this submission seems to be the EGG distribution and related theoretical analysis and empirical verification. The theoretical part, however, is almost the same as Li et al. (2022). Specifically, Theorem 1 focuses on the case where $\eta = 2$, and this statement is the exact same statement as Theorem 2 in Li et al. (2022). Note that the EGG distribution with $\eta = 2$ is equivalent to the generalized Gaussian distribution used in Table 1 of Li et al. (2022). Although this submission does cite Li et al. (2022), its writing seems to claim that Theorem 1 is a novel result, which is definitely false. In fact, the proof of Theorem 1 is essentially the same as Li et al. (2022), Theorem 2:
- Lemma C.1/C.2 (this work) = Proposition F.1 (Li et al.). The only difference is $\eta$, which in my opinion adds almost no additional difficulty to the proof.
- Lemma C.3 (this work) = Lemma F.2 (Li et al.). Only difference is $\eta$.
- Lemma C.4 (this work) = Lemma F.3 (Li et al.). Only difference is $\eta$.
- And so on.

Moreover, Section 4.3 is almost the same as Section 5 in Li et al. (2022), except for $\eta$.

Overall, I don't see any significant difference between this submission before Section 5 and Li et al. (2022), except for $\eta$. The proofs and the algorithms are all essentially the same.

## (b) Does $\eta$ really work? No.
I have pointed out that the only novelty of this submission is $\eta$, so does $\eta$ really break the curse of dimensionality as the authors claim? I don't think so, and I think that there are two fundamental errors:

Theoretically, in Theorem 4 the authors proved an $\Omega(d^{1/\eta})$ lower bound of the certified radius, and claim that this bound gets larger as $\eta$ gets smaller. The problem is that when $\eta$ becomes smaller, then the premise of this result also becomes stronger. Specifically, with the same $\sigma$ and $p$, the $(\sigma, p, \eta)$-concentration assumption is stronger as $\eta$ becomes smaller. So of course with a stronger assumption, the certified radius will become larger. And it seems to me that the authors are aware that $\eta$ does not make the bound tighter, as they mentioned at the top of page 6 as well as in Eqn. (54) in Theorem 4. This even baffles me more why the authors would claim that EGG could lead to "much tighter constant factors" (in the first contributioin).

Empirically, Table 2, if it is correct, shows that EGG only with $\eta = 8.0$ has a very marginal improvement over DSRS. However, before Theorem 1, the authors wrote "all EGG with $\eta \in (0,2)$ have the potential to break the curse", which is not verified by Table 2, and seems to contradict with $\eta = 8.0$ in the experiments.

In conclusion, this work in a very large part is the same as a prior work, its claims are quite misleading, and it contains fundamental errors. Thus, I recommend rejection.

**Questions:**

The authors need to do a very detailed comparison between this submission and Li et al. (2022), including comparison of the methods, the theoretical results and their proofs, the experiment settings and results, and the conclusions.

Also, Theorem 1 is the same as Theorem 2 in Li et al. (2022), so there is no need to write the proof again. The extension of that theorem to $\eta \neq 2$ is very straightforward and only requires a brief explanation. Please also see my comments in the ethics review section.

**Details Of Ethics Concerns:**

This submission, in a very large part before Section 5, is essentially the same as Li et al. (2022). It does raise some research integrity concern. I am not flagging for ethics review, only because it is possible that the first author is a junior student who just started doing research, and I want them to know that a submission with so much similarity with prior work but without proper reference is not acceptable. It is necessary and crucial to clearly label in the paper which results are from prior work, which results are new, and how the new things are different from prior work. For example, if Theorem 1 is exactly the same as a previous result up to paraphrasing, then this previous result must be clearly cited alongside Theorem 1, and the same proof should not be written again. If a theorem is a generalization or an extension of a previous result, then the difference between the two results should be elaborated.

---

> ### Author Response · Authors · 2023-11-11
> **We Thank but Disagree with the Reviewer**
>
> We thank the reviewer for the careful reading on the paper. However, according to reviews, our contributions are largely and comprehensively underesitmated.
> 1. The first title of this paper is actually 'Effects of Exponential Gaussian Distributions on Double Sampling Randomized Smoothing'. The major concern of this paper, as the reviewer mentioned, is on how changing the exponent $\eta$ influences the theoreical and practical results of certifications from (DS)RS. We emphasize that we are not trying to hide this intuition for the work.
> 2. About our Theorem 1. The Theorem 1 is definitely novel, or at least, absolutely a generalization for the conlusion from Li et al. (2022). This is because $\eta$ does not only appear in the concentration assumption, but also in the smoothing distributions. In other words, definition 2 and theorem 1 are distinctive things, where when $\eta=2$, the definition 2 reduces to the same thing in Li et al. (2022) (their definition 3). But theorem 1 is far from a copy, it is describing the fact that under the same concentration assumption in Li et al. (2022), $\eta \in (0,2)$ provides tigher constant factors than $\eta=2$, which is what we specifically describe in 'Study on the constant factor' in page 6. We maintain that this section should be more carefully read and understood.
> 3. About Theorem 4. Different with Theorem 1, the existence of theorem 4 is to show that the exponent $\eta$ can be formally introduced into the lower bound, rather than to prove the $\Omega({\sqrt{d}})$ once again. We have also learned the the concentration assumption here is stricter than that in theorem 1 (see Remark in page 24), and its fundamental equivalence to theorem 1 (top of page 6) or its logical naturalness. These are also why we put theorem 4 in the appendix instead of the text. However, we do not believe the equivalence of $\sigma \sqrt{d}$ to $\sigma_s d^{\frac{1}{\eta}}$ is an obvious conclusion to the community without any clarification, or only showing Eqn. (62)-(64).
> 4. The experimental results. Firstly, we don't understand why our method, that brings holistic improvements on extensive mainstream datasets and training methods to General Gaussian distribution is called 'margined'. Is it because the increment on CIFAR10 is not strong? If so, please also consider the ImageNet results. Overall, the contribution of our work is far beyond breaking the SOTA, because the deep and meticulous study on the smoothing distributions reveal the potenitial restrictions to the current solution the curse of dimensionality and the double-sampling method, and taking other $\eta$ effectively alleviates the limitation of the origin method, both theoretically and pratically.
> 5. The seemlingly contradictory conclusions for theories and experiments. Actually, we have deeply explored and explained this phenomenon in the text. In short, our theories tell that the smaller $\eta$ the better, while the experiments tell that the larger $\eta$ the better. One key insight of our paper is revealing that these findings are not incompatible, which comes from the concentration assumption. See the 2nd paragraph of analyses in page 9. Briefly, if the concentration assumption strictly holds (p=1), smaller $\eta$ is better, otherwise (p<1) larger $\eta$ is better. Our theory is based on the $p=1$ assumption, but the real classifiers can only satisfy $p<1$, which is the decisive reason of the seemingly opposite conclusions. Figure 10 further shows this extreme sensitivity to $p$ of the concentration assumption, that there is a large gap between effects of even $p=0.99999$ and $p=1$.
> 6. Ethic concerns. Our theorem 1 is defintely not found by Li et al. (2022), and is in essence considering the generalization from 2 to real numbers for DSRS. As we have mentioned, definition 2 is indeed the concentration assumption in Li et al. (2022), but we have clarified this in the next paragragh of definition 2. In fact, we feel deeply astonished, confused, and discontented on the statement of so-called '(im)proper reference' and ' integrity concern'.
>
> Finally, the generalization from $\eta=2$ to $\eta\in\mathbb{N_+}$ is absolutely not a trivial improvement like deleting $2$ and press an another number on the keyboard to obtain the results. On the contrary, it needs thorough understanding on Li 2022, methods in Yang 2020 to figure out why we can improve the framework from the perspective of exponent of distribution. This perspective, as far as we are concerned, are never systematically investigated by the community. Moreover, contrary to what the reviewer said, there do exist difficulties for our derivation. In addition to the significant bar on mastering the precedent work, some core details for calculating the certified radius are blank on the Internet, especially on EGG and ESG distributions themselves. Furthermore, stepping into unknown fields for human beings, going deep and offering valuable insights inherently contains big challenges.

---

> ### Comment · Reviewer_FGgM · 2023-11-11
> **Response**
>
> I thank the authors for their response.
>
> 1. I would like to apologize for my earlier misunderstanding. Now I can see the difference between Theorem 1 in this work, and Theorem 2 in Li et al. (2022). However, given that the difference is so subtle, I would like to suggest the authors particularly highlight this difference before or after this result. It was not until the fifth time I read these two statements that I finally found the difference, especially given that they have the exact same bound. Also, I still think that Theorem 2 of Li et al. (2022) has not been properly cited in your Theorem 1. I would suggest you change the name of Theorem 1 to "Generalization of Theorem 2 of Li et al. (2022)".
>
> 2. I would like to remove my comments of ethics concerns (openreview does not allow removing ethics comments though). But still, I think that the proof in your Appendix C is way too similar to the proof in Appendix F of Li et al. (2022). I don't know why there is a need to copy so many things from a previous paper. It would be sufficient for the authors to just describe how to extend Li et al.'s result from $\eta = 2$ to other $\eta$, for example using Yang et al.'s results, and I don't see why this has any significant technical difficulty.
>
> 3. And still, I cannot see how your results give tighter constant factors. Theorem 1 has the exact same bound as Li et al. (2022). And as for Theorem 4, like I mentioned in my review, it requires a stronger assumption.
>
> 4. Regarding the experiments and your fourth point: First, I do not understand why an improvement on CIFAR-10 from 57.4% to 57.6%, and sometimes even worse than DSRS, should be called anything other than "marginal" (by the way, reviewers Ba1f and 2qpb also call it "marginal"). And it is not surprising at all that EGG can be *better* than DSRS, because EGG with $\eta =2$ is exactly DSRS so you can never be worse than DSRS, and by adding one degree of freedom in hyperparameter tuning EGG can get better results without doubt. My point is that this improvement, from any scientific lens, is not significant.
>
> 5. Second, your theorem focused on $\eta \in (0,2)$, but your experiment shows that $\eta = 8.0$ is the best, which is contradictory to your theoretical results. I cannot understand your analysis on this contradiction in page 9, as well as the fifth point in your response. Let me make my question really simple: This paper is proposing "EGG", your theory says that "EGG with $\eta \in (0,2)$ is good", but your experiments say that "EGG with $\eta \in (0,2)$ is actually not good, but EGG with $\eta = 8$ is good". So what is exactly your proposal? EGG with $\eta \in (0,2)$ which is not verifiable in your experiments, or EGG with $\eta = 8$ which is not justified by your theory? Or if you are proposing both, do you have a practical, implementable method of selecting this $\eta$, other than testing over all $\eta$ and hand-picking the best one? If I select $\eta$ with a validation set, could I get the best $\eta$? If this is the case, then you will need to show this with experimental results.
>
> 6. Finally, I still recommend adding a detailed and careful comparison between your work and Li et al. (2022) given their great similarity, including a comparison on the theoretical results, their proofs and the experimental setups and results. This would definitely help anyone reading this work who is not extremely familiar with Li et al. (2022).

---

> > ### Author Response · Authors · 2023-11-15
> > **Response**
> >
> > We thank the reviewer for the professional reply. We now answer all the questions in the feedback.
> > 1. We have now added another reference in the next paragraph of Theorem 1.
> > 2. Though our proofs may look similar to Li et al. 's at the first glance, ours are not simply copying theirs.
> >
> > $\cdot$ For some cases, Li’s proofs were not simplified enough. For instance, compared to the left column on page 32 of their paper, we omitted Eqn. (1)-(5) in our process.
> >
> > $\cdot$ They omitted some essential details for the derivations. For example, we supplemented the details of PDFs in Appendix A.1, the derivations in Eqn. (66)(80)(86), and the details to obtain the Lambert W function in Eqn. (69)-(71).
> >
> > $\cdot$ The proof of our theorem 3 (for ESG) is based on the ‘similar proof’ (for EGG). Leaving out the proof of theorem 2 and only referring the readers to Li et al. (2022) can make our proof baffling. The proof of Th. 3 is necessary, since it further uses the DSRS framework and contains nontrivial cases.
> >
> > In summary, our proofs have multiple details different from Li et al. (2022), which we believe helps the community, especially the beginners, better understand DSRS and Yang et al.'s method, and promote other studies in this field, considering there are few materials relating to exploring distributions on RS currently.
> >
> > 3. Our derivation for the constant factor is based on further using Li et al. 's method. To explain their method better, we have added Appendix F.2, please check it if necessary. Their method utilizes the following properties:
> >
> > $\cdot$ If $R$ can be certified, then all $r<R$ can be certified.
> >
> > $\cdot$ A certified radius provided by sampling probabilities P and Q in DSRS is greater than that provided only by Q.
> >
> > $\cdot$ The certified radius provided by the randomized smoothing method increases monotonically with the shifted sampling probability. In other words, the shifted probability and the certified radius correspond one-to-one and are positively correlated. This is why performing binary search on radius is viable.
> >
> > Here, for the DSRS method, the certified radius provided by P and Q takes the certified radius from only Q as a natural lower bound. Similar to the form in Eqn. (117), for $\mathcal{Q}$, we have Problem (29). To be more clear, we can let the shifted probability
> >
> > $f(r) = \mathbb{E}_u\Psi(\frac{T^2-(\sigma_g(2u)^{\frac{1}{\eta}}-r)^2}{4r\sigma_g(2u)^\frac{1}{\eta}})$.
> >
> > That means, for any radius $r_0$, if $f(r_0)>0.5$, then all $r<r_0$ are certified. This is right how Li et al. proved their theorem: if $f(0.02\sigma{\sqrt{d}})>0.5$, then the $\Omega({\sqrt{d}})$ lower bound is proved. In our work, we discover that for each EGG distribution (with hyperparameters $\eta, k, d$), the maximum certified radius (so as the correspoding constant factor) can be determined by binary search (Algorithm 1). Concretely, $f(r)$ can be transformed into Eqn. (52) (actually, (52) > 0.5 is a sufficient condition for f(r) > 0.5), where performing binary search on the factor $\mu$ is direct. Letting $r=\mu\sigma\sqrt{d}$, we finally obtain Figure 2, showing that $\mu$ monotonically grows when $\eta\in(0,2)$ shrinks (we did not draw $\eta=2$ in the figure, but it does obey the rule). These results were much more regular than our expectation, as chances are high that the curves are highly nonconvex.
> >
> > 4. Please check point 2 in the global response. In addition, though our $\eta=8.0$ result does not significantly improve the baseline from DSRS, it is not even a fair comparison because that baseline is obtained when $\mathcal{Q}\neq$TEGG. Our point is, for $\eta=2.0$ with $\mathcal{Q}=$TEGG, $\eta=8.0$ brings systematic improvements. Plus, we do not totally agree with the statement on ‘degree of freedom’, despite this is also our initial intuition. For instance, the results of ESG show the certification remains almost inert to $\eta$ (Table 3).
> >
> > 5. First, for more exploration on $\eta$, please check point 2 in the global response. Our conclusion can be restated as: if the concentration assumption strictly holds (i.e. B=1 (B is from Problem 4), or 100% accuracy for predicting under noise distribution $\mathcal{Q}$), smaller $\eta$ is better. Otherwise, if the concentration assumption does not hold strictly (i.e. B<1, all the real classifiers satisfy this), then larger $\eta$ is better. Figure 10 has more on this point, where we see if $B=1$, $\eta=1$ performs better, but if $B<1$ (even if extremely close), $\eta=2$ performs better. The performance for $B=1$ and $B=0.99$ differs quite obviously, which is why we say there is a gap between 0.99 and 1. Besides, Figure 1 shows these as well.
> >
> > 6. We agree with the reviewer and have added the comparison to Li et al. (2022) in the first paragraph of Appendix I, including comparisons on theorems, experimental results and proofs. For experimental setups, we wrote details for the changes in the first submitted version, such as the threshold $T$ (please see Eqn. (118)).

---

> > > ### Comment · Reviewer_FGgM · 2023-11-19
> > > **Reviewer Response**
> > >
> > > I thank the authors for their newest response.
> > >
> > > ### 1. Tighter constant factor
> > > If I understand it correctly, the authors are saying that the tightest $r$ that EGG with $\eta$ can certify is given by $F(r; \eta)> 0 $, and this $F$ is given by Eqn. (29). However, Eqn. (29) does not have a closed-form solution, so the authors use binary search to find the best $r$. I have not entirely grasped how this could work out, but this seems interesting. I have the following suggestion to the authors:
> > > - This discussion, especially Eqn. (29), should appear in the main body Theorem 1 because "tighter constant factor" is one of your main claims. It should not be the readers' job to delve so deep into the proof to find out why your main claim makes sense.
> > > - Binary search seems only a method of finding solution to Eqn. (29), and other solvers are also feasible. So "binary search" does not seem to be the main point here, but rather Eqn. (29) $F(r; \eta)> 0 $ is what really makes your bound tighter. Right now "binary search" is confusing, especially after (4) where you wrote "through binary searching on $\lVert \delta \rVert_2$, we obtain the certified radius of example $x_0$"; when I first read this, I had zero idea of how this binary searching would work.
> > > - It would be very helpful if the authors could provide some intuition on why a smaller $\eta$ could lead to a tighter constant factor. Let's consider the extreme case $\eta = 0$, then $G$ and $G_t$ become polynomially tailed distribution. The argument is basically: Suppose the base classifier is exponentially smooth (such that it satisfies the $(\sigma, p, 2)$-concentration property), then using another polynomially tailed distribution to smoothify the base classifier could lead to a larger certified radius. Without having checked the proof very carefully, I have very little intuition how this could work out.
> > >
> > > ### 2. Point 2 of general response
> > > If I understand it correctly, what the authors are saying with the new table is that a larger $\eta$ would not only help DSRS, but also the original NP certification. I have the following questions:
> > > - This would be very interesting if this is true. In this case, why do you choose to only focus on DSRS, instead of the more general and well-known original NP certification where probably the effect of $\eta$ could be understand more easily? Why don't you just say that you improve the constant factor of NP certification?
> > > - "If the concentration property strictly holds, then smaller $\eta$ is better; if not, then larger $\eta$ is better":
> > >     - First, how could one verify whether the concentration property strictly holds or not in practice? I don't think this property is very easy to verify, because it requires a conditional probability to be strictly 1, which cannot be verified by simple Monte-Carlo.
> > >     - Second, can the authors give an example of a real dataset, where the concentration property strictly holds and a smaller $\eta$ works better?
> > >     - Third, it is still hard for me to see why this argument makes sense. For example, suppose there is a base classifier $f$ such that the concentration property strictly holds. One can easily break this property by changing the value of $f$ in a very small neighborhood of $x_0$ with a very small measure. Then, according to the authors, it suddenly changes from "a smaller $\eta$ is better" to "a larger $\eta$ is better". I really cannot see why this is correct.

---

> ### Author Response · Authors · 2023-11-21
> **Response with Updated Paper**
>
> We thank the reviewer for the time spent on careful reading, deep thinking and writing the response on the paper.
>
> **The constant factor.**
>
> This is the right understanding. Taking Eqn. (30) as the example, we let
>
> $F(r; \eta)=\mathbb{E}_u\Psi(\frac{T^2-(\sigma_g(2u)^{\frac{1}{\eta}}-r)^2}{4r\sigma_g(2u)^\frac{1}{\eta}})-\frac{1}{2}$.
>
> Then, for a given $r$, if $F(r; \eta) > 0$, $r$ is certified, else $r$ is not certified. For the points:
>
> $\cdot$ We have reorganized page 6 in the latest version: adding details for Eqn. (30) as Eqn. (8), and moving Theorem 1 to the appendix.
>
> $\cdot$ We are sorry for the ambiguous presentation, and have deleted the confusing ‘binary search’ in the text.
>
> $\cdot$ About little $\eta$. We have considered the mechanisms carefully on why small $\eta$ provides better theoretical results, but find this is beyond our intuition at present.
>
> (1) One possible reason is that a concentration assumption becomes more and more strict for EGG when $\eta$ gets smaller. For this, we can consider fixing a $T$ in Figure 5 (left). Then, for EGG with smaller $\eta$, more mass of the distribution $(r<T)$ is contained by $T$ (meaning larger proportion of perturbed examples will be correctly classified on the base classifier). However, intuitively, the certified radius is not uniquely influenced by their PDF, it can also be affected by other geometric properties. We did not mention this in the text, since it is not systematic enough.
>
> (2) By introducing an additional distribution, it is rational that the NP certification can be improved. This is because the NP lemma is always considering the most conservative case and constructing the 'worst smoothed classifier' based on known information  (such as probabilities from MC sampling) of the given classifier. DSRS improves NP by bringing in extra information for the given classifier, and is capable of constructing a 'better worst smoothed classifier' to provide certification (as well as certified radius) for examples. The intuition to improve certification with additional information also appears in other papers. For instance, [1] and [2] improve the certifications by introducing gradient information of the base classifier.
>
> **Why focus on DSRS.**
>
> Yes, from our experiments, increasing $\eta$ in EGG improves the certification for both NP and DSRS. More specifically, our **theoretical analysis** sets $B=1$ (the concentration assumption is perfectly satisfied), finding smaller $\eta$ in EGG provides tighter lower bounds **for DSRS**. Besides, our **experiments** on real datasets and real base classifiers ($B<1$, the concentration assumption is not perfectly satisfied) show larger $\eta$ in EGG gives better certifications **for NP and DSRS**. In other words, $\eta$ performs great for DSRS both theoretically and practically, but we do not include specialized theories on NP. This is why we take DSRS as the principle line.
>
> **The concentration assumption.**
>
> $\cdot$ On the verification and practical use of concentration assumption. At the current stage, the perfectly satisfied concentration assumption is an ideal case, that we may only realize it by numerical simulation, such as setting $B=1$ when computing. Therefore, it is beyond our ability to train a classifier that gives absolutely right predictions for a noised real dataset, even if the level of the noise is restricted.
>
> $\cdot$ Why our conclusions make sense.
>
> (1) The large gap between 0.99 and 1 also appears in Li et al. (2022) (their figure 2a). In that figure, DSRS w/N = 1e7 is equal to B=0.9999993, DSRS w/N = 1e6 is equal to B=0.999993 in our paper (their probabilities was simulating the Clopper-Pearson lower bound, while we directly set them). As they showed, the large 'gap' between $B=0.99$ and $B=1.0$ exists, and the $B=1$ can provide very large certified radius, but DSRS w/N = 1e7 can only provide $r<3$.
>
> (2) The essential reason for the gap is our Lemma C.3 or Li et al. (2022)'s Lemma F.2. The key point here is, if the assumption holds at 0.99, then it only takes effect on the specified distribution (say, Gaussian). But if it holds at 1, then it takes effect for every distribution, because the specified distribution has positive density everywhere.
>
> (3) Our conclusions are based on rigor derivation and experiments, which are checked multiple times (codes included). We also want our results to look concise, but we must obey science and practice.
>
> Refs:
>
> [1] Mohapatra J et al. Higher-order certification for randomized smoothing[J]. NeurIPS, 2020, 33: 4501-4511.
>
> [2] Levine A et al. Tight second-order certificates for randomized smoothing[J]. arXiv preprint arXiv:2010.10549, 2020.

---

> > ### Comment · Reviewer_FGgM · 2023-11-22
> > **Reviewer Response**
> >
> > I thank the authors for their discussion. I think I have sort of understood why the authors claim that $\eta$ should be smaller when Eqn. (6) holds with probability 1, and $\eta$ should be larger when Eqn. (6) holds with probability less than 1. Here is my understanding:
> > 1. When $\eta$ is smaller, $Q$ becomes more polynomially tailed, that is it is less smooth; when $\eta$ is larger, $Q$ is smoother.
> > 2. "Eqn. (6) holds with probability 1" is in fact a very strong assumption, because regardless of $\eta > 0$, $S(\sigma, \eta)$ is always supported on the entire Euclidean space. Thus, Eqn. (6) is equivalent to $f(x) = y _0 \text{ a.e. } \lVert x - x _0 \rVert _2 \le T$. In other words, $f(x)$ is essentially a constant within a ball centered at $x _0$ with radius $T$.
> > 3. Under this very strong assumption, one can choose $Q$ to be a less smooth distribution, and get a large certified radius as long as this ball has a sufficiently large measure under probability measure $Q$. There is no point to use a large $\eta$, because a larger $\eta$ will make $Q$ more "concentrated" at $x _0$, but it is already assumed that $f$ is a constant within a ball so why bother using a concentrated distribution? In other words, a smaller $\eta$ allows more "exploration" outside this ball.
> > 4. However, if Eqn. (6) holds with probability less than 1, then $f$ does not have this nice constant ball as a foundation. As a result, choosing $Q$ to be a less smooth distribution will not give a larger statistically certifiable radius. And if the probability Eqn. (6) is much lower than 1, then a more concentrated $Q$ (with a larger $\eta$) could increase the ACR.
> >
> > Based on the above understanding, I have the following comments:
> > 1. It seems that Eqn. (6) is a very strong assumption, because it is equivalent to saying that $f$ is a.e. constant within a ball centered at $x _0$. However, if such a ball is **assumed**, then it naturally implies that the model is adversarially robust, so what is the point of certification at all? Thus, my feeling is that Definition 2 is not a reasonable assumption. To make it more reasonable, the probability of Eqn. (6) should be strictly less than 1, say 0.99.
> >
> > 2. I think that "using a larger $\eta$ can lead to a larger certified radius for the original NP certification" would be a super interesting and groundbreaking result if it could be proved. I am not sure if this has been proved in a prior work. If not and if the authors could prove this, then I would give a score of 10 without any doubt. I don't think this conjecture has anything to do with the specific formulation of DSRS. Basically this conjecture is saying: If the model is known to be smooth upon smoothifying with some distribution $P$, then using another smoother distribution $Q$ to smoothify the model could lead to a higher certified radius. This is a very pure and well-posed statistics question, and it is really such a result that could be "going deep and offering valuable insights".
> >
> > 3. I don't think using a small $\eta$ could be beneficial at all, under any reasonable assumptions. Indeed, the authors' experiments show that using a small $\eta$ hurts the performance. Like I said in my earlier response, the extreme case of a small $\eta$ is $\eta = 0$, where $Q$ would become polynomially-tailed, and assuming that the model is only smooth w.r.t. an exponentially-tailed distribution, I couldn't see how this $Q$ would be helpful at all. Theorem 1 gives the illusion that a small $\eta$ would help, only because it is based on a not very reasonable assumption (Definition 2), which by itself has already implied that the model is robust.
> >
> > To sum up, I will still vote to reject this submission, because I believe that the main theoretical result (Theorem 1) which says that a smaller $\eta$ could improve certification is not sound and potentially misleading. The core reason is that its foundation is a very strong assumption. The empirical evidence of my argument is the experimental results provided by the authors themselves.
> >
> > My suggestion to the authors is to consider the following research question:
> > - Suppose Eqn. (6) holds with probability strictly less than 1, say 0.99. Then, for the original NP certification, prove that using a larger $\eta$ could provably lead to a larger certified radius. To prove this, one needs to show a lower bound of the CR for a large $\eta$, and an upper bound for a small $\eta$, and then show that the former is greater than the latter.
> >
> > If such a result could be proved, then I would rate "strong accept" without any doubt, and I can assure the authors that any reviewer sufficiently familiar with this field would rate the same. I can imagine that proving this could be extremely hard, but given that the authors aspire to do deep research, this is a direction I would strongly recommend.

---

> ### Author Response · Authors · 2023-11-23
> **Response**
>
> We thank the reviewer for replying again. Here we have the following points to clarify:
>
> 1.Smoother smoothing distribution $\nRightarrow$ smoother classifier or better certification
>
> The word 'smooth distribution' looks ambiguous. From the context, we speculate it refers to differentiability of the PDF of distributions. If this is the case, then the smoothness of distribution has no direct correlation with the smoothness of the classifier, and also, does not guarantee greater certified accuracy. A simple example: Laplace (ESG, $\eta=1$) is less smooth than Gaussian (ESG, $\eta=2$), but they basically provide similar certifications (Table 3). Likewise, in DSRS, we do not think truncated general Gaussian is smoother than general Gaussian. In other words, the smoothness of a distribution is not a significant factor for choosing the supplementary distribution.
>
> 2. The understanding of the mechanism is inaccurate.
>
> (1)  EGG with larger will not be more concentrated at $x_0$. Instead, it will be concentrated in a shell, at a distance from $x_0$ (which is the thin-shell phenomenon, Zhang et al. (2020)).
>
> (2) In our work, we emphasize the existence of a gap between $B=0.99$ and $B=1$, and the distinct performance of EGG under very similar settings. The reviewers' intuition on $B=1$, is very close to our tentative explanation, which we believe is insufficient for understanding the behavior of EGG under $B<1$.
>
> 3. The concentration assumption.
> The concentration assumption is actually an extrapolation for an observed phenomenon, rather than fabricated from the thin air. It originated from Li et al. (2022), which is essentially enlightened by the observation of some real robust classifiers. For this, we recommend the reviewer to their Figure 4 and their Appendix J.1. In that figure, each blue curve represents the performance of a test data from ImageNet perturbed by 1000 random Gaussian noise on a pre-trained robust classifier. For example, a point (190, 0.99) on a curve means the robust classifier has a 99% accuracy on 1000 perturbed points, and the $l_2$ length of perturbations can reach up to 190. In addition, there are abrupt downfalls in most blue curves, most of which begin to plunge after the green peak in the figure, which is where the Gaussian noises gather. The observations above are why the concentration assumption was proposed. Moreover, even if the concentration assumption strictly holds, it does not mean the classifier is robust, because this robustness is not only a local property, but also limited to specific inputs.
>
> 4. We thank the reviewer for providing a prospective studying project for us, we will consider it in our future research.
>
> Finally, We remark that our contributions are underestimated, and are not fully understood by the reviewer. But we still thank the reviewer for the plenty of time spent on our work.

---

> ### Comment · Reviewer_FGgM · 2023-11-23
> **Reviewer Response**
>
> **Note:** The Program Chairs have decided to reopen discussion on openreview until Dec 4. The following is my response to the authors' newest comments I wrote earlier.
>
> I thank the authors for their discussion.
>
> 1. "Smooth" distribution: By "smooth", I was not talking about the PDF of the distribution. I was just saying that the distribution decays faster since it is more exponentially tailed, so when smoothifying a classifier with this distribution, it puts more weights on its neighborhood than points far away.
>
> 2. "Concentration": By "more concentrated at x", I meant that more masses of the distribution will be close to x. The authors are absolutely right that most masses will be distributed close to a shell, which further indicates why Definition 1 is not a reasonable assumption. Like I pointed out earlier, Eqn. (6) holding with probability 1 is equal to $f$ being a.e. constant in a ball, so if this shell lies within this ball, then the smoothified classifier will output the same class with very high probability, so that there is no point in doing certification with Monte-Carlo sampling.
>
> 3. Appendix J.1 in Li et al. (2022): I don't think the assumption in Li et al. (2022) is reasonable. Like I said earlier, "Eqn. (6) holding with strict probability 1" cannot be statistically verified with Monte-Carlo sampling. Appendix J.1 in Li et al. (2022) was still conducting Monte-Carlo sampling, so it cannot verify or justify "Eqn. (6) holding with strict probability 1". This assumption is equivalent to $f$ being a.e. constant within a neighboring ball, and with this strong assumption there is no point in doing certification. Like I said in the last message, a more reasonable assumption could be P(Eqn. (6)) > 1 - epsilon for some very small but positive epsilon. For instance, even P(Eqn. (6)) > 0.999999 is statistically verifiable, if given enough samples.
>
> 4. Why I am still voting to reject:
>     - Soundness: My biggest concern is that the theoretical part of this work suggests using a small $\eta$, but your own experiments show that using a small $\eta$ is actually harmful, and the explanation is not convincing. As I have mentioned several times, I cannot see any plausible reason why a small $\eta$ would help. In my last reply, I pointed out that the illusion that a small $\eta$ could help might be caused by the not very reasonable concentration property, which requires P(Eqn. (6)) to be strictly 1. And indeed, the authors also observed a huge gap between P(Eqn. (6)) = 1 and P(Eqn. (6)) = 0.999999. I pointed out that the main claim that "using a small $\eta$ is beneficial" is unverifiable and unsound because it is based upon a flawed assumption, and thus suggested the authors to change your concentration assumption to P(Eqn. (6)) = 0.999999, from which you might be able to prove that a larger $\eta$ is better.
>     - Contributions: The authors argue that the "contributions are underestimated". You are fully entitled to your own view, but from my perspective the analysis in this submission is not deep enough, mostly because your own empirical results contradict with your own theoretical results, and your explanation for this contradiction is not convincing at all. I am also pretty sure that many readers who are familiar with this field will get the same feeling. My take is that for any practical scenario, a small $\eta$ is harmful, while a large $\eta$ might be good, and I articulated my intuition in my last response.
>     - Comparing to prior work: Finally, this work is still too similar to Li et al. (2022), and I suggest the authors include a full paragraph of comparison between the two papers. And I think that the notations in Li et al. (2022) are good enough, so I am not sure why there is a need to use a completely new set of notations. There is no need to hide the fact that your work is largely based on a previous work. It is totally fine if your new contributions are sufficient and deep enough.

---

### Official Review · Reviewer_2qpb · 2023-10-31

**Soundness:** 3 good
**Presentation:** 3 good
**Contribution:** 3 good
**Rating:** 8
**Confidence:** 3

**Summary:**

The paper extends the DSRS framework to a family of distributions called the Exponential General Gaussian (EGG) distributions. Assuming Li et al.'s concentration condition, the authors show that the proposed framework is capable of producing better certified radii (in the constant factor). Furthermore, the authors show that for stronger concentration assumptions, the proposed method can produce polynomially better certified radii. The authors also provide a truncated version of the distributions for the additional distribution in the DSRS framework and give an algorithm to solve the optimization problem to compute the certified radii under the extended framework. Finally, the authors show the empirical advantage of the proposed method on real-life datasets, CIFAR10 and Imagenet.

**Strengths:**

- The suggested framework extends the DSRS framework to a larger family of distributions that can provide theoretically better bounds (better in the constant multiplier) under Li et al.'s concentration assumption. The paper also proposes a more general concentration assumption under which the proposed distributions can polynomially better certified radii.
- The proposed method is able to provide better certified accuracy than the current state-of-the-art method on both CIFAR10 and Imagenet datasets. At larger radii, the certified accuracy on Imagenet beats the current SOTA by 4-6%.

**Weaknesses:**

- The empirical performance on the CIFAR10 dataset is only marginally better than the current SOTA. It is not clear why this happens.

**Questions:**

Please check the weaknesses section.

Typos
- When introducing the PDFs, I think you wanna say "We let $S(\sigma, \eta)$ and $G(\sigma, \eta, k)$ be the probability density functions (PDFs) of ESG and EGG, respectively". Currently, the ordering is wrong.
- Similarly for substitution variances, "We let $\sigma_s$ and $\sigma_g$ be the substitution variances of ESG and EGG, respectively"

---

> ### Author Response · Authors · 2023-11-15
> **Response**
>
> We thank the reviewer for the time as well as the encouraging feedback. We would like to deal with the questions as follows.
>
> $\cdot$ We have some additional materials for experiments on CIFAR10 in point 2 of the global response, please check it if necessary. In fact, the comparison between DSRS and EGG, $\eta=8.0$ is not totally fair, because DSRS takes $\mathcal{Q}\neq$ TEGG when providing the baseline for CIFAR10. If we pay attention to EGG, $\eta=2.0$, we can see the improvement brought by EGG, $\eta=8.0$ is significant.
>
> $\cdot$ We have corrected these typos accordingly.
>
> We hope our response has resolved your concerns on the paper, and would welcome any further discussion.

---

### Official Review · Reviewer_Ba1f · 2023-11-01

**Soundness:** 2 fair
**Presentation:** 2 fair
**Contribution:** 2 fair
**Rating:** 5
**Confidence:** 4

**Summary:**

This work conducted some theoretical study on the lower bound of $\ell_2$ certified radius with a tighter constant, compared to the previous work DSRS. The experiments results verified the theory in some cases.

**Strengths:**

1. This work contains some theoretical finding and the corresponding empirical experiments to verify these finding.
2. The paper is easy to follow.

**Weaknesses:**

1. As the title already suggested, the paper is mainly to study the certified radius in high dimensional space. Therefore, making the constant tighter is not of significant interest for research purposes because the dominate term is always the dimension $d$. Making the order of $d$ smaller is definitely of interest.
2. The author needs to include the variance for each experiments. By theory, EGG distribution should yield certified accuracy no smaller than DSRS. However, EGG has smaller certified accuracy in some cases in Table 2, which suggest there is a variance issue. So it's good to report variance.
3. Marginal improvement: for most of the cases in Table 2, the improvement is no larger than 0.5%. This coincides with the intuition mentioned in weakness 1: making the constant tighter in high dimensional space will only yield incremental improvement.

**Questions:**

Minor:
1. It's better to write Neyman-Pearson Lemma instead of NP lemma, as NP can represent many terms.
2. In theorem 1, "let d be a sufficient(ly) large input dimension..."
3. In theorem 1, are the $\eta$ in P and Q the same? It also seems surprised to me that the lower bound doesn't depend on $k$ and $\eta$. This means any values of $k$ and $\eta$ will lead to the same lower bound.

---

> ### Author Response · Authors · 2023-11-15
> **Response**
>
> We provide thanks to the reviewer for the effort put into the review, and give our feedback as follows.
>
> **Weaknesses:**
> 1. One of our initial motivations for the work, quite similar to what the reviewer mentioned, is to improve the order of $d$, and we did have the trial in our work (see the second paragraph on page 6). In our derivation, the exponent $\eta$ can be introduced into the order of $d$ to form $\Omega(d^{\frac{1}{\eta}})$. Despite it being proved an equivalent bound to $\Omega(\sqrt{d})$, our theorem shows EGG with $\eta\in(0,2)$ can improve the constant factors efficiently. We believe our theoretical results, though do not break the  $\sqrt{d}$ order, can propel and enlighten future studies in this topic.
>
> In fact, the initial title of this paper is 'Effects of Exponential General Gaussian on Double Sampling Randomized Smoothing'. That is to say, we emphasize a lot on thinking from the perspective of the exponent, and its effect on the (double sampling) randomized smoothing framework. In addition to the investigation on curse of dimensionality, our work also provide substantial insights on RS and DSRS including:
>
> $\cdot$ Our experimental results for EGG show seemingly contrary rules with theoretical analysis. Specifically, our theorems reveal the lower bound improves as the $\eta$ in EGG gets smaller, while our experiments on real-world datasets show larger $\eta$ provides better certifications. However, this seemingly contrary phenomenon does not mean we made mistakes in the work. According to our further study, we find how $\eta$ influences the certifications is largely decided by whether the concentration assumption strictly holds. That is, if the concentration assumption strictly holds (i.e. B=1 in Problem (4), which is the case of our theorem 1 and Figure 1(a)), the lower bound provided by DSRS improves as the $\eta$ decreases. Otherwise, if the concentration assumption does not strictly hold (i.e. B<1 in Problem (4), which is the experimental case because no classifier perfectly satisfies the assumption), the certifications provided by EGG improves as the $\eta$ increases. Our Figure 10 further clarifies this point by simulating the cases where $B$ is very close to 1. It is clear in the figure that if $B=1$, $\eta=1$ performs better, but if $B<1$ (even if extremely close), $\eta=2$ performs better.
>
> $\cdot$ Our results for ESG demonstrates that a mainstream view in the randomized smoothing community can be augmented. Concretely, the mainstream view (for example, Yang et al. (2020)) believes Gaussian distribution is the best distribution to provide certification in the RS framework. However, our finding is that many members of the ESG distribution can provide certification as great as Gaussian (Gaussian is a special case of ESG when $\eta=2.0$).
> We would like to recommend reading point 1 in the global response for more information on backgrounds and motivations for this paper.
>
> 2. It is a bit ambiguous for 'variance', here we talk about both the cases.
>
> $\cdot$ If it refers to the variance of each distribution, such as $0.5, 1.0$. We can ensure that all variances in this work are aligned by keeping $\mathbb{E}r^2$ a constant (the convention from Yang et al. (2020) and Li et al. (2022)). This is also why we have the substitution variance in Table 1.
>
> $\cdot$ If it refers to the variance between repetitive experiments. In our work, we find the errors on certified accuracy caused by randomness of experiments usually range between 0.0%-0.3%. It will definitely be great to show results of multiple experiments, but its impact may be limited for the theme of this paper. Actually, though our results for $\eta=8.0$ result does not significantly improve the DSRS baseline, the comparison is not fair. In Li et al. (2022), they reached the baseline by setting $\mathcal{Q}\neq TEGG$. If we consider a fair comparison, say comparing $\eta=2.0$ with $\eta=8.0$, then our improvement is significant.
>
> 3. Please check point 2 in the global response. From our perspective, compared to $\mathcal{P}=EGG, \mathcal{Q}=TEGG, \eta=2.0$,  the setting $\mathcal{P}=EGG, \mathcal{Q}=TEGG, \eta=8.0$ provides holistic improvements, no matter which dataset (CIFAR10, ImageNet) and training method (standard, SmoothMix, Consistency) are used. Besides, our results can be further enhanced by letting $\eta$ larger (16, 32). We do not show them in the paper because their errors are more difficult to control than $\eta=8.0$.
>
> **Questions:**
>
> 1, 2. We have improved these representations in the latest paper.
>
> 3. Yes, $\eta$ is the same for $\mathcal{P}$ and $\mathcal{Q}$. As for $\eta, k$, they definitely influence the bound (please check Eqn (51). and Table 4). Theorem 1 only shows the same bound provided by General Gaussian (Li et al. (2022)) can also be provided by EGG, and the section 'study on the constant factor' (page 6) further shows the factor grows as $\eta$ shrinks, for $d-2k$ except 1.

---

> > ### Comment · Reviewer_Ba1f · 2023-11-19
> >
> > I would like to thank the authors for the response. I have read the response and would like to keep my original rating mainly based on weakness 1.

---

### Official Review · Reviewer_Dgzf · 2023-11-07

**Soundness:** 3 good
**Presentation:** 2 fair
**Contribution:** 3 good
**Rating:** 5
**Confidence:** 4

**Summary:**

This work extends double sampling randomized smoothing (DSRS) to smooth with exponential general Gaussian (EGG) and exponential standard Gaussian (ESG) distributions. The authors derive certified robust radii for their proposed methods, and experimentally show that the performance of their methods surpass that of standard DSRS on CIFAR-10 and ImageNet.

**Strengths:**

1. The Introduction is concise and motivates the problem well, and the contributions are clearly outlined.
2. The Experiments are thorough.
3. The Preliminaries section provides a nice concise introduction to the formalisms at hand.
4. The theoretical results (namely, Theorems 1 and 2) appear to be novel.
5. The proposed method's performance (in terms of certified radii) appears to meet/exceed prior state-of-the-art.

**Weaknesses:**

1. What do you mean by "point-to-point certified accuracy" in the Abstract? This terminology may be unclear to the reader, so I suggest replacing it or clarifying what it means.
2. "We let $\sigma_s$ and $\sigma_g$ be the substitution variances of EGG and ESG, respectively." I think you mean "...variances of ESG and EGG, respectfully."
3. "We let... be the probability density functions (PDFs) of EGG and ESG, respectively." I think you mean "... of ESG and EGG, respectfully."
4. "...our theoretical analysis shows EGG distributions can be prospective in providing much tighter lower bounds..." What do you mean by prospective? I think this sentence needs rewriting.
5. Why is $\eta$ restricted to be in a finite set in Theorem 1? Why is $d-2k$ restricted to be less than 30? I don't see where the number 30 appears in the proof of Lemma C.3 at all.
6. When introducing DSRS in Section 3, the values of $A,B$ are somewhat glossed over, leaving the reader to wonder what they are and where they come from. You mention that you can estimate them through Monte Carlo sampling, but no formulas are given to perform those estimates. How are $A$ and $B$ defined mathematically? Also, you state "In a nutshell, we find the maximum $\lVert\delta\rVert_2$ that makes the worst probability...". This is somewhat vague. Are you saying that you maximize over $\delta$ in an "outer-maximization" after solving the minimization (4)? In other words, it would be good to clarify how exactly the certified radius of DSRS is defined mathematically, and how that definition relates to (4).
7. It looks like your statement of Lemma C.3 should be "at labeled example $(x_0,y_0)$", not just "at input $x_0$".
8. The way Theorem 2 is stated, it does not appear that you are solving the dual problem (8), but rather giving specific formulas for the objective and constraints based on specific EGG density functions. If this is indeed the case, then you should re-word your descriptions to be more accurate (i.e., do not call your result a closed-form solution to the optimization problem). How are you actually solving the dual in practice (for the dual variables $\nu_1,\nu_2$)? This should be made more clear to the reader.
8. Overall, the paper is a bit hard to follow at times, as some of the language is cryptic or vague.

**Questions:**

See my questions above in "Weaknesses".

---

> ### Author Response · Authors · 2023-11-15
> **Response**
>
> We thank the reviewer for the time and the consideration. Here we provide point-to-point explanations for the concerns.
> 1. Using 'point-to-point certified accuracy', we were trying to describe our experimental results as better than the baselines at almost every certified radius $r$. For this, please check figures for EGG on page 46-49, where the curves of EGG,$\eta=8$ envelop almost all their corresponing EGG,$\eta=2$. To reduce ambiguity, we now use the term certified accuracy following the convention in the RS community.
>
> 2,3. We have corrected these typos.
>
> 4. Here we wanted to show EGG is a potential choice for providing better certification results, where the word 'prospective' is indeed inaccurate. Currently, this sentence in the text has been rewritten into 'Overall, theoretical analysis above shows taking EGG as the smoothing distribution in DSRS, we are likely to obtain much tighter lower bounds for the certified radius on base classifiers, ...'. We hope the current expression clarifies our intention.
> 5. About $\eta, d, k$. The EGG distributions, defined as $\mathcal{G}(\sigma, \eta, k)$ in the paper, are all considered in discrete combinations of the parameters.
>
> $\cdot$ The reason why $\eta$ belongs to a finite set is that it appears as an argument in the gamma function (for example, Eqn. (12)). As far as we are concerned, it is currently intractable to analyze the continuous property of $\eta$ considering the gamma function, which is also why Li et al. (2022) only considered discrete cases in their Proposition F.6.
>
> $\cdot$ Yes, we do not need to add the restriction $d-2k\in [1,30]\cap \mathbb{N}$ if we just want to prove Lemma C.3, since $d-2k \geq 1$ can all satisfy Eqn. (24). The setting of 30 is following Li et al. (2022) (see their Proposition F.6), where they restricted $d - \frac{k}{2} \in$ {$0.5, 1.0, \cdots, 15.0$}. As we mentioned, the analysis on continous property of EGG distributions is intractable, so only the discrete combinations of $d, k, \eta$ is considered. Concretely, we show the proof for $(d-2k, \eta) \in$ {$1, 2, \cdots, 30$} $\times$ {$1, \frac{1}{2}, \frac{1}{3}, \cdots, \frac{1}{50}$}. For each combination of $d-2k$ and $\eta$, Theorem 1 proves its ability to give the $0.02\sigma\sqrt{d}$ lower bound. Here we take  $d-2k = 2, \eta=\frac{1}{2}$ as an example. Specifically, we need to substitute $\mu=0.02, d-2k=2$ and $\eta=\frac{1}{2}$ into Eqn. (52), and check if the value is greater than 0.5. By checking Table 4, we find the value is 0.782, then the $0.02\sigma\sqrt{d}$ lower bound is proved. To sum it up, Table 4 contains all the discrete combinations of parameters we study (i.e. different EGG distributions), where $d, k, \eta$ appear simultaneously. $d, k, \eta$ also exist in other equations such as Eqn. (51). They do not appear in Theorem 1, because all the limited combinations of $(d-2k, \eta) \in$ {$1, 2, \cdots, 30$} $\times$ {$1, \frac{1}{2}, \frac{1}{3}, \cdots, \frac{1}{50}$} have been proved capable of offering the $\Omega(\sqrt{d})$ lower bound.
>
> 6. About Eqn. (4).
>
> $\cdot$ A and B are both probabilities that the base classifier gives the right prediction for $x_0$, under different noise distributions.
>
> $\cdot$ Just as the reviewer understands, there is an ‘outer-maximization’ after we obtain the best function $\tilde{f}_{x_0}$. This ‘outer-maximization’ is quite simple so that we can solve it only by binary searching on $||\delta||_2$.
>
> To further clarify the definitions of $A,B$, we have rewritten the paragraph below Eqn. (4).
>
> 7. We have improved the statement of Lemma C.3 accordingly.
> 8. We agree with the review on 'closed-form', and now use 'integral form' to describe our theorem in the latest paper. To clarify the process of solving the DSRS problem, we have added Appendix F.2 in the latest paper, which is a short introduction considering the complete derivation is somewhat complicated and voluminous. For $\nu_1, nu_2$, we get their values from the DualBinarySearch algorithm from Li et al. (2022).
> 9. We are sorry for the confusion and puzzle. In fact, we have been trying to clarify the mechanisms in the paper as understandable as possible since we started to write this paper. This paper has been revised for more than 15 times before being submitted to the conference. However, limited to our writing skills, and the innate complexity of this work, sometimes it is still difficult to follow. We welcome further discussions on any confusion on or technical details of the paper, and promise we will carefully handle the questions at our best.

---

> > ### Comment · Reviewer_Dgzf · 2023-11-15
> >
> > I thank the authors for their responses and their updates to the manuscript and its presentation. However, as the improvements over prior methods (Li et al., 2022) are marginal (Table 2) and the theoretical novelties also appear to be marginal based on my understandings of Reviewer FGgM's qualms, I maintain my original score.

---

### Author Response · Authors · 2023-11-15
**Global Response**

We offer thanks to all the reviewers for their insightful questions and suggestions. To better clarify the background of the study, and to deal with issues commonly concerned, we would like to provide the following response.
1. Why is investigating the General Gaussian distribution meaningful?

Here we only talk about studies in the randomized smoothing (RS) community. The General Gaussian distribution was initially introduced into the community by Dinghuai Zhang et al. (2020). In their work, they discovered that the high-dimensional Gaussian distribution is almost concentrated on a thin shell in the space. Trying to solve this issue, they introduced the General Gaussian distribution (in our notations, its PDF $\propto r^{-2k}exp(-\frac{r^2}{\sigma_g^2})$), which can overcome the 'thin-shell' property of Gaussian distribution. However, their experimental results (certified accuracy provided by General Gaussian) showed no significant improvements to the Gaussian results. Zhang et al. 's work is later questioned by Greg Yang et al. (2020) (their paragraph 6, page 17), where Yang et al. pointed out Zhang et al's method is invalid, that General Gaussian can not bring in any improvements to Gaussian. This dispute, as far as we know, remains unresolved till now. Then, General Gaussian reappears in Li et al. (2022), and performed great in the DSRS framework. The dispute about the inconclusive issue on General Gaussian, combined with the its great behaviors under DSRS, triggers our strong interest in it. Does General Gaussian really work for the randomized smoothing framework? If so, how on earth does it work? Our work provides one perspective to view the disputation: from the exponent of distributions.

2. The experimental results on CIFAR10.

According to our experimental results, the improvement brought by EGG ($\eta=8.0$) is not significant compared to the baseline from DSRS (which was not provided by General Gaussian (GG)). However, this work is mainly focused on enhancing the General Gaussian. This is because from the perspective of distribution, the optimal one is still Gaussian, neither GG nor our EGG defeats the Gaussian baseline. One key contribution of this paper is that we reveal the certification (as well as certified accuracy) provided by EGG improves monotonically when $\eta$ gets larger. For this, we have additional experimental results:
| $\eta$ in EGG | 0.25  | 0.5   | 1.0   | 2.0   | 4.0   | 8.0   | 16.0  | 32.0  |
|---------------|-------|-------|-------|-------|-------|-------|-------|-------|
| $ACR_{NP}$      | 0.213 | 0.277 | 0.341 | 0.401 | 0.448 | 0.478 | 0.495 | 0.502 |
| $ACR_{DSRS}$    | 0.276 | 0.338 | 0.392 | 0.427 | 0.469 | 0.489 | 0.499 | 0.503 |
| Growth from DSRS| 29.4% | 22.0% | 15.1% | 9.0%  | 4.7%  | 2.3%  | 0.8%  | 0.2%  |

In the table above, all the data are obtained from the same base classifier, which is augmented by Gaussian distribution ($\sigma=0.5$). The results show that:

$\cdot$ If we take EGG as the distribution for sampling and certifying on the a fixed base classifier, the certification results monotonically improves with the $\eta$ in EGG, while the increment gained by DSRS shrinks with $\eta$.

$\cdot$ The improvement of certification brought by $\eta$ is not limitless. For both NP and DSRS certification, the ACR grows slower and slower when $\eta$ gets larger. This rule can also be observed from Figure 1(b), Figure 4(a) and Figure 4(b) in our text.

$\cdot$ DSRS provides decreasing improvements for EGG distributions as the $\eta$ improves, which is highly similar with the results of Gaussian, where DSRS provides negative growth (Figure 4(d)). The reason remains unknown, but probably because when $\eta$ gets larger, EGG becomes more and more similar to Gaussian (Figure 5).

$\cdot$ Currently, the mainstream view in the community believes Gaussian distribution is the best distribution to provide certification in the randomized smoothing framework (only with Neyman-Pearson lemma). However, our results in the table are essentially challenging this view. For example, EGG with $\eta=32.0$ provides $ACR=0.502$, while Gaussian provides $ACR=0.491$ (Zhai et al. (2020)).

The table above is not reported in our paper because setting very large $\eta$ can introduce uncontrollable numerical error, so that we are not absolutely sure about accurate results for large $\eta$. In summary, though we only provide experimental results for $\eta=8.0$ out of conservatism and reliability, we believe the motivation and observation behind our results is valuable and enlightening for the community.

---

### Meta-Review · Area_Chair_3gTW · 2023-12-15

**Metareview:**

This work extends double sampling randomized smoothing to exponential general Gaussian and exponential standard Gaussian distributions. The reviewers raised concerns regarding the significance of the contributions in this work, particularly the marginal improvements over existing methods and the absence of substantial theoretical novelties. Despite the authors' rebuttal efforts, the provided response did not convince the reviewers to increase their scores.

**Justification For Why Not Higher Score:**

I reached this decision by evaluating the contributions and novelty of the work, taking into consideration both the reviews and the responses from the authors.

**Justification For Why Not Lower Score:**

N/A

---

### Decision · Program_Chairs · 2024-01-16

Reject